# Explicit silicate cycling in the Kiel Marine Biogeochemistry Model, version 3 (KMBM3) embedded in the UVic ESCM version 2.9

Karin Kvale[1,2], David P. Keller[1], Wolfgang Koeve[1], Katrin J. Meissner[3,4], Chris Somes[1], Wanxuan Yao[1], and Andreas Oschlies[1]

[1]GEOMAR Helmholtz Centre for Ocean Research Kiel, Düsternbrooker Weg 20, D-24105 Kiel, Germany
[2]present address: GNS Science, 1 Fairway Drive, Avalon 5010, PO Box 30368, Lower Hutt 5040, NZ
[3]Climate Change Research Centre, Level 4 Mathews Bldg., UNSW, Sydney, NSW, AU
[4]ARC Centre of Excellence for Climate Extremes, Australia

**Correspondence:** K. Kvale (kkvale@geomar.de)

**Abstract.** We describe and test a new model of biological marine silicate cycling, implemented in the Kiel Marine Biogeo-chemical Model version 3 (KMBM3), embedded in the University of Victoria Earth System Climate Model (UVic ESCM) version 2.9. This new model adds diatoms, which are a key component of the biological carbon pump, to an existing ecosystem model. This new model combines previously published parametrisations of a diatom functional type, opal production and ex-port with a novel, temperature-dependent dissolution scheme. Modelled steady-state biogeochemical rates, carbon and nutrient distributions are similar to those found in previous model versions. The new model performs well against independent ocean biogeochemical indicators and captures the large-scale features of the marine silica cycle to a degree comparable to similar earth system models. Furthermore it is computationally efficient, allowing both fully-coupled, long-timescale transient simula-tions, as well as "offline" transport matrix spinups. We assess the fully-coupled model against modern ocean observations, the historical record since 1960, and a business-as-usual atmospheric $CO_2$ forcing to the year 2300. The model simulates a global decline in net primary production (NPP) of 1.4% having occurred since the 1960s, with the strongest declines in the tropics, northern mid-latitudes, and Southern Ocean. The simulated global decline in NPP reverses after the year 2100 (forced by the extended RCP 8.5 $CO_2$ concentration scenario), and NPP returns to 98% of the pre-industrial rate by 2300. This recovery is dominated by increasing primary production in the Southern Ocean, mostly by calcifying phytoplankton. Large increases in calcifying phytoplankton in the Southern Ocean offset a decline in the low latitudes, producing a global net calcite export in 2300 that varies only slightly from pre-industrial rates. Diatom distribution moves southward in our simulations, following the receding Antarctic ice front, but diatoms are out-competed by calcifiers across most of their pre-industrial Southern Ocean habitat. Global opal export production thus drops to 75% of its pre-industrial value by 2300. Model nutrients phosphate, sili-cate, and nitrate build up along the Southern Ocean particle export pathway, but dissolved iron (for which ocean sources are held constant) increases in the upper ocean. This different behaviour of iron is attributed to a reduction of low-latitude NPP (and consequently, a reduction in both uptake and export and particle, including calcite, scavenging), an increase in seawater temperatures (raising the solubility of particulate iron), and stratification that "traps" the iron near the surface. These results are meant to serve as a baseline for sensitivity assessments to be undertaken with this model in the future.

# 1 Introduction

It has become apparent in recent decades that the representation of elemental cycles of nutritive elements (N, P, Si, Fe) is important in order to simulate critical biogeochemical feedbacks (Heinze et al., 2019). Here, we describe an updated representation of ocean biogeochemistry (the Kiel Marine Biogeochemical Model, version 3; KMBM3) embedded in an earth system model, the University of Victoria Earth System Climate Model version 2.9, second level updates (UVic ESCM; Weaver et al., 2001; Eby et al., 2009), to which we have added a silicate cycle.

Silicate (considered here as any anion that combines silicon, Si, and between 2 and 4 oxygen molecules) is important for simulating ocean biogeochemistry for several reasons. It regulates the growth of diatoms, which are phytoplankton responsible for 40% of the particulate carbon export in the modern ocean (Jin et al., 2006). Fluctuations in the relative abundance of diatoms might therefore affect air-sea $CO_2$ exchange, with a globally significant influence on atmospheric $CO_2$ concentrations over millennial and longer timescales (e.g. Matsumoto et al., 2002; Renaudie, 2016). High latitude diatoms with low N:P ratios

have been demonstrated to exert global control on the nitrogen budget (Weber and Deutsch, 2012), and are speculated to exert seasonal control on carbonate chemistry through competition with calcifiers (Merico et al., 2006). Silicic acid ($Si(OH)_4$, and $SiO(OH)_3^-$) distributions are tightly controlled by diatom primary production (biogenic silica, or opal; $SiO_2$) and subsequent dissolution (e.g., only about 34% global average ocean silicic acid is preformed, as opposed to biologically regenerated; Holzer et al. 2014). First attempts to constrain the marine silicate cycle demonstrate that representations of the silica cycle are well

suited to parameter optimisation (Holzer et al., 2014; Pasquier and Holzer, 2017). Due to diatoms' important role in biological carbon export, evaluating simulated silicate and diatoms presents a unique opportunity to constrain ocean carbon cycling.

    In the following sections we describe our new model of explicit diatoms and silicate cycling in the ocean. As far as we are aware, this model is unique among ecosystem models embedded within earth system models in that it combines established parameterisations of a diatom functional type (i.e. fast growth rates and inefficient nutrient utilisation, Tréguer et al. 2018) with

a novel temperature-dependent opal dissolution parameterisation and bottom-up and top-down competition for resources with explicit, fully independent calcifying phytoplankton and nitrogen fixing diazotrophs. These assumptions bias KMBM3 diatoms towards large species in the well-mixed high latitudes, and early seasonal succession. Modelled silicate boundary conditions include hydrothermal silicate inputs and a benthic transfer function to approximate sediment sequestration. A ballast model applied to the calcifiers (Kvale et al., 2015b) further differentiates KMBM3, as does the ability to operate in an "offline"

parameter optimisation framework (Kvale et al., 2017; Yao et al., 2019).

# 2 Model Description

## 2.1 Physics

The University of Victoria Earth System Climate Model (UVic ESCM) version 2.9 is a coarse-resolution ($1.8° \times 3.6° \times 19$ ocean depth layers) ocean-atmosphere-biosphere-cryosphere-geosphere model. Model structure and physics are described in

Weaver et al. (2001), Meissner et al. (2003) and Eby et al. (2009). We use the second level updates of version 2.9. Additionally, an ideal age tracer (Koeve et al., 2015) is merged into this version of the model to allow for explicit tracing of water mass age.

## 2.2 Biogeochemistry

Development of the Kiel Marine Biogeochemistry Model has progressed in recent years. Beginning with a base composition of nitrate, phosphate, DIC, and alkalinity as well as general phytoplankton, diazotrophs and one zooplankton functional type
(Schmittner et al., 2005, 2008), Keller et al. (2012) updated zooplankton grazing and included a seasonally cycling iron mask (KMBM1). To this version prognostic iron tracers were implemented by Nickelsen et al. (2015) (KMBM2), and prognostic $CaCO_3$ and calcifying phytoplankton were added by Kvale et al. (2015b). Somes et al. (2013) included benthic denitrification and Muglia et al. (2017) included hydrothermal iron in MOBI, a separate branch of the UVic ESCM biogeochemistry. Kvale and Meissner (2017) examined the sensitivity of primary production in the model to changes in the light attenuation parameter
following a correction in the calculation of light availability (Partanen et al., 2016). To the Keller et al. (2012) version Kvale et al. (2017) added offline transport matrix method (TMM) capability, allowing for rapid matrix-based computation of model mean ocean tracer states and coupling to alternative physical models (Khatiwala et al., 2005; Khatiwala, 2007). The current model description unites divergent code, assimilating the Nickelsen et al. (2015); Kvale et al. (2015b, 2017) models as well as the benthic denitrification code of Somes et al. (2013) and hydrothermal iron of Muglia et al. (2017), and expands the code to
include explicit diatoms and a silicic acid tracer.

With up to 17 model tracers, KMBM3 biogeochemistry is now fairly complex. An abbreviated description is provided in the following sections, with only the relevant model developments described here. All simulations use the fully-coupled UVic ESCM framework.

We introduce a new phytoplankton functional type to which have been assigned key physiological attributes of diatoms. This
new model version therefore contains phytoplankton functional types "slow growing low latitude phytoplankton (LP)", "slow growing phytoplankton which can fix nitrogen when necessary, including diazotrophs (DZ)", "moderate growth phytoplankton, including calcifiers (CP)", and "fast growing diatoms (DT)". Zooplankton contribute to $CaCO_3$ production. Prognostic tracers include nitrate, phosphate, oxygen, DIC, alkalinity, iron, $CaCO_3$, silicic acid, and particulate forms of iron and organic detritus. The new model schematic is shown in Figure 1 and state variables are given in Table 1. Diatoms compete with the other
functional types for light and nutrients. The diatoms' growth is also limited by dissolved silicic acid availability; they implicitly produce opal that instantly remineralises back into dissolved silicic acid throughout the water column. To ensure absolute model conservation of silica, any opal that is lost due to implicit burial at the seafloor is replaced by external sources (prescribed atmospheric dust deposition, sediment release via a benthic transfer function, prescribed hydrothermal vent release, and river fluxes that can be set to compensate for any remainder; see below for details). The phytoplankton and detritus production and
remineralisation are linked to nutrients through fixed Redfield stoichiometry using a base unit of mmol nitrogen $m^{-3}$.

In the following model description, notation will generally follow the symbols used in Kvale et al. (2015b), with the abbreviations LP, CP, DT, DZ representing the phytoplankton functional types, and "Z" representing zooplankton when a distinction is necessary. The most important biogeochemical model parameters are listed in Tables 2 to 5. The model description here covers

only the most relevant equations, and equations that have changed in this newest version; please see Kvale et al. (2015b);
Nickelsen et al. (2015); Keller et al. (2012); Schmittner et al. (2005), and Schmittner et al. (2008) for original references and a complete description of the other equations.

### 2.2.1 Sources and Sinks

Tracer concentrations ($C$) vary according to:

$$\frac{\partial [C]}{\partial t} = T + S + B \tag{1}$$

with $T$ including all transport terms (advection, diffusion, and convection), $S$ representing all source and sink terms, and $B$ representing air-sea interface boundary (including virtual evaporation-precipitation correction) fluxes and hydrothermal inputs, where relevant.

### 2.2.2 Phytoplankton

Phytoplankton ($X$ representing all types except diazotrophs) biomass (in units of mmol nitrogen m$^{-3}$) source and sink terms are:

$$S(X) = J_X X - G_X - \mu_X^* X - m_X X \tag{2}$$

where growth ($J$), mortality ($m$), and fast recycling ($\mu^*$) rates are described below, and losses to zooplankton grazing ($G$) are described in the zooplankton equations. The diazotroph equation is similar except that it does not include a linear mortality loss, only a loss to fast recycling.

As in Keller et al. (2012), the maximum possible growth rate of phytoplankton ($J_{\max}$) is a modified Eppley curve (Eppley, 1972), and is a function of seawater temperature ($T$) in degrees Celsius, an e-folding temperature parameter $T_b$, and iron limitation ($u_{\mathrm{Fe}}$) modifying a growth parameter ($a$). This parameterisation assumes sufficient iron is required for the utilisation of other nutrients (Galbraith et al., 2010; Keller et al., 2012; Nickelsen et al., 2015).

$$J_{\max} = a \exp^{\frac{T}{T_b}} u_{\mathrm{Fe}}. \tag{3}$$

As in earlier versions of the model, diazotroph growth rate is calculated following ordinary phytoplankton (now LP) and then multiplied with an additional handicap.

Nickelsen et al. (2015) assigned a constant iron half saturation ($k_{\mathrm{Fe}}$) to diazotrophs but in ordinary phytoplankton this parameter varied as a function of biomass ($X$) to implictly represent different cell sizes in the model. Because KMBM3 contains multiple phytoplankton functional types we revert to a prescribed but unique $k_{\mathrm{Fe}}$ for all functional types and calculate iron limitation as:

$$u_{\mathrm{Fe}} = \frac{[\mathrm{Fe}]}{k_{\mathrm{Fe}} + [\mathrm{Fe}]}. \tag{4}$$

The maximum potential growth rate is then multiplied by the minimum nutrient limitation term ($u$) for nitrate, phosphate, and dissolved silicic acid (the latter for diatoms only) to calculate potential growth under nutrient limitation but replete light, where $k_N$, and $k_P$ are fixed half saturation constants unique to each functional type (and $k_P$ is calculated from 16 P to 1 N Redfield stoichiometry). These equations are applied to obtain maximum possible growth rates as a function of temperature and nutrients following Liebig's law of the minimum:

$$u_{NO_3^-} = \frac{[NO_3^-]}{k_N + [NO_3^-]} \tag{5}$$

$$u_{PO_4^{3-}} = \frac{[PO_4^{3-}]}{k_P + [PO_4^{3-}]} \tag{6}$$

$$u_{Si} = \frac{[Si]}{k_{Si} + [Si]} \tag{7}$$

Silica limitation uses the empirical fit to experimental data scaling of $k_{Si}$ in mmol Si m$^{-3}$ from Aumont et al. (2003):

$$k_{Si} = 0.8 \, \text{mmol Si m}^{-3} + 7.2 \, \text{mmol Si m}^{-3} \frac{[Si]}{k_{Si}^* + [Si]} \tag{8}$$

with a $k_{Si}^*$ value adopted from Aumont et al. (2003) of 30 mmol Si m$^{-3}$.

The potential growth rate under limited light availability ($J_I$) but replete nutrients is calculated as:

$$J_I = \frac{J_{max} \alpha^{chl} \theta I}{(J_{max}^2 + (\alpha^{chl} \theta I)^2)^{\frac{1}{2}}} \tag{9}$$

Nickelsen et al. (2015) introduced this parameterisation to the model and discussed it at length; interested readers are recommended to read their Section 2.3.2. In the above equation, $\alpha^{chl}$ is the initial slope of the photosynthesis versus irradiance ($I$) curve in chlorophyll units:

$$\alpha^{chl} = \alpha^{chl}_{min} + (\alpha^{chl}_{max} - \alpha^{chl}_{min})u_{Fe}, \tag{10}$$

and $\theta$ is a Chl:C ratio (noting, chlorophyll is not a prognostic variable, Nickelsen et al., 2015):

$$\theta = \theta_{min} + (\theta_{max} - \theta_{min})u_{Fe}. \tag{11}$$

The approximation of $\theta$ is simplistic and neglects other factors, e.g. irradiance, which can affect the ratio. For simplicity, the same maximum and minimum values of $\alpha^{chl}$ and $\theta$ are used for all functional types.

Light attenuation by algae biomass and by coccoliths is included in the calculation of available irradiance at each depth level:

$$I = I_{z=0} \, \text{PAR} \, e^{-k_w \tilde{z} - k_c \int_0^{\tilde{z}} (LP + CP + DZ + DT)dz - k_{CaCO_3} \int_0^{\tilde{z}} [CaCO_3]dz} \left(1 + a_i(e^{-k_i(h_i + h_s)} - 1)\right) \tag{12}$$

where PAR stands for the photosynthetically available radiation, $k_w$, $k_c$, $k_{CaCO_3}$, and $k_i$ are the light attenuation coefficients for water, all phytoplankton, $CaCO_3$, and ice, $\tilde{z}$ is the effective vertical coordinate, $a_i$ is the fractional sea ice cover, and $h_i$ and $h_s$ are calculated sea ice and snow cover thickness. Opal generated by diatoms is not explicitly traced and is therefore not included in the underwater light field.

The actual growth rate ($J_X$) of phytoplankton is taken to be the minimum of the growth functions described above:

$$J_{LP} = \min(J_I, J_{\max} u_{NO_3^-}, J_{\max} u_{PO_4^{3-}}), \tag{13}$$

$$J_{CP} = \min(J_I, J_{\max} u_{NO_3^-}, J_{\max} u_{PO_4^{3-}}), \tag{14}$$

$$J_{DT} = \min(J_I, J_{\max} u_{NO_3^-}, J_{\max} u_{PO_4^{3-}}, J_{\max} u_{Si}), \tag{15}$$

$$J_{DZ} = \min(J_I, J_{\max} u_{PO_4^{3-}}). \tag{16}$$

In Equation 2, two loss terms other than predation are considered. Non-grazing mortality is parameterised using a linear mortality rate ($m$). Temperature-dependent fast remineralisation is a loss term used to implicitly account for the microbial loop and dissolved organic matter cycling, and is parameterised using a temperature dependency multiplied by a constant ($\mu_0^*$) (Schmittner et al., 2008):

$$\mu^* = \mu_0^* \exp^{\frac{T}{T_b}}. \tag{17}$$

With this formulation, increasing seawater temperature increases the return of nutrients to the upper ocean.

### 2.2.3 Zooplankton

Changes in zooplankton population ($Z$) are calculated as the total ingested food (phytoplankton, zooplankton, and organic detritus) scaled with a growth efficiency coefficient ($\varpi$) minus mortality. In addition to mortality from higher trophic level predation calculated with a quadratic mortality function ($m_Z Z^2$), intra-guild predation is represented with a separate term ($G_Z$) (Keller et al., 2012; Kvale et al., 2015b; Nickelsen et al., 2015).

$$S(Z) = \varpi (G_{LP} + G_{CP} + G_{DZ} + G_{DT} + G_{Detr_{tot}} + G_Z) - m_Z Z^2 - G_Z. \tag{18}$$

Zooplankton grazing ($G$) follows Keller et al. (2012). Relevant parameters are listed in Table 4. Grazing of each food source is calculated using a Holling II function, where a calculated maximum zooplankton grazing rate ($\mu_Z^{\max}$) is reduced by a scaling

that is weighted by a relative food preference ($\psi_X$, where "$X$" stands for any of the food sources and the sum of all preferences must equal 1), the total prey population and a half saturation constant for zooplankton ingestion ($k_z$):

$$G_X = \mu_Z^{\max} Z \frac{\psi_X X}{\psi_{LP} LP + \psi_{CP} CP + \psi_{DZ} DZ + \psi_{DT} DT + \psi_{Det} Detr_{\text{tot}} + \psi_Z Z + k_z}. \tag{19}$$

The calculated maximum potential grazing rate is a function of a maximum potential grazing rate at $0\,^\circ$C ($\mu_Z^\theta$), temperature, and oxygen, where grazing activity is capped when temperatures exceed $20\,^\circ$C (Keller et al., 2012):

$$\mu_Z^{\max} = \mu_Z^\theta \max(0, r_{sox}^{\text{O}_2} b^{c \cdot \min(20^\circ\text{C}, T)}). \tag{20}$$

Grazing is also reduced under suboxic conditions ($r_{sox}^{\text{O}_2}$):

$$r_{sox}^{\text{O}_2} = 0.5 \left(\tanh([\text{O}_2] - 8\,\text{mmol m}^{-3}) + 1\right) \tag{21}$$

where $\text{O}_2$ is dissolved oxygen in mmol m$^{-3}$.

### 2.2.4 Organic Detritus

As was introduced in Kvale et al. (2015b), organic carbon detritus sources and sinks are split into "free" ($Detr_{\text{free}}$) and "ballast" ($Detr_{\text{bal}}$) pools using a fixed ratio (R$_{\text{bal:tot}}$). Ballast detritus is formed of the CaCO$_3$-protected portion of calcifying phytoplankton ($CP$), and zooplankton. This protected portion does not interact with nutrient pools directly, and instead transfers from the "ballast" to the "free" detrital pool at the rate of CaCO$_3$ dissolution ($\lambda_{\text{CaCO}_3}$):

$$Detr_{\text{tot}} = Detr_{\text{bal}} + Detr_{\text{free}} \tag{22}$$

$$
\begin{aligned}
S(Detr_{\text{bal}}) = {} & (1 - \gamma)\left(G_{Detr_{\text{bal}}} + (G_Z + G_{CP})\text{R}_{\text{bal:tot}}\right) + (m_Z Z + m_{CP} CP)\text{R}_{\text{bal:tot}} - G_{Detr_{\text{bal}}} \\
& - \frac{\text{R}_{\text{bal:tot}} \lambda_{\text{CaCO}_3} [\text{CaCO}_3]}{\text{R}_{\text{CaCO}_3:\text{POC}} \text{R}_{\text{C:N}}} - w_C \frac{\partial Detr_{\text{bal}}}{\partial z}
\end{aligned} \tag{23}
$$

where $\gamma$ is the food assimilation efficiency, R$_{\text{CaCO}_3:\text{POC}}$ is a fixed production ratio of CaCO$_3$ to organic detritus, R$_{\text{C:N}}$ is a Redfield molar ratio, and $w_C$ is the sinking speed of calcite. Free detritus is described as:

$$
\begin{aligned}
S(Detr_{\text{free}}) = {} & (1 - \gamma)\left(G_Z + G_{DZ} + G_{DT} + G_{Detr_{\text{free}}} + G_Z(1 - \text{R}_{\text{bal:tot}}) + G_{CP}(1 - \text{R}_{\text{bal:tot}})\right) \\
& + m_Z Z^2(1 - \text{R}_{\text{bal:tot}}) + m_{DZ} DZ + m_{DT} DT + m_{LP} LP \\
& + m_{CP} CP(1 - \text{R}_{\text{bal:tot}}) - \mu_D Detr_{\text{free}} - G_{Detr_{\text{free}}} \\
& + \frac{\text{R}_{\text{bal:tot}} \lambda_{\text{CaCO}_3} [\text{CaCO}_3]}{\text{R}_{\text{CaCO}_3:\text{POC}} \text{R}_{\text{C:N}}} - w_D \frac{\partial Detr_{\text{free}}}{\partial z}
\end{aligned} \tag{24}
$$

where $\mu_D$ is the detrital remineralisation rate and $w_D$ is the sinking speed of organic detritus.

### 2.2.5 Detrital Iron

Iron in detritus follows Nickelsen et al. (2015), but with the additional contribution of calcifiers and diatoms:

$$
\begin{aligned}
S(Detr_{\text{Fe}}) =\ & R_{\text{Fe:N}} \left( (1-\gamma)(G_{LP} + G_{DZ} + G_{DT} + G_{Detr_{\text{tot}}} + G_Z + G_{CP}) \right. \\
& \left. + m_{LP}LP + m_{DZ}DZ + m_{DT}DT + m_Z Z^2 + m_{CP}CP - G_{Detr_{\text{tot}}} \right) \\
& + [\text{Fe}]_{\text{orgads}} + [\text{Fe}]_{\text{orgads}_{ca}} + [\text{Fe}]_{\text{col}} - \mu_D Detr_{\text{Fe}} - w_D \frac{\partial Detr_{\text{Fe}}}{\partial z}.
\end{aligned}
\tag{25}
$$

$R_{\text{Fe:N}}$ is a fixed iron to nitrogen ratio that converts total organic nitrogen detritus sources into iron units. The remineralisation and sinking of detrital iron occurs at the same rates as free organic detritus. Two scavenging processes are considered: adsorption of dissolved iron onto organic detritus ($[\text{Fe}]_{\text{orgads}}$):

$$
[\text{Fe}]_{\text{orgads}} = k\text{Fe}_{\text{org}} [\text{Fe}'] \left( Detr_{\text{free}} R_{\text{C:N}} M_C \right)^{0.58}
\tag{26}
$$

and particulate $CaCO_3$ ($[\text{Fe}]_{\text{orgads}_{ca}}$):

$$
[\text{Fe}]_{\text{orgads}_{ca}} = k\text{Fe}_{\text{ca}} [\text{Fe}'] \left( [CaCO_3] M_{CaCO_3} \right)^{0.58}
\tag{27}
$$

calculated as a function of free organic detritus and carbonate particles converted to carbon units, respectively, a scavenging rate constant ($k\text{Fe}_{\text{org}}$ and $k\text{Fe}_{\text{ca}}$), iron available for scavenging ($[\text{Fe}']$), and inorganic precipitation of dissolved iron into colloidal material ($[\text{Fe}]_{\text{col}}$), which is calculated as a linear function independent of organic particle concentration and unchanged from Nickelsen et al. (2015). While calcite is known to be a powerful scavenger of iron and other trace metals (Olsson et al., 2014), the strength of plankton-derived calcite in scavenging dissolved iron is unquantified. A test of this sensitivity is planned at a later stage, so for now the calcite scavenging parameter value is set equal to that of organic detritus.

### 2.2.6 Calcite

As in Kvale et al. (2015b), the source and sink terms for $CaCO_3$ include both phytoplankton calcifier and zooplankton sources from grazing and mortality, and losses from dissolution and sinking:

$$
S([CaCO_3]) = ((1-\gamma)(G_{CP} + G_Z) + m_{CP}CP + m_Z Z)\, R_{CaCO_3\text{:POC}}\, R_{\text{C:N}} - \lambda_{CaCO_3}[CaCO_3] - w_C \frac{\partial [CaCO_3]}{\partial z}
\tag{28}
$$

where $w_C$ is the sinking speed of calcite, and the dissolution specific rate ($\lambda_{CaCO_3}$) is calculated using a parameterisation from Aumont et al. (2003); Kvale et al. (2015b).

Calcite associated with living plankton is calculated separately (but not traced) as:

$$
S[CaCO_3]_{\text{liv}} = (S(CP) + S(Z))\, R_{CaCO_3\text{:POC}}\, R_{\text{C:N}}.
\tag{29}
$$

### 2.2.7 Opal

The production of biogenic opal is calculated as a function of the diatom grazing and linear mortality loss terms:

$$
Pr(\text{Opal}) = ((1-\gamma)\, G_{DT} + m_{DT}DT)\, R_{\text{Opal:POC}}\, R_{\text{C:N}}
\tag{30}
$$

where $R_{Opal:POC}$ is a production ratio that varies as:

$$R_{Opal:POC} = R_{Opal:POC,0} \, (\min(\frac{[Si]}{k_{Si}}, 1) \, (4 - 3 \min(\frac{[Fe]}{k_{FeDT}}, 1))).$$ (31)

This parameterisation was introduced by Aumont et al. (2003) and yields an average surface opal:free detritus export value of
around 1 Si:C across the Southern Ocean, using a fixed average ratio ($R_{Opal:POC,0}$) of 0.5. Production of lithogenic opal occurs
mostly on land (Tréguer and De La Rocha, 2013), so its contribution to marine silicate cycling is included simplistically via
the dissolved silicate river flux calculation. In linking opal production to diatom mortality, our model essentially focuses on
the importance of its export pathway in establishing vertical gradients. Opal that is not exported is effectively assumed to be
recycled to dissolved silicate within the surface layer.

Dissolution of opal in the water column is calculated by assuming instantaneous sinking of the vertically integrated produc-
tion, where the local flux of opal ($F_{Opal}$ in mol Si m$^{-2}$) is distributed down the water column using the e-folding temperature
parameterisation (unitless), scaled by a dissolution rate constant ($\lambda_{Opal}$, in day$^{-1}$), which is divided by a sinking rate ($w_{Opal}$,
in meters day$^{-1}$):

$$Di(Opal) = F_{Opal} \frac{d}{dz} = F_{Opal} \, (\frac{\lambda_{Opal}}{w_{Opal}} \, \exp^{\frac{T}{T_b}}).$$ (32)

Dissolution is presented in units of mol Si m$^{-3}$ day$^{-1}$. This parameterisation results in greater dissolution at warm temper-
atures and is similar to the instant-sinking-and-dissolution function applied to model calcite (Schmittner et al., 2008). Opal
dissolution in the water column is primarily controlled by temperature and silicate saturation, with higher temperatures and
lower silicate saturations leading to faster dissolution (Sarmiento and Gruber, 2006; Ridgwell et al., 2002). Secondary drivers
include the aluminium content and surface area of diatom shells (Van Cappellen et al., 2002), and the presence or absence of
organic coatings, where the loss of the coating increases opal dissolution rates (Bidle and Azam, 1999). We do not consider
silicate or aluminium concentration dependencies explicitly in this model because we do not explicitly calculate thermody-
namic dissolution of opal (nor do we simulate aluminium), but the temperature dependency exerts an influence on dissolution
of the same sign (faster dissolution in warmer, i.e. shallower and undersaturated, water). We find this parameterisation of-
fers improved model fit to World Ocean Atlas silica distributions relative to other parameterisations that we tested, e.g. the
temperature-dependent parameterisation of Gnanadesikan (1999) or the temperature and oxygen-dependent parameterisation
of Enright et al. (2014). The Gnanadesikan (1999) parameteristion yields lower dissolution rates at low temperatures than the
Enright et al. (2014) parameterisation, which is similarly formulated but which includes an additional oxygen scaling. The
Enright et al. (2014) oxygen scaling is not justified in their model description, but it has the effect of increasing Si dissolution
rates in the deep ocean (exacerbating the overestimation of Si dissolution in this region by the Gnanadesikan (1999) scaling
described in Ridgwell et al. (2002)) and decreasing Si dissolution rates (to a lesser extent) in the near-surface. Our temperature
scaling has the effect of raising dissolution rates at the surface. Greater dissolution rates at the surface may be necessary to
compensate for the low vertical resolution of the model. Global average implicit opal export, dissolution, and seafloor loss rate
profiles are plotted in Fig. 2 for model year 2014. Consistent with the estimates of Tréguer et al. (2021), opal cycling in the
open ocean largely occurs in the shallower depths, with opal export following a power law decrease with increasing depth. The

 global ocean average opal dissolution rate at 1000 m depth is only half that at the ocean surface, and the average flux of opal at the seafloor at 500 m depth is only half that at the surface. Shallow shelf areas are hence an important and relatively fast region of turnover for our simulated silicate cycle.

### 2.2.8 Particle Sinking

Detritus (Schmittner et al., 2005), calcite (Kvale et al., 2015b), and iron (Nickelsen et al., 2015) particles are exported from the surface with a sinking speed ($w$) that increases linearly ($wdc$, $wdd$) with depth ($z$; Berelson 2001) for calcite and ballasted organic detritus:

$$w_C = w_{C,0} + wdc \times z \tag{33}$$

and free detritus and associated iron:

$$w_D = w_{D,0} + wdd \times z. \tag{34}$$

Alternative parameterisations exist and their effects on fluxes and model performance make for interesting comparisons (e.g., Kriest and Oschlies, 2011; Cael and Bisson, 2018), but we do not explore them here. The initial surface sinking speeds of particles are assigned different values to represent the denser structure of $CaCO_3$ relative to that of organic detritus. Ballasted detritus sinks at the $CaCO_3$ speed, but once it enters the free pool it uses the organic detrital sinking speed and remineralisation rate. Any organic detritus reaching the sediments is dissolved back in to the water column to ensure conservation of carbon and phosphate in the model domain. Calcite particles that reach the seafloor enter the sediment model (Kvale et al., 2015b) if it is active, though we do not use it here (in this case the particles dissolve instantly at the seafloor). Iron detritus reaching the sediments is lost from the ocean, unless bottom water oxygen falls below $5 \ \mathrm{mmol \ m^{-3}}$, whereupon detrital iron reaching the bottom is dissolved back into the water column (Nickelsen et al., 2015). By definition, iron is not mass-conserving in the model, but is formulated with open boundary conditions (atmosphere and sediments) - hence it is not just the oxygen threshold that can cause a loss or gain of marine iron. In the model the oxygen threshold is only met along coastal boundaries in the north Pacific under modern conditions. A sedimentary release of dissolved iron ($[\mathrm{Fe}]_{\mathrm{sed}}$) is prescribed as in Nickelsen et al. (2015):

$$[\mathrm{Fe}]_{\mathrm{sed}} = R_{\mathrm{Fe:P_{sed}}} \ F_{\mathrm{POP}} \ \exp^{\frac{T}{T_b}} \tag{35}$$

where $R_{\mathrm{Fe:P_{sed}}}$ is a ratio of iron released from sediment in proportion to the flux of phosphorus in the organic detritus ($F_{\mathrm{POP}}$, which includes free and ballast detritus) reaching the bottom.

Opal reaching the seafloor is either returned to the silicic acid tracer or considered lost to the sediments ($[\mathrm{Si}]_{\mathrm{sed}}$) according to criteria laid out in Sarmiento and Gruber (2006):

$$\text{For F(Opal)} > 2 \ \mathrm{mmol \ m^{-2}d^{-1}} \ \ [\mathrm{Si}]_{\mathrm{sed}} = 0.3 \, \mathrm{F(Opal)}$$
$$\text{For F(Opal)} \leq 2 \ \mathrm{mmol \ m^{-2}d^{-1}} \ \ [\mathrm{Si}]_{\mathrm{sed}} = 0.05 \, \mathrm{F(Opal)} \tag{36}$$

Opal flux, F(Opal), is calculated as depth-integrated opal production (Pr(Opal)) less the depth-integrated instantaneous dissolution term (Di(Opal)). The model conserves silica when river fluxes are allowed to compensate for the net change due to the ocean sedimentary sink, hydrothermal input, and atmospheric deposition.

The sub-grid bathymetric scaling that was introduced for the iron model in Nickelsen et al. (2015) is extended here to apply to all particle fluxes reaching the bottom ocean grid. This scaling feature calculates a sedimentary exchange factor based on the proportion of each grid cell that falls outside of the model grid's depth according to a high-resolution bathymetric dataset. This scaling is used to account for high-relief bathymetric features, such as ridges and troughs, in the sediment transfer functions. We do not use the sediment model in this manuscript.

### 2.2.9 Dissolved Inorganic Tracers

Ocean nutrient sources and sinks (in concentration/time units) follow:

$$
\begin{aligned}
S([\mathrm{PO_4^{3-}}]) =\ & (\mu_D Detr_{\mathrm{free}} + \mu_{LP}^* LP + \mu_{CP}^* CP + \mu_{DT}^* DT + (\gamma - \varpi)(G_{LP} + G_{CP} + G_{DT} + G_{DZ} \\
& + G_{Detr_{\mathrm{free}}} + G_Z) - J_{LP} LP - J_{CP} CP - J_{DT} DT - J_{DZ} DZ)\, \mathrm{R_{P:N}}
\end{aligned}
\tag{37}
$$

$$
\begin{aligned}
S([\mathrm{NO_3^-}]) =\ & (\mu_D Detr_{\mathrm{free}} + \mu_{LP}^* LP + \mu_{CP}^* CP + \mu_{DT}^* DT + (\gamma - \varpi)(G_{LP} + G_{CP} + G_{DT} + G_{DZ} \\
& + G_{Detr_{\mathrm{free}}} + G_Z) - J_{LP} LP - J_{CP} CP - J_{DT} DT - u_{\mathrm{NO_3}} J_{DZ} DZ)(1 - 0.8\, \mathrm{R_{O:N}}\, r_{sox}^{\mathrm{NO3}})
\end{aligned}
$$

where $\mathrm{R_{P:N}}$ and $\mathrm{R_{O:N}}$ are Redfield molar ratios and $u_{\mathrm{NO_3}}$ is nitrate availability (diazotrophs use nitrate when available). In suboxic water (less than $5\,\mathrm{mmol\,m^{-3}}$), oxygen consumption is replaced by denitrification ($r_{sox}^{\mathrm{NO_3^-}}$):

$$
r_{sox}^{\mathrm{NO_3^-}} = \max(0, 0.5(1 - \tanh([\mathrm{O_2}] - 5\,\mathrm{mmol\,m^{-3}}))).
\tag{38}
$$

There are no additions of phosphate, nitrate, or oxygen along the boundary (with the exception of air/sea gas exchange in the case of oxygen; Keller et al., 2012).

Dissolved iron includes sources and sinks of particulate iron mentioned above, as well as prescribed dust deposition ($[\mathrm{Fe}]_{\mathrm{dust}}$) as in Nickelsen et al. (2015) and hydrothermal iron ($[\mathrm{Fe}]_{\mathrm{hydr}}$) (Muglia et al., 2017) boundary terms (also in concentration/time units):

$$
\begin{aligned}
S([\mathrm{Fe}]) =\ & \mathrm{R_{Fe:N}}\, (\mu_D Detr_{\mathrm{tot}} + \mu_{LP}^* LP + \mu_{CP}^* CP + \mu_{DT}^* DT + (\gamma - \varpi)(G_{LP} + G_{CP} + G_{DT} + G_{DZ} \\
& + G_{Detr_{\mathrm{tot}}} + G_Z) - J_{LP} LP - J_{CP} CP - J_{DT} DT - J_{DZ} DZ) \\
& - [\mathrm{Fe}]_{\mathrm{orgads}} - [\mathrm{Fe}]_{\mathrm{orgads}_{ca}} - [\mathrm{Fe}]_{\mathrm{col}} + \mu_D Detr_{\mathrm{Fe}}
\end{aligned}
\tag{39}
$$

$$
B([\mathrm{Fe}]) = [\mathrm{Fe}]_{\mathrm{sed}} + [\mathrm{Fe}]_{\mathrm{dust}} + [\mathrm{Fe}]_{\mathrm{hydr}}
\tag{40}
$$

The Nickelsen et al. (2015) model applied the iron dust flux to the surface ocean after the biological routine and before the mixing routine. This resulted in low model sensitivity to iron dust inputs, because the dust was mixed away faster than the

biological processes had a chance to access it. In this version, the iron dust flux is added prior to the biological routine (which operates on a shorter timestep than ocean mixing), and results in a greater biological sensitivity to iron dust flux.

DIC and alkalinity tracer sources and sinks are a function of sources and sinks of prognostic $CaCO_3$ (Kvale et al., 2015b).

$$
\begin{aligned}
S([\text{DIC}]) \;=\;& S([\text{PO}_4^{3-}]) \, \text{R}_{\text{C:P}} + \lambda_{\text{CaCO}_3}[\text{CaCO}_3] - S[\text{CaCO}_3]_{\text{liv}} \\
& -((1-\gamma)(G_Z + G_{CP}) + m_Z Z + m_{CP} CP) \, \text{R}_{\text{CaCO}_3:\text{POC}} \, \text{R}_{\text{C:N}}
\end{aligned}
\tag{41}
$$

$$
\begin{aligned}
S([\text{Alk}]) \;=\;& -S([\text{PO}_4^{3-}]) \, \text{R}_{\text{N:P}} + 2\,(\lambda_{\text{CaCO}_3}[\text{CaCO}_3] - S[\text{CaCO}_3]_{\text{liv}}) \\
& -2\,((1-\gamma)(G_Z + G_{CP}) + m_Z Z + m_{CP} CP) \, \text{R}_{\text{CaCO}_3:\text{POC}} \, \text{R}_{\text{C:N}}
\end{aligned}
\tag{42}
$$

With the exception of air/sea gas exchange of $CO_2$ there are no boundary additions of DIC or alkalinity when the sediment model is deactivated (as is presented here).

Dissolved silicic acid tracer sources, sinks, and boundary terms (all in concentration/time units) follow:

$$
S([\text{Si}]) = \text{Di}(\text{Opal}) - \text{Pr}(\text{Opal})
\tag{43}
$$

$$
B([\text{Si}]) = [\text{Si}]_{\text{riv}} + [\text{Si}]_{\text{dust}} - [\text{Si}]_{\text{sed}} + [\text{Si}]_{\text{hydr}}
\tag{44}
$$

where discharge from rivers ($[\text{Si}]_{\text{riv}}$) is used as a budget balancing term to compensate for any remainder in the other external sources and sinks: windborne dust ($[\text{Si}]_{\text{dust}}$), hydrothermal silicate ($[\text{Si}]_{\text{hydr}}$), and loss to the sediments ($[\text{Si}]_{\text{sed}}$). River inputs of silicate, alkalinity, and DIC are scaled against seasonally-variable river flow using the standard UVic ESCM version 2.9 O_rivflux.F forcing file. Dust deposition from the atmosphere is prescribed using an interpolated monthly pre-industrial dust flux derived from the NCAR's Community Climate System Model (Mahowald et al., 2006). The silica content of the dust is derived from maps (Zhang et al., 2015), with a global average dust solubility of 3% assumed to produce annual bioavailable fluxes in agreement with estimates (Sarmiento and Gruber, 2006; Tréguer and De La Rocha, 2013). Observations demonstrate wide spatial variability in the solubility parameter (Tréguer and De La Rocha, 2013) so our single value represents a simplification. Silicate from hydrothermal sources are prescribed using a static mask scaled from the hydrothermal iron mask (Muglia et al., 2017) using a Fe:Si ratio to obtain the estimated annual total contribution from hydrothermal sources in Sarmiento and Gruber (2006) (0.6 Tmol Si y$^{-1}$). This estimate has recently been revised upward to $1.7\pm0.8$ Tmol Si y$^{-1}$ (Tréguer et al., 2021), thus our hydrothermal contribution is under-estimated. Model sensitivity to both sources will be explored in future tests.

## 3 Model Assessment in a Modern Climate

The model is spun up to equilibrium (greater than 15,000 years) at year 1765 boundary conditions, then forced with historical data for comparison to observational datasets. Forcing includes historical atmospheric $CO_2$ concentrations, agricultural land

cover, volcanic radiative forcing, sulphate aerosol and CFC concentrations, changes in land ice and solar forcing (Machida et al., 1995; Battle et al., 1996; Etheridge et al., 1996, 1998; Flückiger et al., 1999, 2004; Ferretti et al., 2005; Meure et al., 2006). Solar insolation at the top of the atmosphere, wind stress, and wind fields vary seasonally (Kalnay et al., 1996), and

wind fields are geostrophically adjusted to air temperature anomalies (Weaver et al., 2001).

Table 6 lists key biogeochemical properties diagnosed by the model for a modern climate, as well as corresponding properties diagnosed from KMBM1 (Keller et al., 2012). We compare model output at year 2004, although data sources reflect a range of collection and publication dates. Net primary production (NPP) is similar to previous model versions (53.60 Pg C $y^{-1}$, e.g. compared to 54.33 Pg C $y^{-1}$ in Keller et al. 2012), and still within the literature range of 44–78 Pg C $y^{-1}$ (e.g., Carr

et al., 2006; Jin et al., 2006). Calcite production is similar to and nitrogen fixation rates are somewhat improved with respect to earlier model versions, though remain too low, as do deep particle fluxes of particulate organic carbon (POC) and shallow and deep calcite (also referred to as PIC, for particulate inorganic carbon). Diagnosed surface opal production is also too low, and below the range of a recent estimate (Tréguer et al., 2021). Accordingly, deep ocean (2 km depth) opal fluxes are also too low (only about 50% of the observationally-based deep ocean estimates of Tréguer and De La Rocha (2013)). However,

upper ocean (130 m) opal flux is about 9% larger than the estimate provided by Tréguer et al., 2021. This may be due to the model underestimating upper-ocean dissolution (10.3 Tmol Si $y^{-1}$ in our model compared to 124 Tmol Si $y^{-1}$ estimated in the surface mixed layer by Tréguer et al., 2021). There is too little dissolution of opal throughout the water column (74% of the observational estimate in Tréguer et al., 2021, but the ratio of total water column dissolution to biogenic production is too high, with a ratio of 0.95 compared to 0.67 calculated by Tréguer et al., 2021). The calculated river flux is 2.38 Tmol

Si $y^{-1}$; lower than the Tréguer et al. (2021) estimate of 8.1 Tmol Si $y^{-1}$. Overall, an increase in production and subsequent processes is required to improve the model agreement with independently estimated fluxes. Low bias in fluxes is also found in calcite production and nitrogen fixation, as well as in deep particle fluxes, which is due at least in part to the model bias of a too-sluggish deep circulation (Kriest et al., 2020). The calculated flux of silicate through the seafloor is only 70% of that estimated by Tréguer et al. (2021). An underestimation of seafloor flux is potentially corrected with an increase in pelagic opal

production rates or an decrease in benthic return. Benthic fluxes are potentially improved with newer models of benthic transfer (e.g., Dale et al., 2021), and will be explored in the future. A recent review suggests the global silicate cycle sources and sinks are roughly balanced, but uncertainties are still substantial (Tréguer et al., 2021). It should be noted that the global silica cycle may be unbalanced between sources and sinks at present and observations of global input and output rates are both affected by anthropogenic perturbation and have large uncertainties (Tréguer and De La Rocha, 2013). Atmospheric and hydrothermal

silica inputs are fixed in these simulations, but each source and sink of silicate produces a unique spatial distribution in the water column. These terms are slated for automated calibration in the future (e.g., Yao et al., 2019; Kriest, 2017; Kriest et al., 2017), thus, our hand-tuned simulations are meant to serve as a baseline for future improvements to the model. Also note, that despite the deficiencies described above, the KMBM3 demonstrates reasonable performance against a suite of state-of-the-art earth system models (described below).

Total global phytoplankton biomass is on the low end, but within the range of, previous estimates (0.55 Pg C, virtually unchanged from Keller et al. 2012, compared to 0.5-2.4 Pg C from Buitenhuis et al. 2013; Table 6). KMBM3 explicitly

represents only a fraction of the ecological complexity found in the real ocean, which causes our biomass estimates for the phytoplankton functional types to look dissimilar to the Buitenhuis et al. (2013) biomass estimates (e.g., with low latitude mixed phytoplankton (a model-specific category, LP) having a biomass only 11% of the lowest picophytoplankton estimate

and both diazotrophs (DZ) and calcifiers (CP) having more biomass than the upper observational estimate). These functional types proved particularly difficult to tune by hand, with small variations in parameter values causing extinction. An over-estimate of calcifiers is compensated by the low PIC:POC production ratio (0.07), which is meant to represent 7% of the phytoplankton class having a PIC:POC production ratio of 1 (compare to a ratio for *Emiliana huxleyi* of 0.51–2.30 from Paasche 2001). Overestimated diazotroph biomass results in an increase (0.04 Pg N $y^{-1}$) in nitrogen fixation compared to earlier

model versions, which is improved with respect to, but still lower than the observational independent estimate. In KMBM3, diazotrophs use preformed nitrate when available. Thus in our modelling context, this functional type can be considered "slow-growing phytoplankton capable of fixing nitrogen when necessary". Constraints on this functional type will be explored in the future. Diatom biomass estimates are too low and outside the Buitenhuis et al. (2013) range, although the functional type is represented in the same relative proportion to low latitude phytoplankton (LP) as is reported for picophytoplankton-to-diatoms

by Buitenhuis et al. (2013).

     Looking next at spatial distributions of biological rates, Figure 3 compares KMBM3 NPP at year 2014 to the Westberry et al. (2008) model applied to MODIS (NASA, 2018) climatology from 2012-2018. As is repeatedly found in the KMBM (see plot of the Keller et al. 2012 model NPP), open-ocean primary production rates are generally too high, particularly in the eastern equatorial Pacific and northern Indian Ocean. These very high-production zones compensate in the globally integrated

rate estimate for the diffuse and more widespread production calculated from satellites. The spatial pattern in NPP has changed somewhat from earlier model versions, with a continued under-estimate of NPP occurring within gyres and a new over-estimate of NPP in the 40-60°N and S ranges.

     Spatial biases in NPP are also held in surface calcite concentrations. Figure 4 compares KMBM3 $CaCO_3$ averaged between 2004-2014 with the 2002-2018 climatology data from MODIS (NASA, 2018). Model calcite is too high in the North Pacific

and Southern Ocean between 40° and 60°S, and too low south of 60°S. Upwelling zones and the Indian Ocean have an over-estimated role in calcite production, and coastal calcite production is either not resolved, or under-estimated.

     Phytoplankton biomass is compared to the MAREDAT datasets (Leblanc et al., 2012; Luo et al., 2012; O'Brien, 2012) in Figure 5. Globally integrated diatom biomass is too low compared to observational estimates but the diatom geographical distributions agree roughly with the very sparse Leblanc et al. (2012) dataset (highest concentrations in the Southern Ocean, North

Pacific, and North Atlantic). Like the globally integrated biomass estimate, calcifiers (CP) are universally over-estimated compared to O'Brien (2012). As explained above, this functional type is meant to represent a variety of moderate growth, moderate nutrient affinity phytoplankton types, including those that calcify. Therefore it is not unexpected to have an over-estimate of the biomass. Diazotroph biomass is primarily concentrated in the tropics, which agrees spatially but not in magnitude with the limited data of Luo et al. (2012). Mixed low-latitude phytoplankton (LP) is a model-specific category with no clear analogue

in the MAREDAT dataset.

We compare CMIP6 model output relative to KMBM3 model output due to the very low normalised correlation (0.04-0.15) and normalised standard deviation (0.10-0.22) of all models against the sparse Leblanc et al. (2012) diatom biomass dataset. At the time of writing, available CMIP6 simulated annual average diatom biomass shows diverse quantities (maximum concentrations from 0.0035 to 0.03 mol C m$^{-3}$), and spatial distributions ranging from global maximum concentrations at the Equator (CanESM5-CanOE, Swart et al. 2019), to shallow seas and coastlines (IPSL-CM6A-LR, Boucher et al. 2018), to the high latitudes (CMCC-ESM2, EC-Earth3-CC, GFDL-ESM4, CESM2; Lovato et al. 2021; EC-Earth Consortium 2021; Krasting et al. 2018; Danabasoglu 2019, respectively). Fig. 6 contextualises these diatom biomass estimates relative to the KMBM3 diatom biomass, simulated at year 2014. The KMBM3 is most closely correlated with GFDL-ESM4. The KMBM3 has a near-perfect standard deviation and second-closest correlation with CMCC-ESM2, a higher-resolution fully-coupled earth system model with full representation of the global carbon cycle and ocean biogeochemistry. The CMCC-ESM4, KMBM3, CESM2, and CanESM5-CanOE models all fall outside the cluster in correlation and standard deviation of EC-Earth3-CC, IPSL-CM6A-LR, and GFDL-ESM4. EC-Earth3-CC and IPSL-CM6A-LR both utilise the PISCES biogeochemical model (and are therefore expected to show similar correlation and standard deviation relative to a reference model), but GFDL-ESM4 utilises COBALT (an independent biogeochemical model lineage). As stated above, phytoplankton biomass is difficult to tune for (particularly when multiple functional types are represented) due to the under-constrained parameter space, the theoretical nature of phytoplankton functional categories, and the sparsity of gridded, annually-averaged biomass datasets. Therefore, a wide range in model biomass estimates is expected across the CMIP6 ensemble.

KMBM3 interior ocean particle flux performance is encouraging. Figure 7 compares model POC, PIC, and opal fluxes at 2 km depth to the Honjo et al. (2008) data compilation. The model shows a general under-estimate with respect to the observations for all deep ocean particle fluxes, particularly for intermediate rates. Root mean square error is improved beyond Kvale et al. (2015a) for PIC (101.4 compared to 147.1 mmol C m$^{-2}$ y$^{-1}$), and POC (96.7 versus 98.0 mmol C m$^{-2}$ y$^{-1}$ in Kvale et al. 2015a). Root mean square error for opal flux at 2 km depth is 223.0 mmol Si m$^{-2}$ y$^{-1}$. However, the model captures large-scale features of high flux rates in the north west Pacific and Southern Ocean, as well as moderate flux rates in the Indian and North Atlantic basins. The KMBM3 demonstrates moderate skill at simulating annual average deep ocean opal fluxes relative to CMIP6 models (Fig. 8), with two of six models having a lower correlation with the Honjo et al. (2008) data compared to KMBM3, and four of six having a lower, or equal, normalised standard deviation. Two of six have an over-estimated bias in opal flux, relative to KMBM3. The GFDL-ESM4 performs best against the Honjo et al. (2008) dataset, of the model simulations examined here.

Particle flux rates, alongside the model's ocean circulation, impact ocean tracer distributions. Figure 9 shows carbon and nutrient profiles, globally-averaged and for each basin, compared to GLODAP (Key et al., 2015; Lauvset et al., 2016) and World Ocean Atlas (WOA) (Garcia et al., 2014a, b) data. Nutrients are generally too high at depth, and oxygen is too low, partly as a result of over-estimated deep ocean POC flux rates in the Southern Ocean. Global root mean squared error is listed in the figure for each model dissolved inorganic tracer. All tracers except DIC, phosphate and oxygen show improvement with respect to the Keller et al. (2012) model version. Alkalinity is primarily affected by PIC flux and attenuation rates, and shows relatively good agreement (global RMSE value of 0.34 $\mu$mol kg$^{-1}$) with observations. Silicic acid has a global root mean

squared error of 0.388 mmol Si m$^{-3}$. Concentrations are too high in the Atlantic by as much as 50 mmol Si m$^{-3}$ and too low in the Indian by as much as 50 mmol Si m$^{-3}$.

Greater basin and surface detail in carbon and nutrient concentrations is displayed in Figures 10-15. Interior ocean alkalinity is too high by as much as 50 $\mu$mol kg$^{-1}$, and surface alkalinity too low by as much as 50 $\mu$mol kg$^{-1}$, in the Atlantic, as it was with previous model versions (Eby et al., 2009; Keller et al., 2012; Kvale et al., 2015a). Alkalinity in the interior and surface Pacific is too high by as much as 20 $\mu$mol kg$^{-1}$ along the Equator, whereas previously it was over-estimated at depth and under-estimated at the surface (Kvale et al., 2015a). The physical circulation has not changed since previously published model versions. Differences might be partly due to shifts in phytoplankon biogeography and calcification rates. Note however, surface alkalinity is also subject to evaporation-precipitation effects that might have also changed since previous model versions. The Indian Ocean interior alkalinity is under-estimated, by as much as 60 $\mu$mol kg$^{-1}$. Deep ocean DIC is not improved with respect to past model versions (Figure 11), though (at least compared to Kvale et al. 2015a) this discrepancy might be partly attributed to the use of a transient year 2014 model output that includes an anthropogenic signal, instead of the pre-industrial spinup. The deep north Pacific shows a low bias of up to 40 $\mu$mol kg$^{-1}$, similar to previous versions. Surface DIC anomalies are also similar to previous model versions, with DIC being too low in the western Pacific (a consequence of too high export production in the model and physical biases), and too high in the surface Southern Ocean. As with earlier model versions, deep ocean phosphate concentrations (Figure 12) are also generally too high, particularly in the Southern Ocean-sourced deep and intermediate water masses in the Indian and Pacific sectors. Nitrate, however, is too low in these water masses (Figure 13), although in basin average the depth profiles of KMBM1 and KMBM3 are nearly identical for the Southern Ocean, Pacific, and Indian basins below 3000 m. Oxygen anomalies mirror nutrient biases (Figure 14), with oxygen being up to 50 mmol m$^{-3}$ too low along the Southern Ocean particle export pathway, and up to 50 mmol m$^{-3}$ too high in the sub-surface tropical ocean. The oxygen bias in the deep water masses might be addressed with better tuning of the biological production, export and flux parameters, but the tropical ocean deficiencies will require improvements to both the biogeochemistry as well as the physics (Oschlies et al., 2017).

Silicic acid distributions are reasonably well captured by the model (Figures 15, 16). The KMBM3 demonstrates the second-highest correlation with gridded observations (Garcia et al., 2014b) of dissolved silicic acid at year 2014 compared to CMIP6 model output (Fig. 16). Two of 7 models, MPI-ESM1.2-LR and CMCC-ESM2, have a greater difference in normalised standard deviation than KMBM3. Both correlation and standard deviation are generally better, and show less inter-model spread, for dissolved silicic acid than for other metrics. KMBM3 simulates the deep Southern Ocean maximum to within 20 mmol Si m$^{-3}$, with a slight high-bias in the deep Atlantic, Pacific, and especially Indian sectors (Figure 15). This spatial weighting to the Indian sector might represent the high-bias in diatom biomass in this basin (also clearly seen in the opal flux plots). North Atlantic silicic acid gradients are well represented, while a low-bias is apparent in the deep North Pacific of around 20 mmol Si m$^{-3}$. A low-bias is also simulated in the surface North Pacific, which possibly suggests deficiencies in the circulation within and between regional marginal seas (Nishioka et al., 2020). The North Pacific appears to perform well with respect to deep opal flux (Figure 7), but small biases in fluxes can compound with time in dissolved nutrient fields. It might also be that sedimentary processes that influence deep water silicic acid concentrations would be important to resolve in this region. Our low-bias result

in the deep North Pacific is interesting because the origins of the silicic acid-rich deep ocean plume in this region are still under debate (Tréguer and De La Rocha, 2013).

Seasonal succession in functional types follows the general progression of zonal maxima in diatoms preceeding calcifiers by a few weeks (albeit, in separate zonal ranges, Figure 17). This succession is due to the higher nutrient requirements, and faster growth rates, of the diatoms, which are able to take advantage of winter mixing early in the growing season. Once surface nutrient concentrations start to decline, the calcifying phytoplankton become relatively more successful. This pattern is most pronounced in the Southern Ocean. In the lower latitudes, diazotroph biomass peaks earlier in the growing season than the low-latitude, non-calcifying phytoplankton (LP). Diazotrophs have a growth handicap with respect to LP, but their ability to fix nitrate gives them an advantage in more stratified summer conditions. This fixed nitrate is then used by the LP in the winter months.

## 4  Model Assessment Under Climate Change

In addition to the historical model forcings described earlier, from year 2005 to 2300 the simulations are forced using increasing $CO_2$ and non-$CO_2$ greenhouse gas concentrations, projected changes to the fraction of the land surface devoted to agricultural uses (calculated to year 2100 by Hurtt et al. (2011), and then held constant after), and changes in the direct effect of sulphate aerosols following "business-as-usual" RCP scenario 8.5 (RCP8.5, Riahi et al., 2007; Meinshausen et al., 2011). The wind fields continue to be geostrophically adjusted to air temperature anomalies.

### 4.1  Historical Changes (1964-2014)

We next explore model trends from the 1960s to the 2010s and compare to available data from this period. Significant changes are simulated to have already occurred in NPP and phytoplankton biomass, with most of the change over this period occurring since the 1980s (Figure 18). Global NPP is simulated to have declined 1.4% between 1964 and 2014, with the strongest declines in the tropics, northern mid-latitudes, and Southern Ocean due to increasing thermal stratification. In KMBM3, that decline is dominated by the simulated loss of diatom biomass (a 24.4% decline) due to their high nutrient requirements, largely in the Southern Ocean (Figure 19). Diazotrophs are simulated to have experienced a 2.0% loss in biomass globally over this period. Calcifying phytoplankton and low latitude non-calcifying phytoplankton are simulated to have experienced a 0.4% and 1.5% loss, respectively, in net biomass. Increases in calcifying phytoplankton are simulated to have occurred in the Arctic and in the Southern Ocean and decreases are simulated to have occurred in the middle latitudes, where LP have increased their biomass. Warming, and a lengthening growing season (but increased stratification; see Arctic temperature trend in Figure 20) in the Arctic benefits calcifiers. Expansion of coccolithophores into the Arctic has been observed over recent decades (Neukermans et al., 2018). In KMBM3 the Southern Ocean is also simulated to have experienced increasingly favourable conditions for calcifiers, with diatom biomass declining as calcifiers increase. Note, however, this model does not resolve ocean acidification effects on calcification.

KMBM3 simulates a slower global decline in productivity and biomass than the scarce, and controversial, satellite and in-situ chlorophyll record reconstructed by Boyce et al. (2010), who calculated a 1% per year decline in chlorophyll. KMBM3 also does not simulate a large decline prior to the 1950s (Figure 21). Wernand et al. (2013) found no historical trend in chlorophyll in their Forel-Ule ocean colour scale proxy from 1899 to 2000. However, both chlorophyll reconstructions (and KMBM3) suggest strong regional variation in the historical trends. KMBM3 also simulates a smaller decline in global NPP than an earlier modelling effort, which obtained a 6.5% decline between 1960-2006 (Laufkötter et al., 2013). Their simulation similarly resulted in large declines in production in the low latitudes, which they attributed to warming and stratification over the historical period. KMBM3 simulates strong warming in the low latitudes (up to 0.5 degree warming in the zonal mean between 1964 and 2014; Figure 20). Zonal mean trends in idealised natural radiocarbon (simulated without bomb or Suess effects, also Figure 20) are also positive in the upper ocean, suggesting enhanced stratification and reduced vertical mixing, particularly in the tropical and subtropical Pacific and Indian Ocean basins. Laufkötter et al. (2013) also found strong changes in phytoplankton biogeography in the Southern Ocean and north Atlantic, which they attributed to increasing zonal wind stress and vertical mixing, and surface freshening inducing stratification, respectively. However, our models do not agree in the response of diatoms and high-latitude calcifiers, for which their model simulates high latitude increases in diatoms. Our results are more similar to those of Rousseaux and Gregg (2015), who combined a model with satellite data to reconstruct ocean surface changes from 1998-2012. They simulated a global decline in diatoms over this period, which they attributed to an increase in nutrient limitation and photosynthetically available radiation, which favours other functional types. However, the Southern Ocean in their model showed no clear trend in phytoplankton community composition between 1998-2012 (Rousseaux and Gregg, 2015). The model we present here is fully competition-driven. Variable particle sinking and remineralisation rates, and explicit $CaCO_3$ ballasting, further differentiate KMBM3. All of these factors may contribute to the different behaviour reported here.

Trends in biomass and NPP affect deep particle fluxes. Low latitude declines in primary production result in less deep ocean particle export in the western Pacific and Indian Ocean basins, while deep POC and PIC export is simulated to have slightly increased in the Indian and Pacific sectors of the Southern Ocean, and in the Arctic. Trends in deep ocean carbon particle export in KMBM3 generally follow trends in calcifier (CP) biomass (Figure 19), as $CaCO_3$ (PIC) ballasting contributes significantly to deep export (Kvale et al., 2015b, a, 2019). Declining POC export in the North Atlantic is also simulated to have occurred, though in this region the trend is driven by an increase in diatoms, which are less efficient exporters of organic carbon. Deep ocean opal export is simulated to have declined almost globally, with the largest loss in the Southern Ocean. Opal dissolution is temperature-dependent, therefore regional warming is almost certainly contributing to the reduction of deep ocean opal flux, but the loss of diatom biomass is the major driver of this trend in the Southern Ocean. Global losses of particle export across 2 km depth between 1964 and 2014 are calculated at 2.8% (POC), 2.2% (PIC), and 6.2% (opal).

Unfortunately there is no comparable historical reconstruction of deep particle fluxes over this time period. Just as with NPP, Laufkötter et al. (2013) simulated a larger decline (8%) in export production (POC) between 1960 and 2006, which they calculated at 100 m depth. They also simulated the largest declines in the Indian Ocean and west-central Pacific basins, driven by declines in NPP, which in turn were driven by declines in nutrient availability owing to increasing stratification. The models

disagree with respect to trends in the north Atlantic, with KMBM3 producing a decline in POC and PIC deep export (due to a decrease in calcifiers), and only a small increase in opal (due to an increase in diatoms). Both Laufkötter et al. (2013) and KMBM3 simulate a historical increase in diatoms in this region; the difference in export trends can be explained by which functional type (calcifiers or diatoms) is more efficient at POC export, with calcifiers being the more efficient carbon exporter in our formulation. Likewise, KMBM3 simulates different POC export trends in the Southern Ocean as Laufkötter et al. (2013), due to differences in model structure. Increasing calcifiers in the Indian sector in KMBM3 increase POC and PIC export there, while in Laufkötter et al. (2013) diatoms regionally increase.

Changes in carbon and nutrient profiles between 1964 and 2014 (Figure 22) reflect a combination of physical-chemical uptake of anthropogenic $CO_2$ and the changes in ocean circulation on tracer accumulation (Figures 20 and 24). Dissolved inorganic carbon (DIC) is simulated to have increased by more than 10 mmol C m$^{-3}$ in zonal mean over this time period in both the Southern Ocean and North Atlantic. Both regions, but particularly the Southern Ocean, are thought to be the primary regions for anthropogenic carbon uptake into the ocean interior (e.g. Khatiwala et al., 2009). The low latitudes are primary regions of carbon storage (Frölicher et al., 2015), and strong (greater than 25 mmol C m$^{-3}$) increases in DIC are also seen here, in all ocean basins, in the zonal mean profiles.

Declines in phosphate and nitrate are simulated in the upper ocean in all basins, with the largest zonal mean declines of up to 0.1 mmol P m$^{-3}$ and 0.5 mmol N m$^{-3}$). Silicic acid increases by up to 20 mmol Si m$^{-3}$ in the central North Pacific surface (Figure 23) due to a regional shift from DT to CP dominance. A recent reconstruction suggests a positive silicate concentration trend in the upper 300 m in this region over the 1970-2017 time period (Stramma et al., 2020), although the estimated magnitude is lower ($0.013 \pm 0.013$ mmol Si m$^{-3}$ y$^{-1}$). Dissolved iron shows an increasing trend in the tropical surface despite using constant atmospheric sources. Surface and subsurface (300 m depth) concentration trends (Figure 23) reveal spatial heterogeneity; with declining surface and increasing sub-surface phosphate occurring in the subarctic North Pacific (a result of increasing particle export), and increasing concentrations along the Humboldt Current (driven by a regional reduction in nitrate, not shown). These results compare favourably with surface declines in phosphate recorded in the North Pacific between 1961 and 2012 (Yasunaka et al., 2016). They also compare favourably with the recent data compilation of Stramma et al. (2020), which shows a decline in surface nutrients, and an increase in subsurface nutrients, in the subarctic Pacific. In KMBM3, this pattern is produced by the replacement of diatoms with calcifiers, who are more efficient exporters of nutrients. The UVic ESCM lacks a fully dynamic atmosphere model, and therefore does not simulate multi-decadal oscillations in climate, which have been implicated in recent Pacific interior nutrient trends (Stramma et al., 2020). This may be why KMBM3 does a poor job reproducing observations of nutrient trends observed in the central Pacific (Stramma et al., 2020). Also, increases in nitrate are found in the observed record, which KMBM3 does not simulate. This may be because KMBM3 does not include anthropogenic sources of nitrate (summarized by Stramma et al., 2020).

Significant increases in North Atlantic deep ocean concentrations (below 3000 m depth) occur in phosphate, nitrate, and silicic acid (Figure 22). A decline in maximum meriodional overturning (MOC) of about 1 Sv is apparent over this time, and the water has warmed more than 0.3 degrees (Figure 24). Warming of the Gulf Stream increases particle remineralisation rates, thereby raising nutrient concentrations along the North Atlantic Deep Water pathway.

Southern Ocean-sourced intermediate and deep water is simulated to have increased nutrient concentrations from 1964 to 2014, with a positive trend in phosphate and nitrate outcropping along the Antarctic ice margin, in qualitative agreement with the limited observations of an increasing trend in the Indian sector from 1965-2008 (Iida et al., 2013). The Southern Ocean is simulated to have experienced sub-surface increases in phosphate in the regions also experiencing increases in silicic acid and iron; this is due to the less efficient export of particles by diatoms (locally increasing over this time), relative to calcifiers.

Taken together, these results suggest declines in NPP and export production upstream of the Southern Ocean have introduced excess nutrients to the basin, raising the nutrient concentrations in Antarctic Bottom and Intermediate water masses despite declining Southern Ocean NPP. These production and export effects have been exacerbated by physical changes in the circulation; Figure 20 shows a decline in ideal age (greater than 5 years in the zonal mean) in Pacific Intermediate water and the Indian Ocean, which leads to less particle remineralisation (and hence, lower nutrient concentrations) there. Increasing water mass age in Southern Ocean-sourced intermediate and deep water masses (upwards of 30 years in the deep Pacific), likewise has the effect of producing more complete particle remineralisation, resulting in higher nutrient concentrations.

Oxygen is simulated to have declined in all ocean basins, with the exception of the sub-surface low latitudes. It has been previously estimated that the global ocean lost more than 2% of its oxygen since the 1960s (Schmidtko et al., 2017). KMBM3 simulates a 0.7% decline in total oxygen content from 1964-2014, which is an underestimate resulting from physical biases in our model (i.e., deficiencies in simulating low latitude ventilation), though biogeochemical deficiencies might also be contributing (Oschlies et al., 2017). In the deep ocean, the ageing of water masses (and associated more complete particle remineralisation) contributes to the simulated decline in oxygen. In the upper ocean, warming has reduced oxygen solubility, lowering near-surface concentrations.

## 4.2    Long-term Future Changes (2014-2294)

More significant changes in ocean biogeochemistry are still to come, if applied boundary forcing assumptions hold over the next centuries. Figures 25 to 28 extend the previous analysis to year 2294, with respect to year 2014 biogeochemistry. Critically, spatial patterns in NPP trends reverse, with strong increases, in places exceeding 80 gC m$^{-2}$ y$^{-1}$ in the zonal mean, in the Southern Ocean (Figure 25). This trend is dominated by the increase in calcifiers (CP), and to a lesser geographical extent, diatoms (DT) (Figure 19). Calcifiers also continue their historical expansion into the Arctic (again, noting adverse physiological effects of ocean acidification are not simulated in this model), while diazotrophs are simulated to significantly increase in the middle latitudes after year 2100. Low latitude phytoplankton (LP) also broadly increase in biomass between 20°S and 60°N, though the net NPP trend in the low latitudes remains negative.

This new ecosystem model responds differently to forcing than previous versions. Kvale et al. (2015a) compared the responses of the KMBM1 biogeochemistry (no calcifiers) to two versions of calcifier model, also integrated to 2300 using RCP8.5 forcing, and found the application of calcifiers eliminated the global reduction of NPP found in the Keller et al. (2012) version until around the year 2100. However, the introduction of diatoms and iron, and reorganisation of the phytoplankton community structure, produces an even larger decline in global NPP to 2100 (close to 5 Pg C y$^{-1}$) in KMBM3 than found in KMBM1 (less than 1 Pg C y$^{-1}$). The difference appears to be in the low-latitude response, where calcifiers with low nutrient

requirements in the previous version maintained NPP despite stratification, partly by supporting diazotroph nitrogen fixation through efficient ballast removal of surface nitrate (Kvale et al., 2019). In the current model, calcifiers have a lower biomass in the low latitudes, and do not establish this symbiotic relationship with diazotrophs to the same extent. As a consequence, diazotrophs decline more strongly than previously. After 2100, NPP increases abruptly (Figure 21). Rising NPP over the long-term is a long-standing feature of the KMBM biogeochemical model formulation, and occurs because of the acceleration of nutrient recycling by the temperature-sensitive microbial loop in the low and middle latitudes (Schmittner et al., 2008). In this latest model version, this increase in NPP is much smaller (less than 4 Pg C y$^{-1}$ by 2300, compared to 11-13 Pg C y$^{-1}$ in previous model versions). Again, this reduced sensitivity in the low latitudes is due to the reduction of low-latitude ballast-forming calcifiers in the new version. The response of calcifiers outside of the low latitudes to ocean changes is also changed from previous versions, in that our model now simulates increases in biomass in the Southern Ocean as well as the Arctic. Whether these differences are competition effects, or due to other model changes, such as the correction of light attenuation, is difficult to assess, but it confirms the finding of Fu et al. (2016) of a strong dependence on phytoplankton community structure in model response to climate change.

Trends in phytoplankton biomass and productivity can be explained by the physical changes in the model over this time (Figures 24 and 26). Maximum meridional overturning circulation (MOC) declines from 18 to 11 Sv over the period 1800 to 2200, before starting to increase after 2200. Slowing overturning helps to accelerate surface warming, and zonal mean temperatures in the upper north Atlantic rise more than 5 degrees by 2294. Globally increased radiocarbon in the upper ocean suggests widespread increased stratification, as well as a more complete separation between upper and lower water masses globally (as previously reported by Kvale et al., 2018, with only slightly different forcing conditions). Ideal age trends similarly show the lengthening Southern Ocean-sourced deep water pathway that extends to ventilate all ocean basins from the south, replacing the shoaled northern ventilation pathways (also described in detail in Kvale et al., 2018). Ideal ages increase more than 300 years in the deep southern Pacific and Atlantic basins, and in the deep north Atlantic, where North Atlantic Deep Water formation has declined. The net effect of these physical changes is an overall decline in low-latitude productivity (driven by increased nutrient limitation) but a strong increase in Southern Ocean productivity, with a faster biogeochemical connection between the surface ocean south of the Polar Front, and the abyssal basins (see the improved ventilation in the radiocarbon plots). At the poles, fast-growing, nutrient-demanding phytoplankton functional types (DT, CP) thrive, while in the lower latitudes it is the more efficient nutrient consumers (DZ and LP) who benefit.

Long-term particle export trends generally follow the historical trend, but with increasing magnitude (Figures 21 and 27). Globally integrated particle fluxes decline, and remain suppressed with respect to pre-industrial rates, for POC and opal. PIC surface export rates change very little and deep export rates increase with climate forcing as a response to increasing surface calcifier POC export fluxes (e.g., Kvale et al., 2015a). The production of both opal and PIC are scaled against their respective plankton types' POC production, so it is expected that PIC and opal fluxes follow the POC export production trend. Just as with NPP, the POC export production decline in KMBM3 is larger than in previous versions (about 2.0 Pg C y$^{-1}$ by 2100, rather than 1.5 PgC y$^{-1}$).

POC and PIC fluxes increase where calcifying phytoplankton biomass also increases; south of 40°S, in the eastern equatorial Pacific upwelling zone, and along the Kuroshio Current into the North Pacific (Figure 27). Strong decreases in calcifiers, and associated deep carbon export, occur in the Indian Ocean and North Atlantic. Opal export declines by more than 100 mmol Si $m^{-2}$ $y^{-1}$ both south of 40°S and north of 40°N (with only small changes in the Arctic). However, opal export (and diatoms) increase south of 60°S, where increased nutrients (Figure 28), particularly iron, and a short growing season favours diatoms over calcifiers.

The historical trend in carbon and nutrients is similarly extended, with continuing increases in DIC in the upper ocean (as atmospheric $CO_2$ continues to enter the ocean), declines in low latitude upper ocean nutrients phosphate, nitrate, and silicic acid (due to decreasing resupply from the deep ocean), increases in the deep ocean in the same nutrients, and widespread declines in oxygen (Figure 28). Oxygen declines along the Southern Ocean- abyssal global ocean pathway due to both warming and increasing particle remineralisation, which is also responsible for the increasing nutrient concentrations in the deep ocean. Decreasing phosphate and nitrate concentrations in the sub-surface tropical ocean basins are a product of declining particle remineralisation there, brought about by both warming, which shoals remineralisation and increases respiration rates, and a shift to less efficiently exporting phytoplankton (LP).

The striking trend in dissolved iron that emerges in these future projections of strongly increasing (more than 80 nmol $m^{-3}$ in the zonal mean) concentrations in the upper ocean was previously described by Nickelsen et al. (2015). They attributed the increase to stratification "trapping" aerosol iron near the surface. However, the regions showing the greatest increases in dissolved iron are also the regions experiencing both strong declines in NPP (and hence, lower iron uptake) and strong declines in particle export (and hence, less particulate iron scavenging and removal). The loss of calcifiers in the Indian Ocean and central Pacific particularly increases iron concentrations there, because of the dual effect of reduced POC and PIC scavenging of iron.

Our future simulation results broadly agree with other long-term simulations in the sustained, and significant, increase in Southern Ocean primary production that couples with a reorganisation of deep ocean circulation to produce a long term "nutrient trapping" effect in Southern Ocean-sourced interior water masses (e.g., Moore et al., 2018; Kvale et al., 2019). Near-surface increases in iron, and decreases in nitrate, phosphate, and silicic acid, have also been observed to 2100 in a comparison of 9 other earth system models by Fu et al. (2016). These same models also simulate weak to strong increases in diatoms in the Southern Ocean to 2100, though in most, if not all, of them, diatoms are the most efficient exporters of carbon and nutrients (unlike in KMBM3). Phytoplankton community composition and export formulation were discussed by Fu et al. (2016) to be of critical importance in determining trends in NPP, nutrients, and particle export over the coming century, thus a diversity of model formulations benefits our understanding of how the global ocean ecosystem might change in the future.

## 5 Conclusions

Our manuscript describes a new model of the marine silicate cycle (KMBM3), evaluates its performance against previous KMBM versions and other earth system models, as well as key biogeochemical data derived from observations of the ocean,

and compares long-term ecosystem projections to similar models available in the literature. We find our new model shows general improvement in the representation of nutrients and particle fluxes with respect to previous versions of KMBM and is mechanistically more realistic, with the added complexity of iron, calcite, and silicate merged into a single model code. Furthermore, its representation of the silicate cycle has similar performance with other state-of-the-art biogeochemical models.

Simulations using our new model suggest diatoms have been, and will continue to be, the losers as the earth system warms. Their high nutrient requirements prove a disadvantage as the upper ocean stratifies, and small gains in productivity provided by sea ice retreat cannot compensate for the fact that their southern bound is ultimately limited geographically. Calcifying phytoplankton with more moderate nutrient requirements are the big winners across the high latitudes, while in the tropics slow-growing, less nutrient-hungry phytoplankton are projected to thrive. From a deep ocean carbon sequestration perspective, the loss of diatom export production is of transient importance, as the calcifying phytoplankton increase their role in carbon export, efficiently sinking organic carbon as well as carbonate.

Our simulations also reveal the past may not accurately portray future trends, as evidenced by simulated historical declines in NPP in the Southern Ocean that reverse as conditions become more favourable for calcifiers. Significant and rapid increases in dissolved iron in the low latitude tropical ocean is another potential biogeochemical "surprise", still to come, if anthropogenic emissions of carbon follow the present trajectory.

Several aspects of KMBM3, including iron scavenging by calcite, silicate source and sink strengths, and different zooplankton grazing preferences are slated for further study. The impact of variable stoichiometry is another important potential aspect of biogeochemical modelling that is not explored here. More complete parameter assessment is planned in the context of offline parameter optimisation and model calibration experiments (e.g., Kriest, 2017; Kriest et al., 2017; Yao et al., 2019) in the future, as is merging this new biogeochemical model into the latest UVic ESCM version 2.10 (Mengis et al., 2020). We look forward to further refinements, and the many applications of this model to come.

*Code and data availability.* Data and model code used in the writing of the manuscript is available on the OPeNDAP GEOMAR server at https://dx.doi.org/20.500.12085/34412098-27f9-4cbb-992d-12d0d342aa45. The KMBM3 code released with our manuscript is only part of all model code required to use the UVic ESCM version 2.9, Updates 02. The UVic ESCM model code is found at http://terra.seos.uvic.ca/model/. The KMBM3 code is provided freely, but with the requirement that prospective users contact K Kvale with their research plans to avoid parallel projects emerging.

**Instructions for model use**

The KMBM3 code described, and released, with our manuscript is only part of all code needed to use the UVic ESCM version 2.9. UVic ESCM 2.9 Updates 02 must be downloaded from http://terra.seos.uvic.ca/model/. Please follow the instructions on this webpage for installation and use.

KMBM3 code released with our manuscript is available at https://dx.doi.org/20.500.12085/34412098-27f9-4cbb-992d-12d0d342aa45. The code should be called first in the "mk.in" control file, with subsequent calls to the base model code.

*Author contributions.* KK designed the model, and wrote the model code and paper. DPK, WK, CS, and WY contributed model code and bug fixes from previous model versions. KJM and AO contributed to the design of the model. All co-authors contributed to the analysis of the transient simulations and edited the manuscript.

*Competing interests.* All authors declare no competing interests.

*Acknowledgements.* The authors would like to acknowledge computer resources made available by Kiel University, as well as topical discussion with M. Eby at the University of Victoria, plotting scripts from A. Schmittner (Oregon State University) and I. Kriest (GEOMAR), plotting scripts and data from M. Pahlow (GEOMAR), H. Geisinger who created a hydrothermal silicate mask, H. Saini for code testing, and the rest of the Biogeochemical Modelling Department at GEOMAR for their kindness and expertise. K.Kvale acknowledges support from GEOMAR Helmholtz Centre for Ocean Research, Kiel, and the New Zealand Ministry of Business, Innovation and Employment through the Global Change through Time Program. KJM acknowledges support from the Australian Research Council (DP180100048 and DP180102357). Model tuning was performed in part at the NCI National Facility at the Australian National University, through awards under the National Computational Merit Allocation Scheme, the Intersect allocation scheme, and the UNSW HPC at NCI Scheme. Figure plotting used the Ferret plotting program. Ferret is a product of NOAA's Pacific Marine Environmental Laboratory.

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

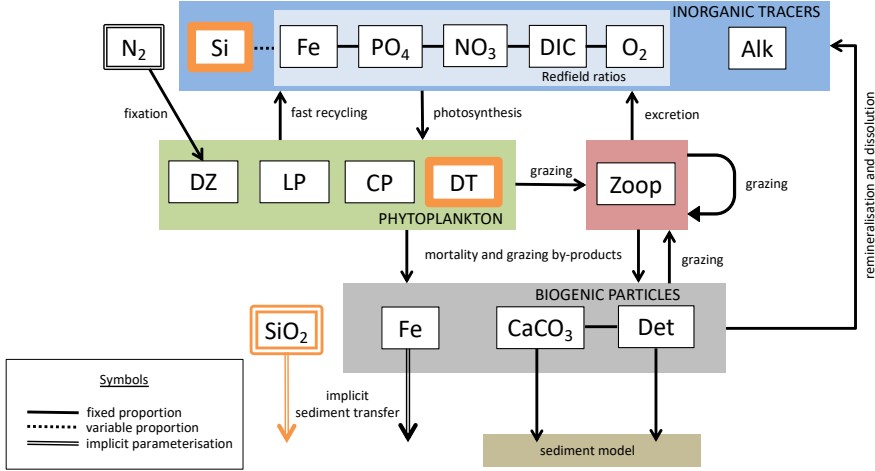

**Figure 1.** The latest biogeochemical model structure for the KMBM3. Previously unpublished features are shown in orange.

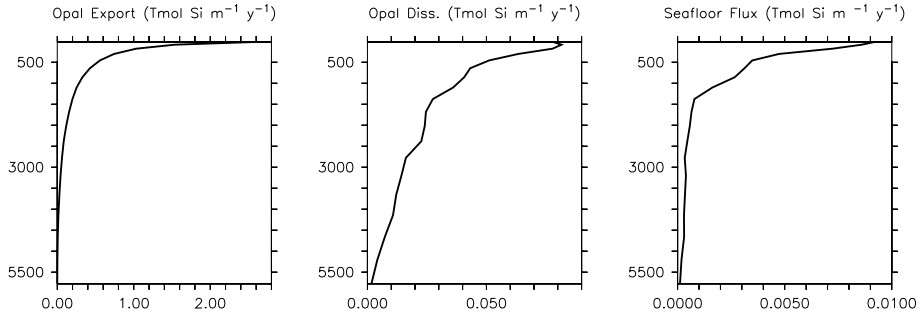

**Figure 2.** Implicit opal export (left panel), water column dissolution (middle panel), and seafloor burial (right panel) at model year 2014. Rates are globally integrated in the latitudinal and longitudinal planes but depth information is retained. Thus, opal export, dissolution, and seafloor flux are reported in Tmol Si flux per meter of depth per year.

H., Schnur, R., Schulzweida, U., Six, K., Stevens, B., Voigt, A., and Roeckner, E.: MPI-M MPI-ESM1.2-LR model output prepared for CMIP6 CMIP historical, https://doi.org/10.22033/ESGF/CMIP6.6595, 2019.

Yao, W., Kvale, K. F., Achterberg, E., Koeve, W., and Oschlies, A.: Hierarchy of calibrated global models reveals improved distributions and fluxes of biogeochemical tracers in models with explicit representation of iron, Environmental Research Letters, 14, 114 009, https://doi.org/10.1088/1748-9326/ab4c52, 2019.

Yasunaka, S., Ono, T., Nojiri, Y., Whitney, F. A., Wada, C., Murata, A., Nakaoka, S., and Hosoda, S.: Long-term variability of surface nutrient concentrations in the North Pacific, Geophysical Research Letters, 43, 3389–3397, https://doi.org/10.1002/2016GL068097, 2016.

Zhang, Y., Mahowald, N., Scanza, R. A., Journet, E., Desboeufs, K., Albani, S., Kok, J. F., Zhuang, G., Chen, Y., Cohen, D. D., Paytan, A., Patey, M. D., Achterberg, E. P., Engelbrecht, J. P., and Fomba, K. W.: Modeling the global emission, transport and deposition of trace elements associated with mineral dust, Biogeosciences, 12, 5771–5792, https://doi.org/10.5194/bg-12-5771-2015, 2015.

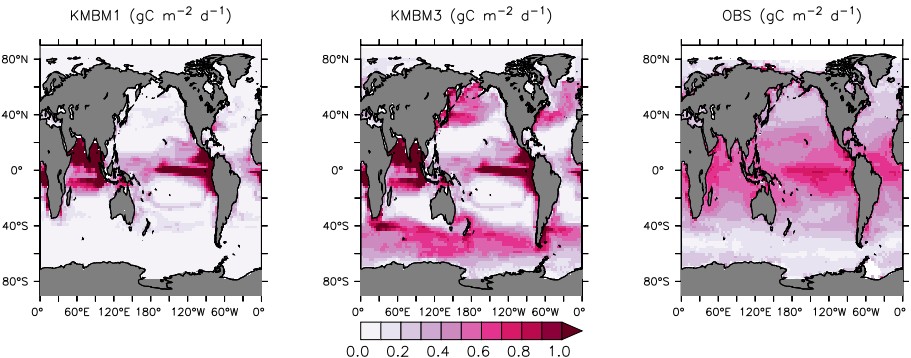

**Figure 3.** Comparison of model NPP output at 2014 (left panel; Keller et al. (2012), and middle panel; this model) and satellite-derived NPP climatology (NASA, 2018), 2012-2018 (right panel) in gC m$^{-2}$ day$^{-1}$.

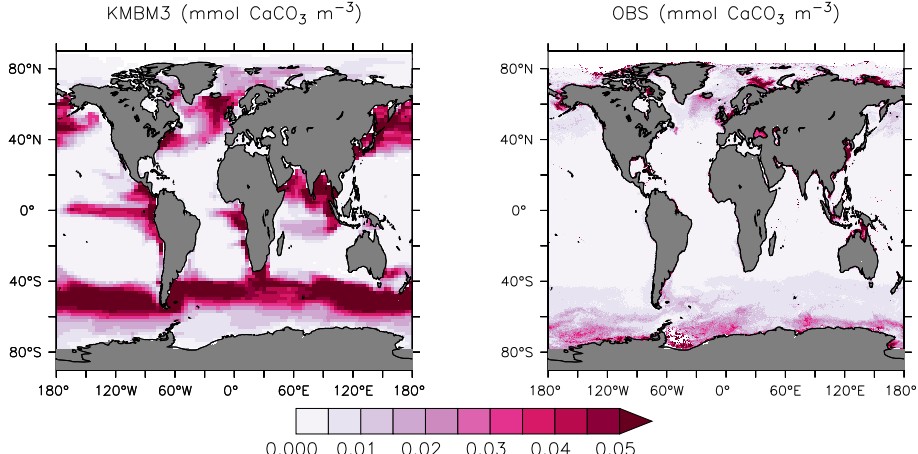

**Figure 4.** Comparison of model average 2004-2014 (left panel) and satellite CaCO$_3$ climatology, 2002-2018 (right panel) in mmol CaCO$_3$ m$^{-3}$. Data product is scaled by the model grid in the z direction and in both plots only the upper 20 meters are represented, where uniform coccolith concentration is assumed (Balch and Utgoff, 2009).

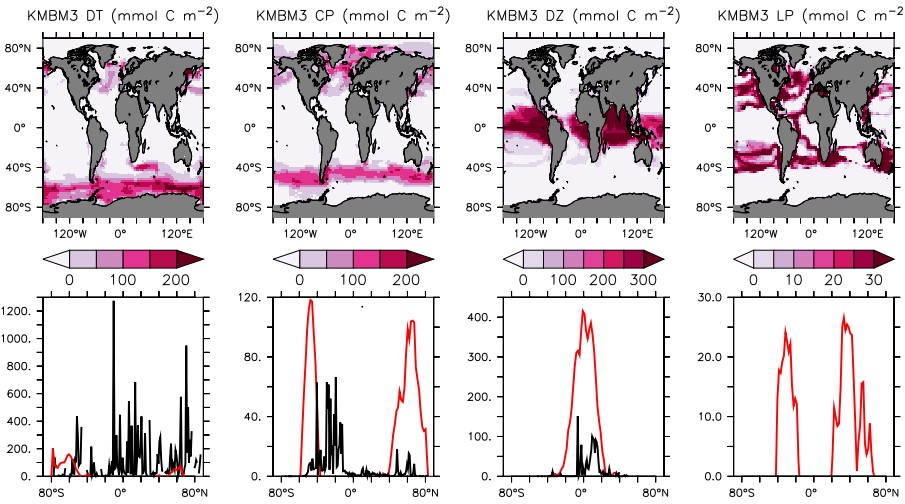

**Figure 5.** Annually-averaged and depth-integrated model biomass at year 2014 (top row). Zonally-averaged and depth-integrated model biomass plotted with the equivalent MAREDAT compilations (bottom row; Leblanc et al., 2012; O'Brien, 2012; Luo et al., 2012, no data compilation of "mixed" phytoplankton, a model-specific category). All units are mmol C m$^{-2}$. Black lines represent MAREDAT, red lines are the model output.

**Table 1.** KMBM3 state variables.

| Variable | Subscript Symbol | Units |
|---|---|---|
| Silica | Si | mol Si m$^{-3}$ |
| Iron | Fe | mol Fe m$^{-3}$ |
| Phosphate | $PO_4^{3-}$ | mol P m$^{-3}$ |
| Nitrate | $NO_3^-$ | mol N m$^{-3}$ |
| Dissolved inorganic carbon | DIC | mol C m$^{-3}$ |
| Calcite | $CaCO_3$ | mol C m$^{-3}$ |
| Living calcite | $CaCO_{3liv}$ | mol C m$^{-3}$ |
| Alkalinity | Alk | mol C m$^{-3}$ |
| Oxygen | $O_2$ | mol O m$^{-3}$ |
| Diazotrophs | DZ | mol N m$^{-3}$ |
| Diatoms | DT | mol N m$^{-3}$ |
| Low latitude phytoplankton | LP | mol N m$^{-3}$ |
| Calcifiers | CP | mol N m$^{-3}$ |
| Zooplankton | Z | mol N m$^{-3}$ |
| Free detritus | $Detr_{free}$ | mol N m$^{-3}$ |
| Ballasted detritus | $Detr_{bal}$ | mol N m$^{-3}$ |
| Detrital iron | $Detr_{Fe}$ | mol Fe m$^{-3}$ |

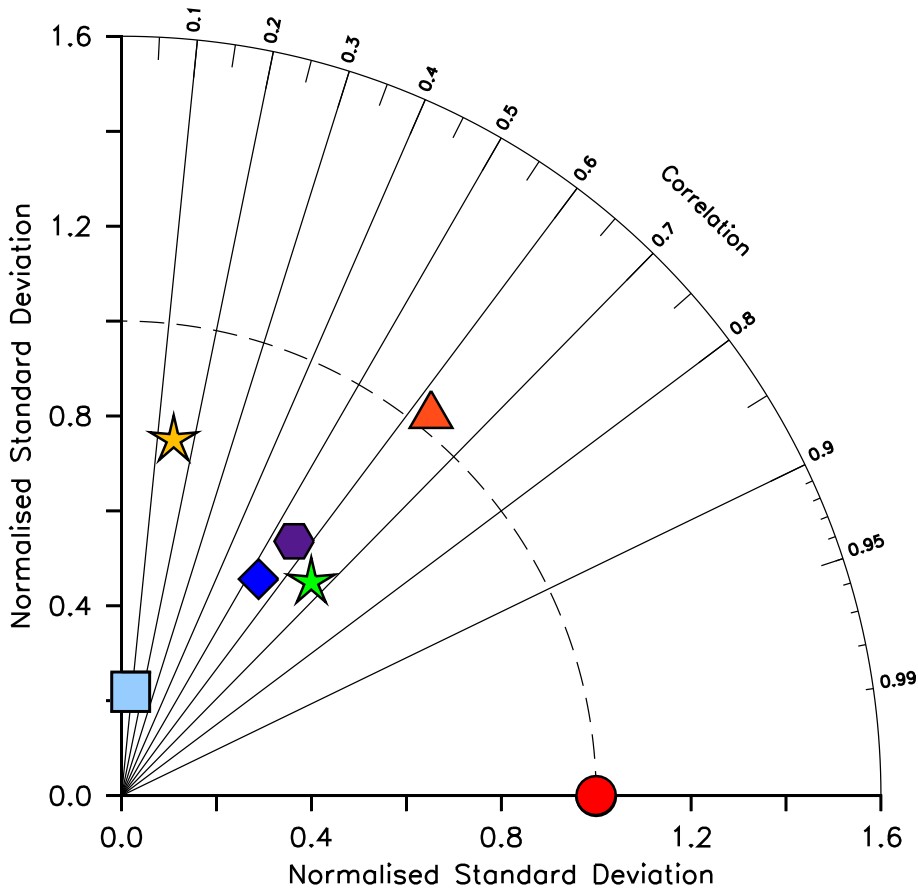

**Figure 6.** Taylor diagram (Taylor, 2001) of CMIP6 annual average diatom biomass concentrations (in mol C m$^{-3}$) at year 2014 referenced to KMBM3. The distance to the origin represents the normalised standard deviation. Normalised correlation with KMBM3 is read from the azimuthal position. Perfect agreement with KMBM3 is a normalised standard deviation of 1 and a normalised correlation of 1. Normalisation against the model, rather than observations, is shown due to the models having greater similarity with each other than with the very sparse observations of diatom biomass currently available. Models are KMBM3 (red circle), GFDL-ESM4 (green star; Krasting et al., 2018), CanESM5-CanOE (yellow star; Swart et al., 2019), CESM2 (light blue square; Danabasoglu, 2019), CMCC-ESM2 (orange triangle; Lovato et al., 2021), EC-Earth3-CC (blue diamond; EC-Earth Consortium, 2021), and IPSL-CM6A-LR (purple hexagon; Boucher et al., 2018). With the exception of KMBM3, the data for all models were obtained by mining the CMIP6 database (https://esgf-node.llnl.gov/search/cmip6/) using the following search terms: CMIP/phydiat/historical/annual output. Data were accessed between 01-05.02.2021.

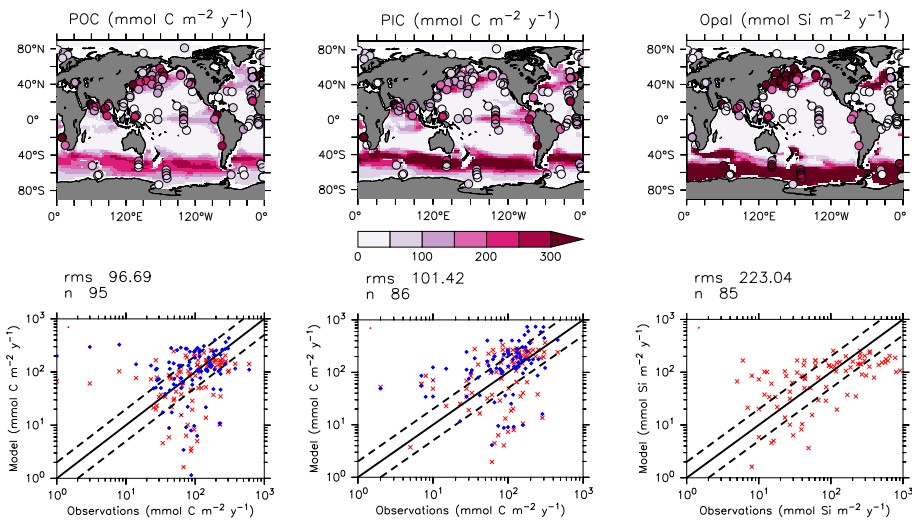

**Figure 7.** Comparison of model POC (left panels), CaCO₃ (middle panels), and opal (right panels) fluxes at year 2004 and 2 km depth with Honjo et al. (2008) data. Red points are this model (KMBM3), blue points are the Keller et al. (2012) model (KMBM1). Statistics are given for KMBM3 only.

**Table 2.** Miscellaneous KMBM3 parameters.

| Parameter | Symbol | Units | Value |
|-----------|--------|-------|-------|
| E-folding temperature | $T_b$ | $^\circ$C | 15.65 |
| Molar organic P:N ratio | $R_{P:N}$ | unitless | 0.0625 |
| Molar organic C:N ratio | $R_{C:N}$ | unitless | 6.625 |
| Molar organic O:N ratio | $R_{O:N}$ | unitless | 8.46 |
| Molar sedimentary Fe:P ratio | $R_{Fe:P_{sed}}$ | unitless | 0.004 |
| Molar organic Fe:N ratio | $R_{Fe:N}$ | unitless | 6.625E-6 |
| Molar mass of carbon | $M_C$ | g M$^{-1}$ | 12.011 |
| Molar mass of carbonate | $M_{CaCO_3}$ | g M$^{-1}$ | 60.01 |
| Light attenuation by phytoplankton | $k_c$ | (m mmol m$^{-3}$)$^{-1}$ | 0.07 |
| Light attenuation by CaCO₃ | $k_{CaCO_3}$ | (m mmol m$^{-3}$)$^{-1}$ | 0.2 |
| Light attenuation by ice | $k_i$ | m$^{-1}$ | 5.0 |
| Light attenuation by water | $k_w$ | m$^{-1}$ | 0.04 |

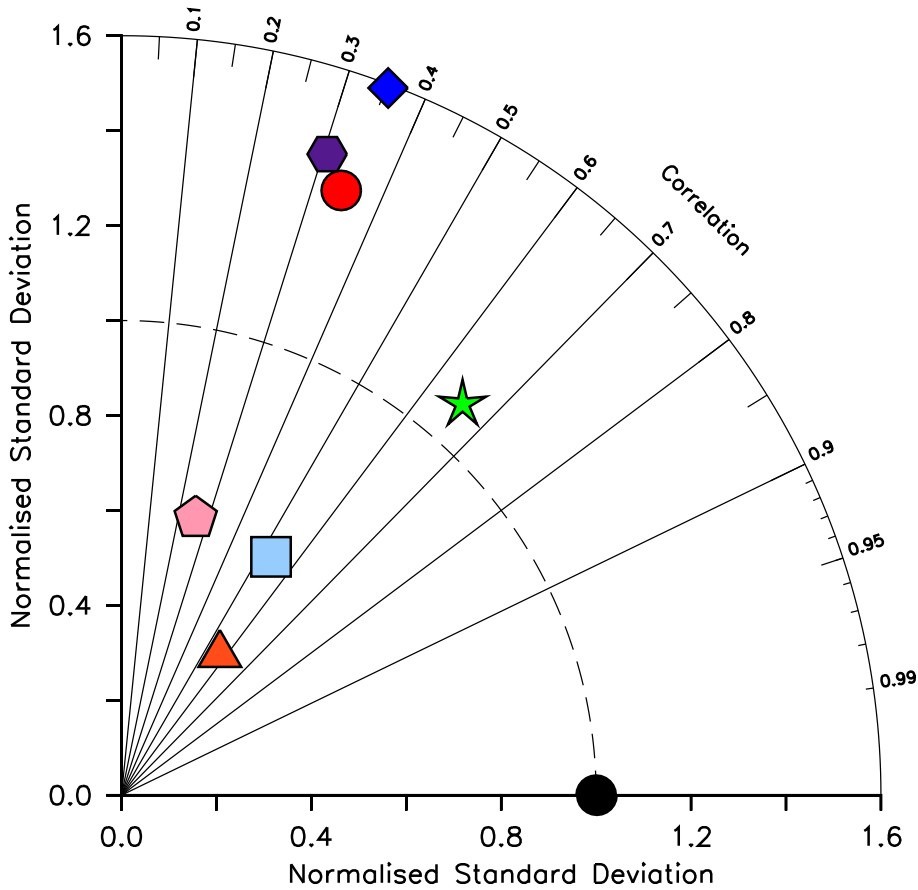

**Figure 8.** Taylor diagram (Taylor, 2001) of CMIP6 annual average opal flux (in mol Si m$^{-2}$ y$^{-2}$) at 2 km depth, for year 2014, normalised to the Honjo et al. (2008) dataset. The distance to the origin represents the normalised standard deviation. Normalised correlation with observations is read from the azimuthal position. Perfect agreement with observations is a normalised standard deviation of 1 and a normalised correlation of 1. Models are KMBM3 (red circle), GFDL-ESM4 (green star; Krasting et al., 2018), CESM2 (light blue square; Danabasoglu, 2019), MPI-ESM1.2-LR (pink pentagon; Wieners et al., 2019), CMCC-ESM2 (orange triangle; Lovato et al., 2021), EC-Earth3-CC (blue diamond; EC-Earth Consortium, 2021), and IPSL-CM6A-LR (purple hexagon; Boucher et al., 2018). With the exception of KMBM3, the data for all models were obtained by mining the CMIP6 database (https://esgf-node.llnl.gov/search/cmip6/) using the following search terms: CMIP/expsi/historical/annual output. Data were accessed between 01-05.02.2021.

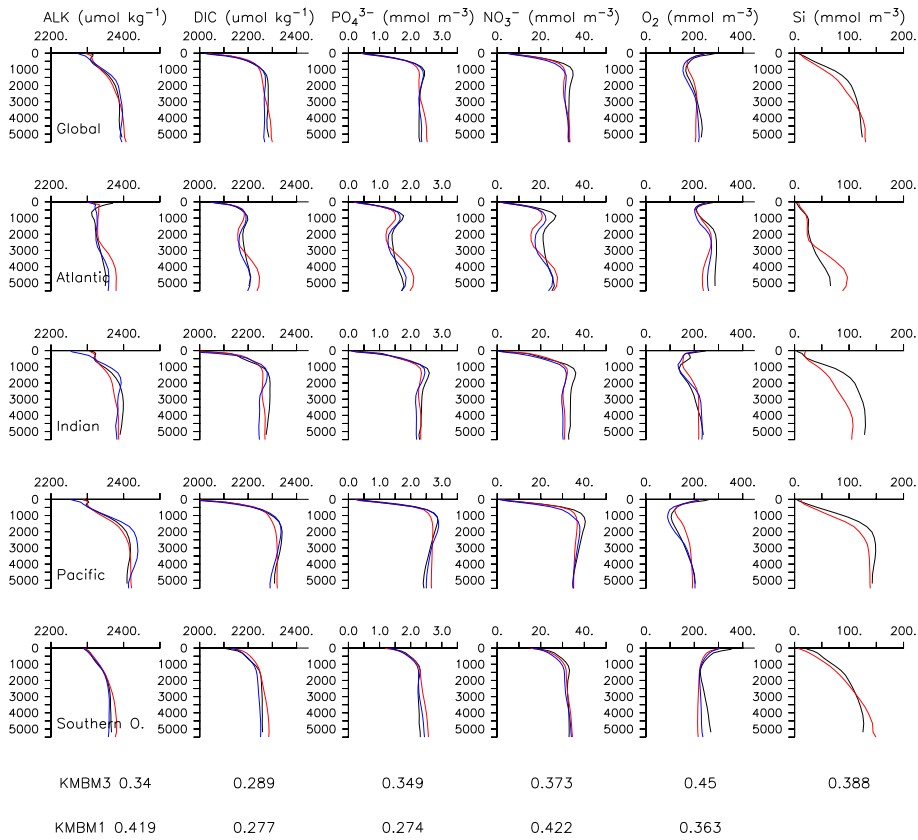

**Figure 9.** Model global and basin average nutrient and carbon depth profiles at year 2014 (KMBM3; red lines), compared to model output from the Keller et al. (2012) version (KMBM1; blue lines) and observational data (black lines; Garcia et al., 2014a, b; Key et al., 2015; Lauvset et al., 2016). Global root mean square error is given below each column.

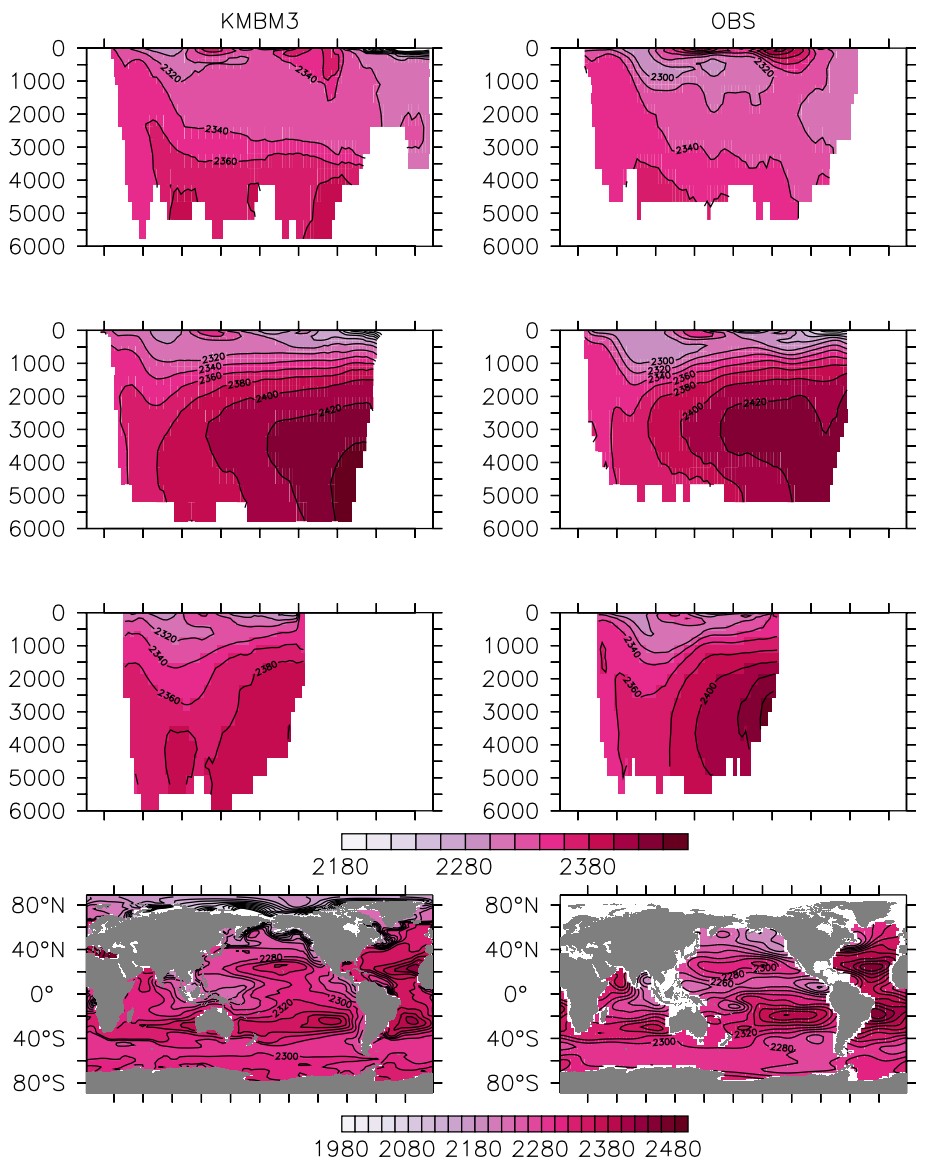

**Figure 10.** Model (left column) year 2014 alkalinity ($\mu$mol kg$^{-1}$) averaged by basin compared to GLODAP (right column; Key et al., 2015; Lauvset et al., 2016). Regions are as follows: Atlantic (top row), Pacific (second row), Indian (third row) and global surface (bottom row).

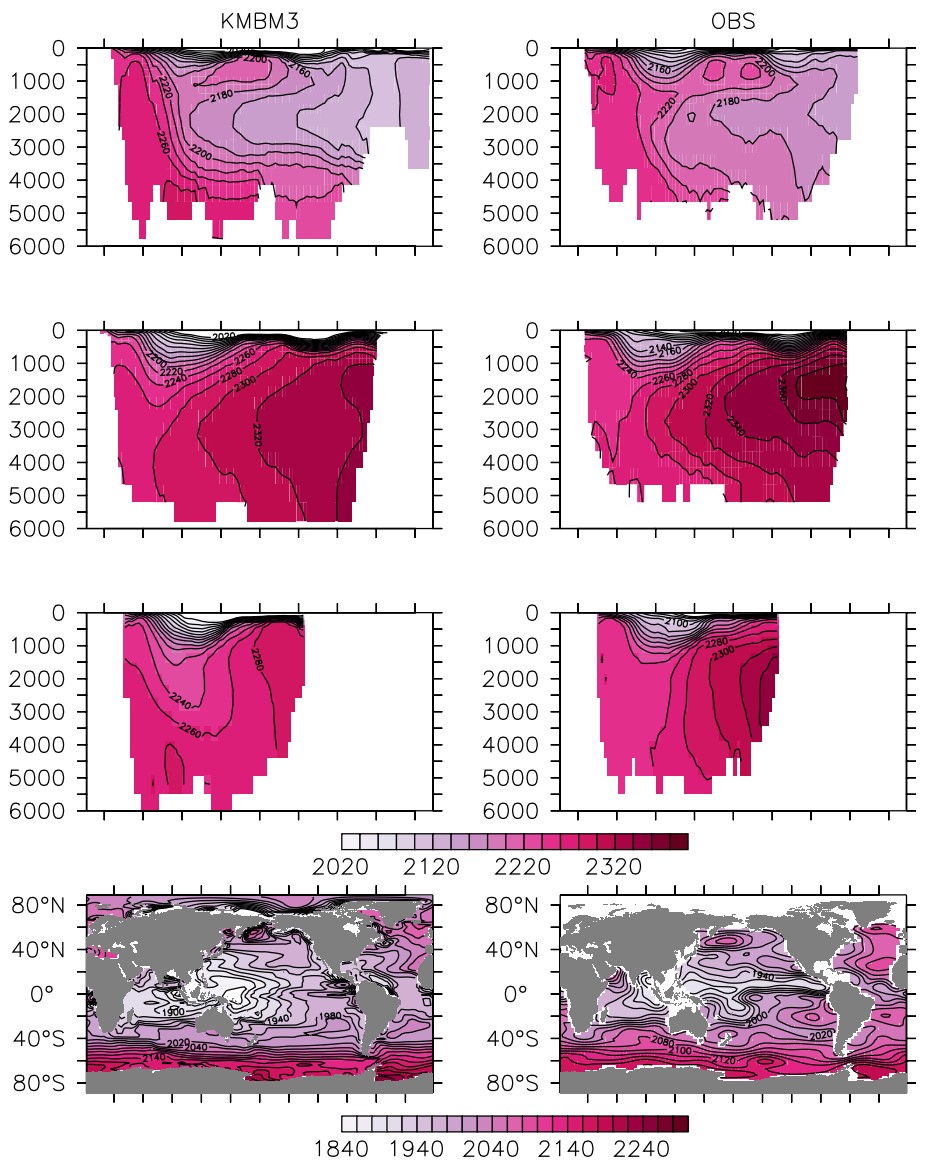

**Figure 11.** Model (left column) year 2014 DIC ($\mu$mol kg$^{-1}$) averaged by basin compared to GLODAP (right column; Key et al., 2015; Lauvset et al., 2016). Regions are as follows: Atlantic (top row), Pacific (second row), Indian (third row) and global surface (bottom row).

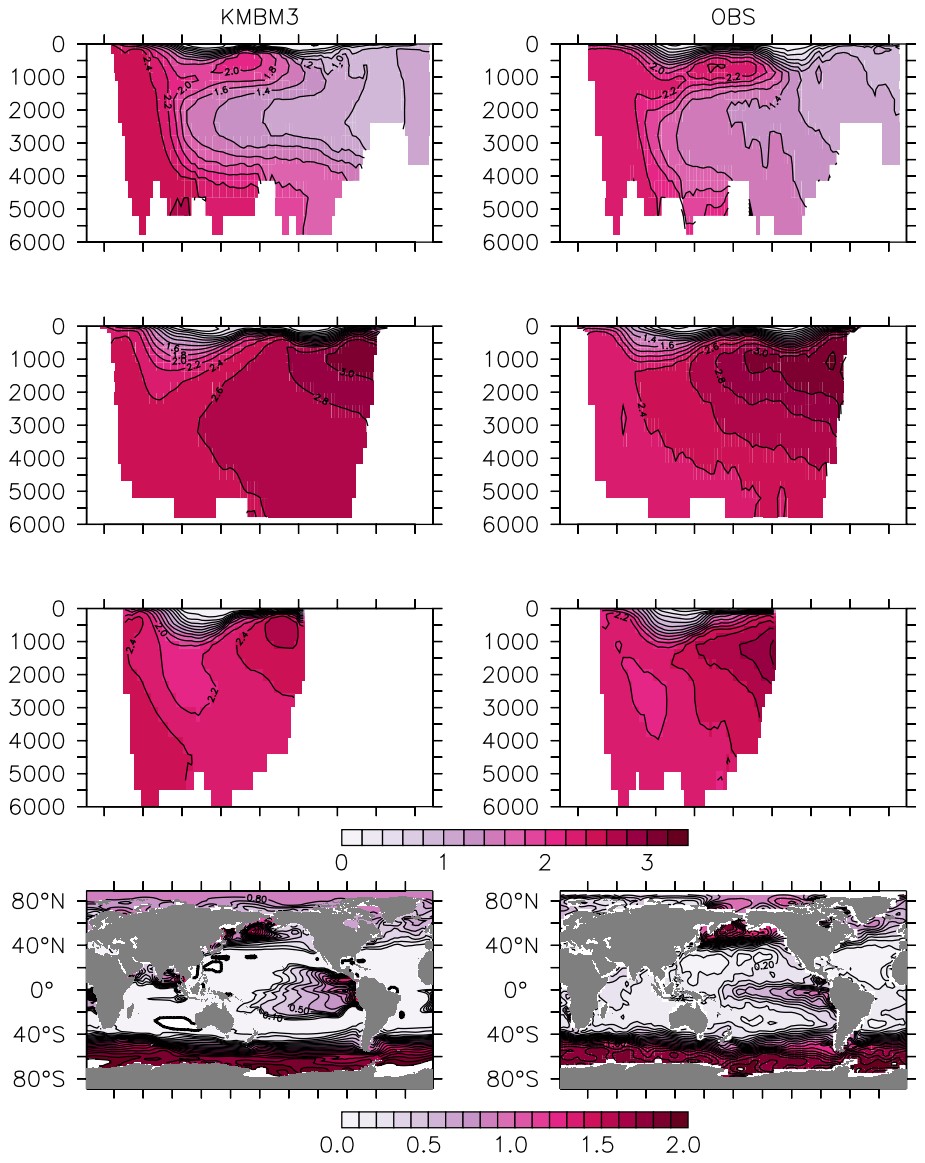

**Figure 12.** Model (left column) year 2014 $PO_4^{3-}$ (mmol m$^{-3}$) averaged by basin compared to WOA (right column; Garcia et al., 2014b). Regions are as follows: Atlantic (top row), Pacific (second row), Indian (third row) and global surface (bottom row).

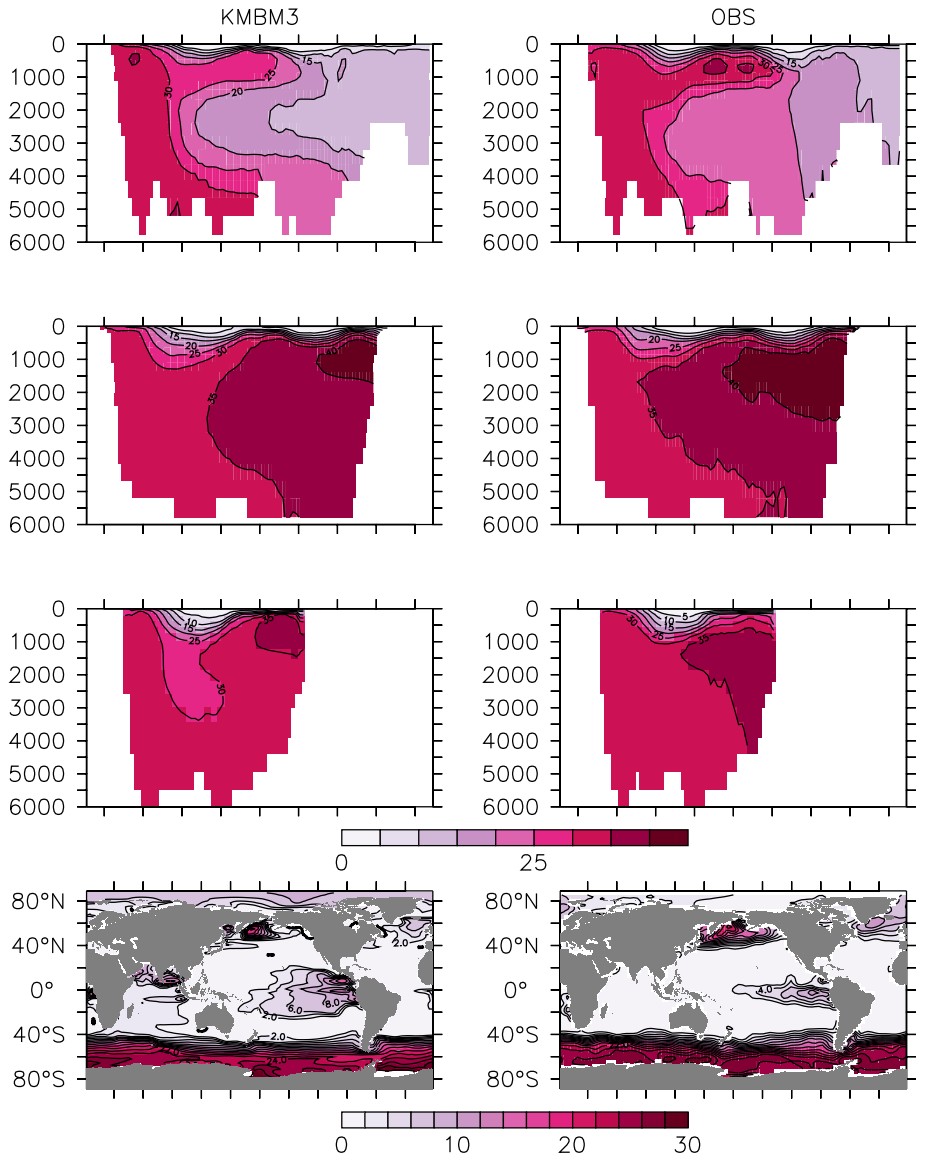

**Figure 13.** Model (left column) year 2014 $NO_3^-$ (mmol m$^{-3}$) averaged by basin compared to WOA (right column; Garcia et al., 2014b). Regions are as follows: Atlantic (top row), Pacific (second row), Indian (third row) and global surface (bottom row).

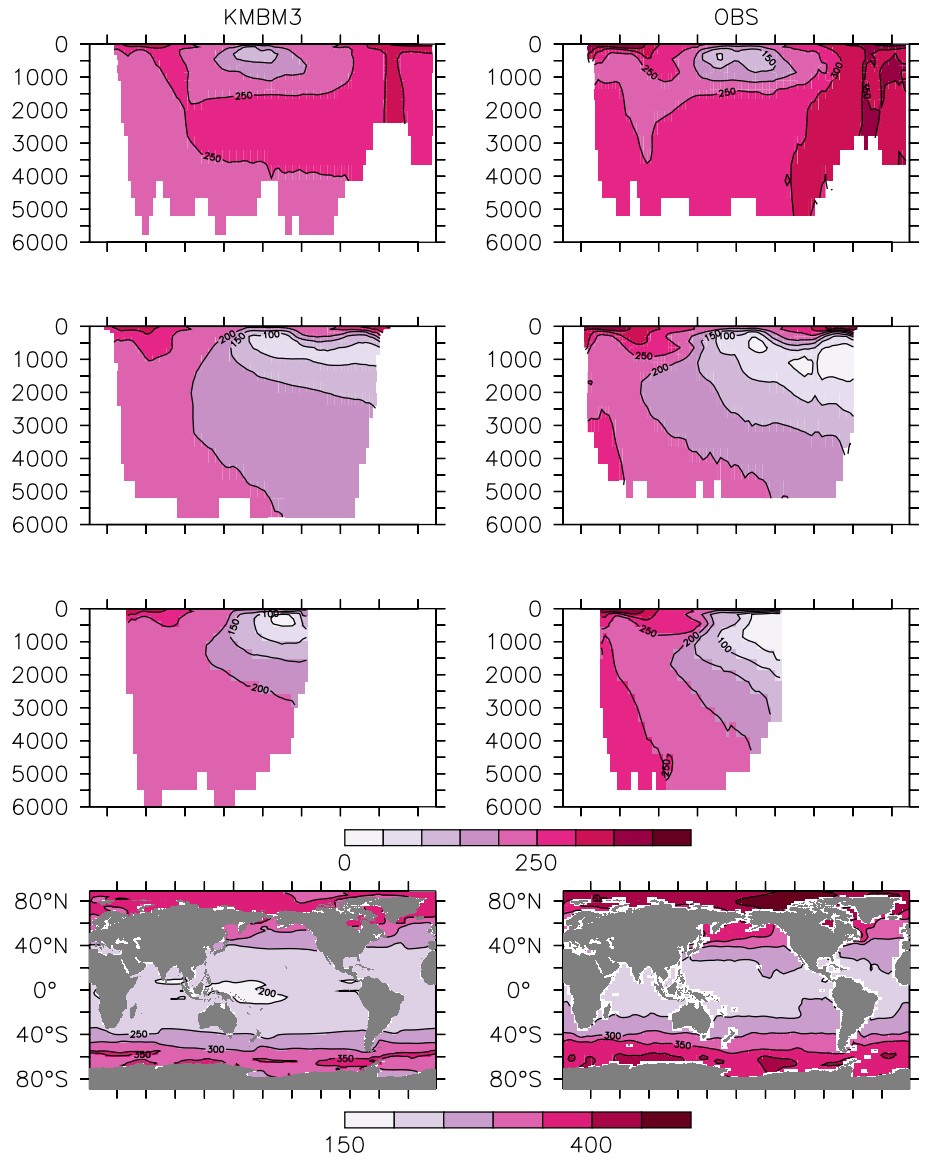

**Figure 14.** Model (left column) year 2014 $O_2$ (mmol m$^{-3}$) averaged by basin compared to WOA (right column; Garcia et al., 2014a). Regions are as follows: Atlantic (top row), Pacific (second row), Indian (third row) and global surface (bottom row).

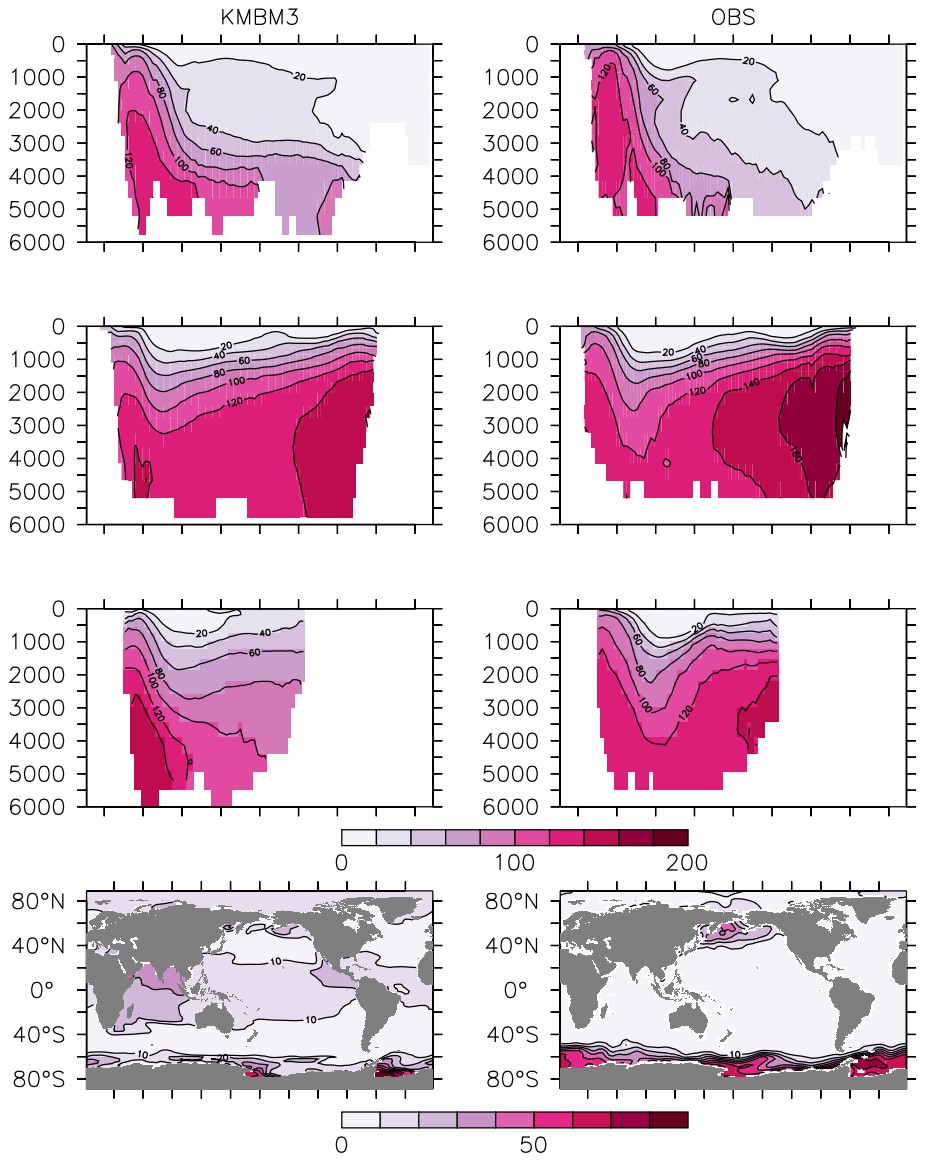

**Figure 15.** Model (left column) year 2014 Si (mmol m$^{-3}$) averaged by basin compared to WOA (right column; Garcia et al., 2014b). Regions are as follows: Atlantic (top row), Pacific (second row), Indian (third row) and global surface (bottom row).

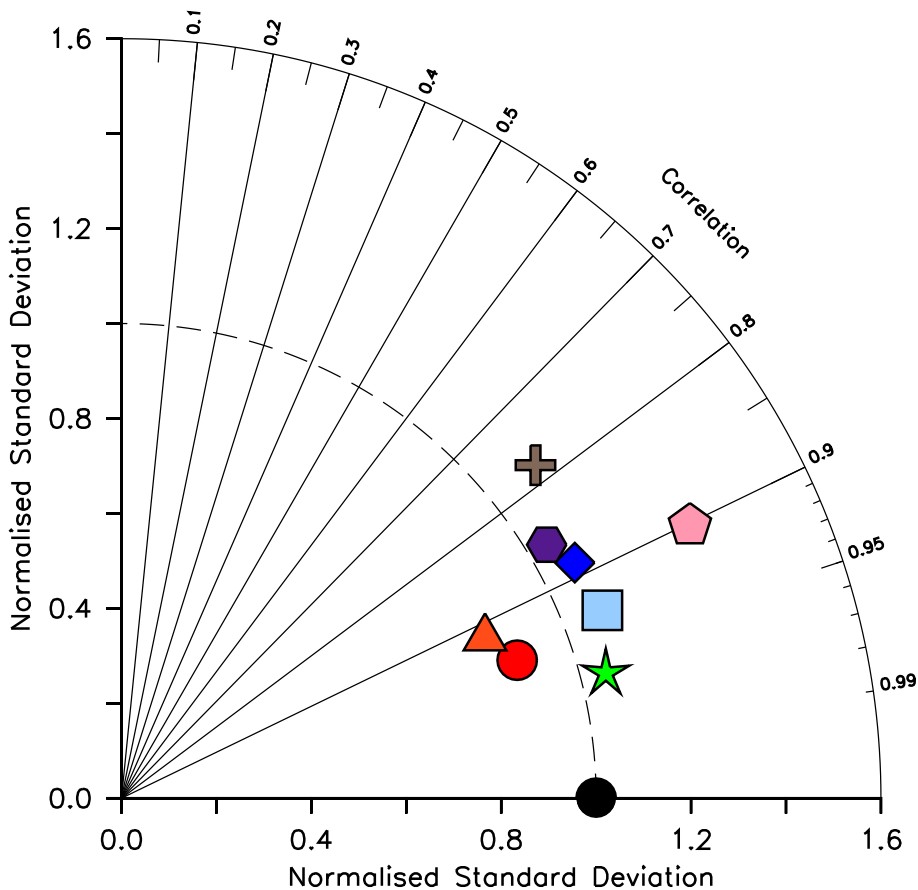

**Figure 16.** Taylor diagram (Taylor, 2001) of CMIP6 annual average dissolved silicic acid distribution (in mol Si m$^{-3}$) for year 2014, normalised to the Garcia et al. (2014b) dataset. The distance to the origin represents the normalised standard deviation. Normalised correlation with observations is read from the azimuthal position. Perfect agreement with observations is a normalised standard deviation of 1 and a normalised correlation of 1. Models are KMBM3 (red circle), GFDL-ESM4 (green star; Krasting et al., 2018), CESM2 (light blue square; Danabasoglu, 2019), MPI-ESM1.2-LR (pink pentagon; Wieners et al., 2019), NorESM2-LM (brown plus; Seland et al., 2019), CMCC-ESM2 (orange triangle; Lovato et al., 2021), EC-Earth3-CC (blue diamond; EC-Earth Consortium, 2021), and IPSL-CM6A-LR (purple hexagon; Boucher et al., 2018). With the exception of KMBM3, the data for all models were obtained by mining the CMIP6 database (https://esgf-node.llnl.gov/search/cmip6/) using the following search terms: CMIP/si/historical/annual output. Data were accessed between 01-05.02.2021.

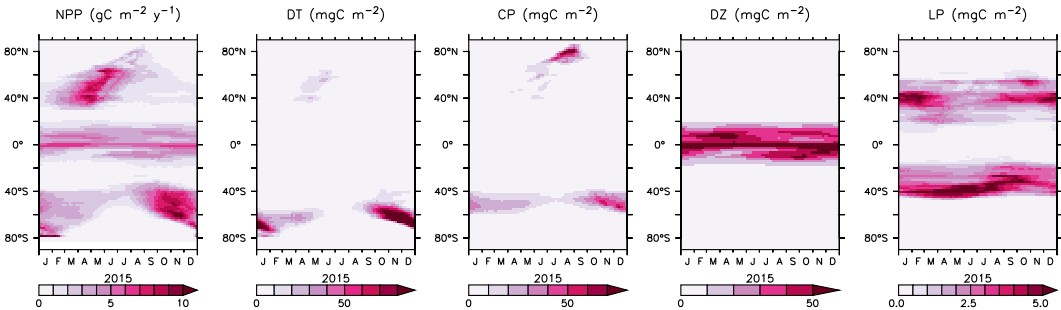

**Figure 17.** Model year 2015 zonally averaged surface NPP and phytoplankton biomass.

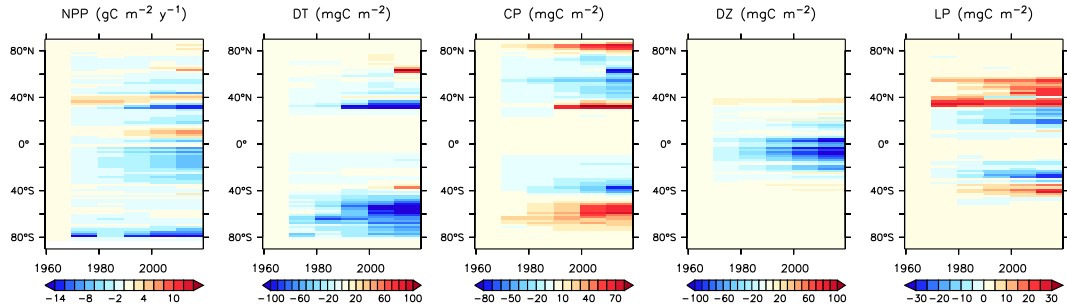

**Figure 18.** Modelled historical changes in zonally averaged and depth integrated NPP, and phytoplankton biomass, from 1964 to 2014.

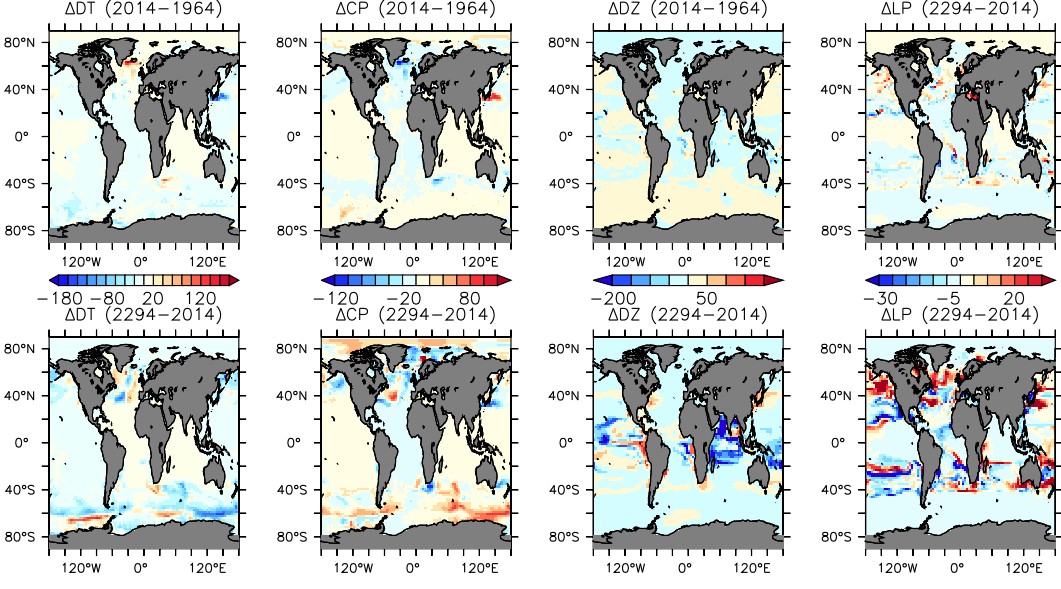

**Figure 19.** Modelled changes in phytoplankton biomass (mmol C m$^{-3}$), from 1964 to 2014 (top row), and from 2014 to 2294 (bottom row).

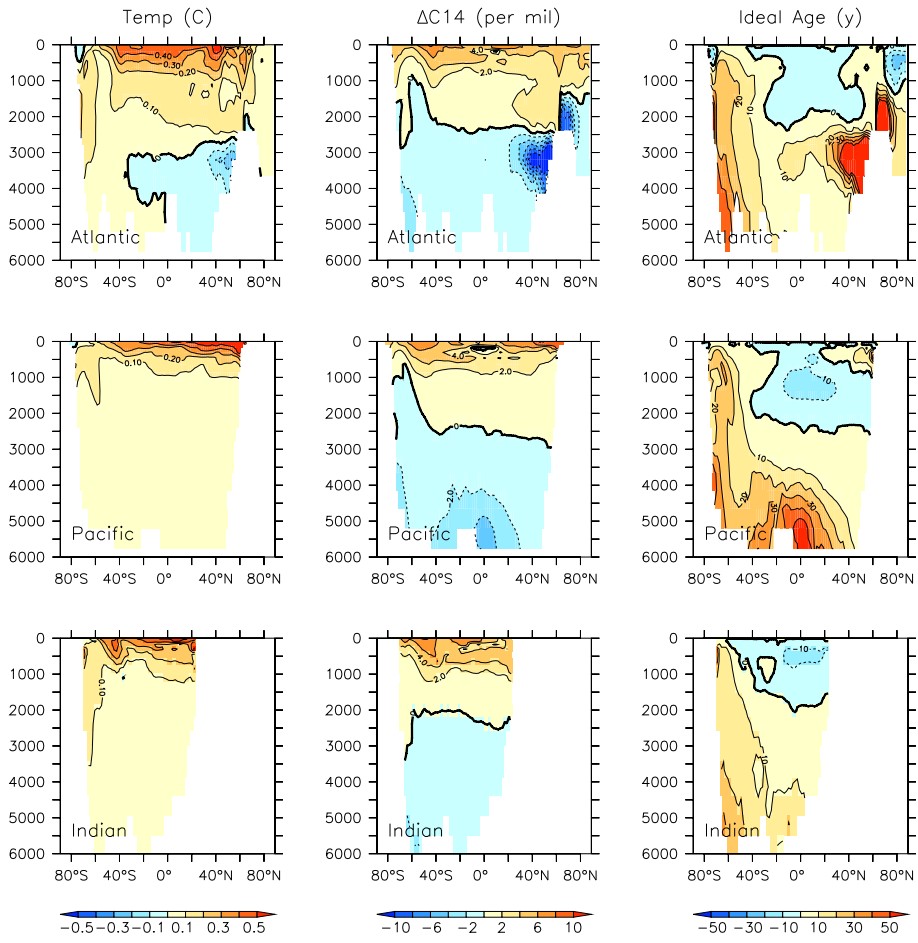

**Figure 20.** Modelled change in temperature, radiocarbon, and ideal age profiles by major ocean basin, from 1964 to 2014.

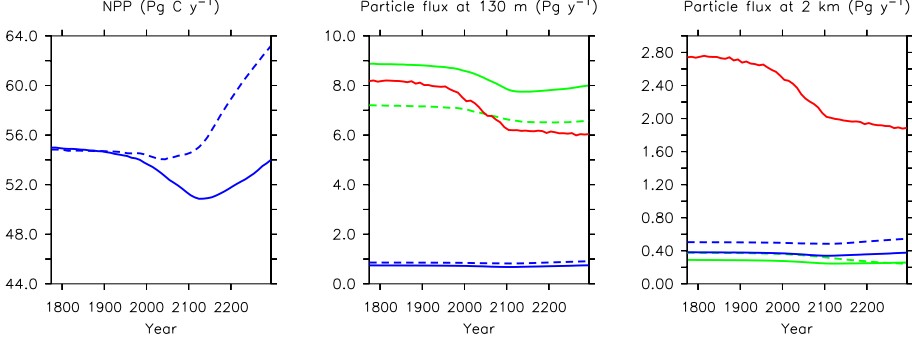

**Figure 21.** Major biogeochemical fluxes in the model, 1776 to 2300. Left panel is total net primary production, middle panel is total particle fluxes at 130 m depth, right panel is total particle fluxes at 2 km depth. Blue lines in the middle and right panels are PIC, green lines are POC, red lines are opal. Solid lines are from this model (KMBM3), dashed lines from the Keller et al. (2012) version (KMBM1).

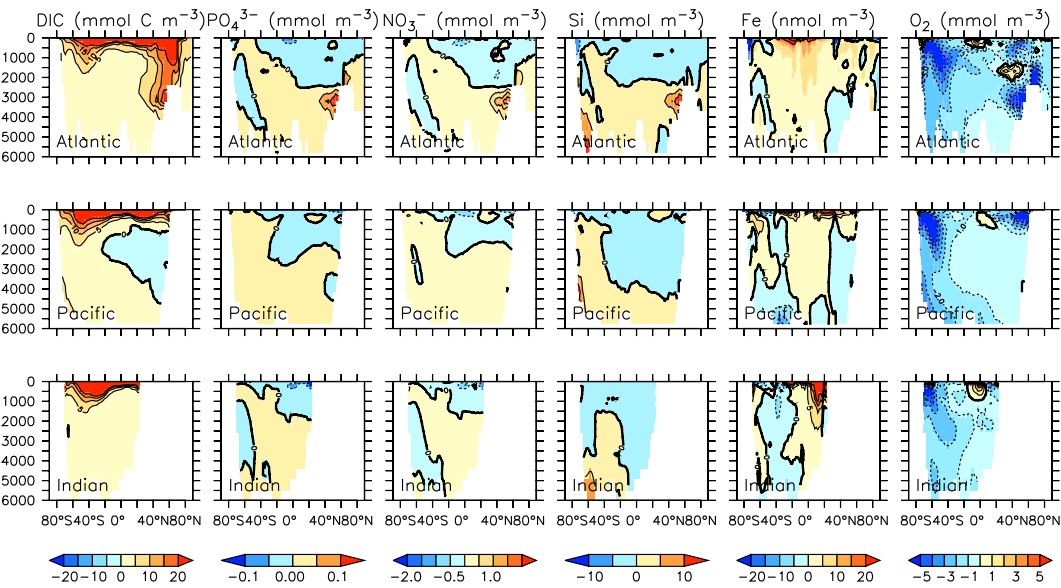

**Figure 22.** Modelled change in carbon and nutrient profiles by major ocean basin, from 1964 to 2014.

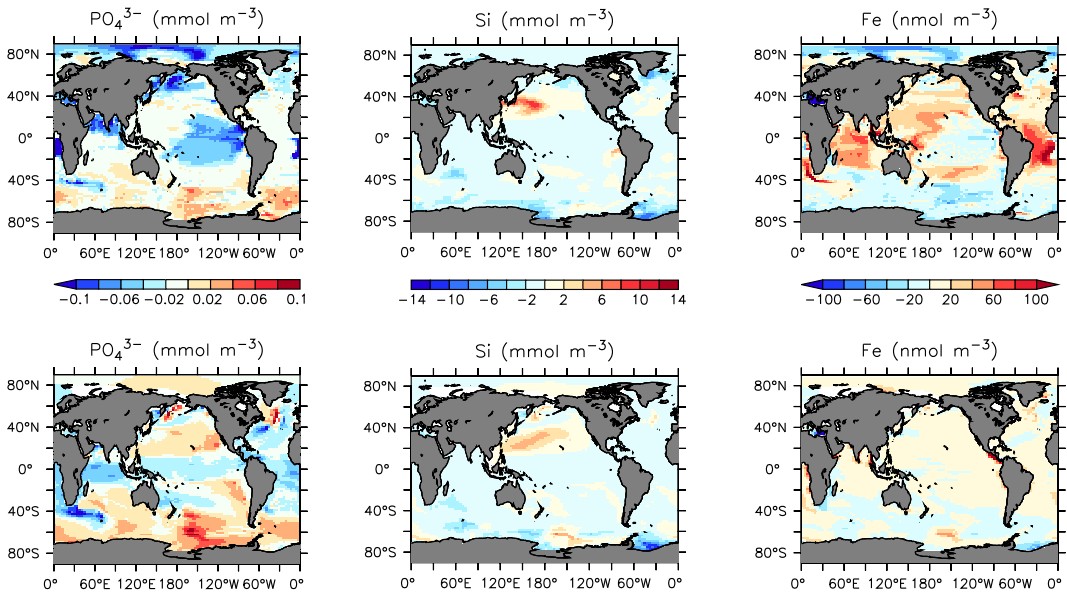

**Figure 23.** Modelled change in nutrients at the surface (top row) and 300 m depth (bottom row), from 1964 to 2014.

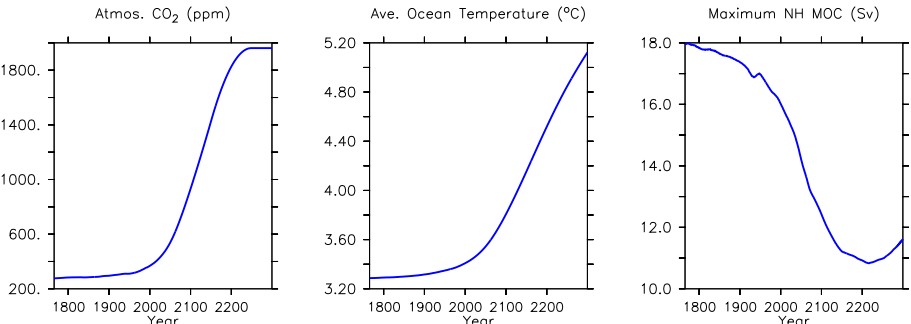

**Figure 24.** Major physical changes in the model, 1776 to 2300.

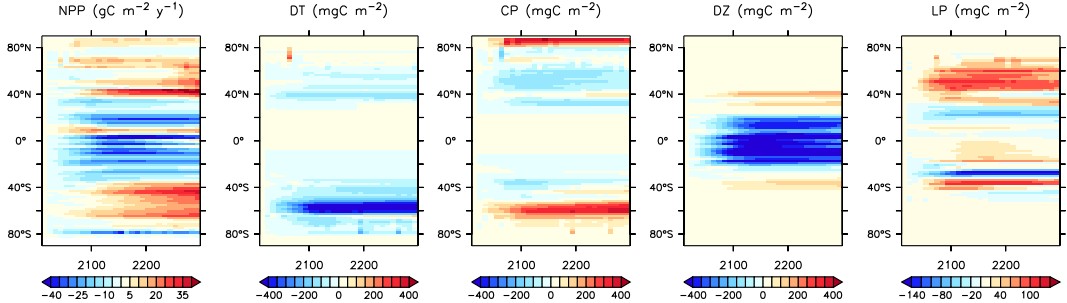

**Figure 25.** Modelled changes in zonally averaged and depth integrated NPP, and phytoplankton biomass, from 2014 to 2294.

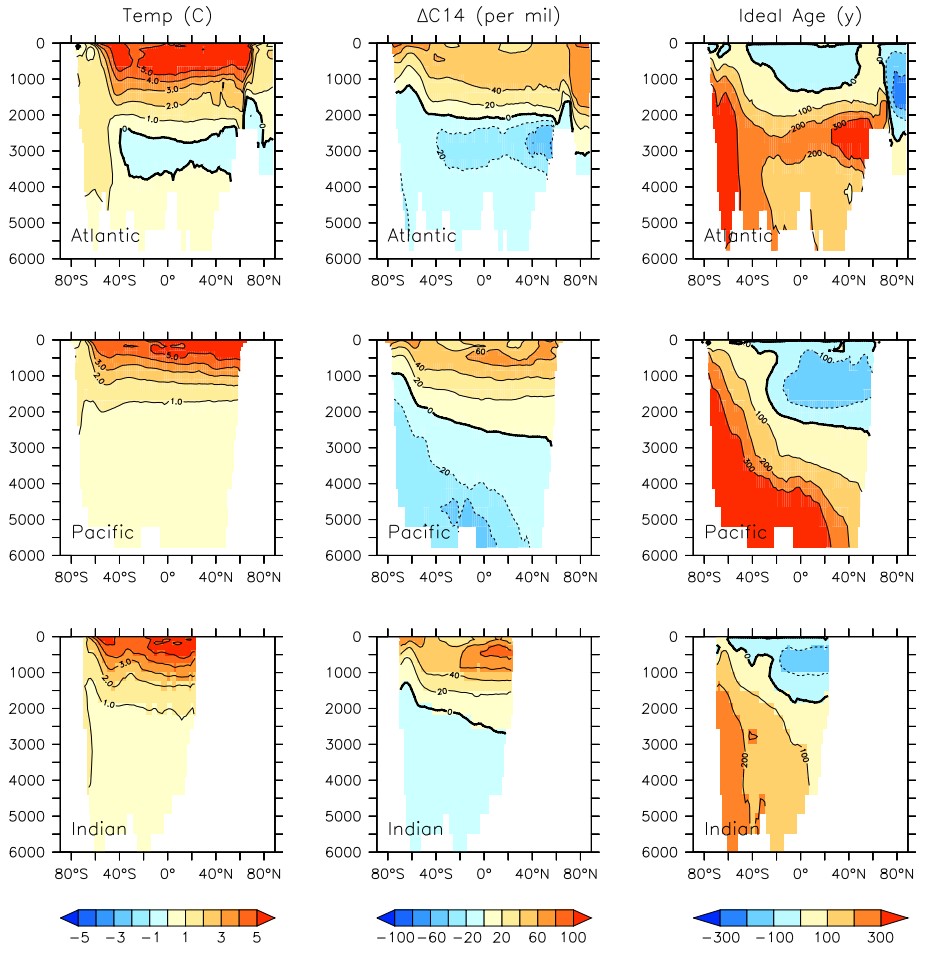

**Figure 26.** Modelled change in temperature, radiocarbon, and ideal age profiles by major ocean basin, from 2014 to 2296.

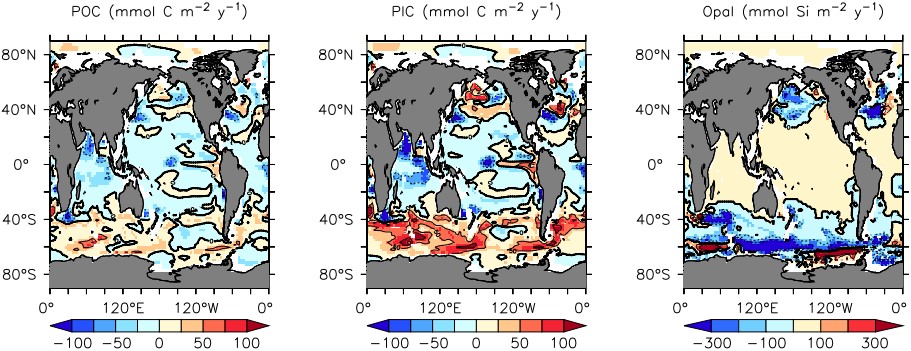

**Figure 27.** Modelled change in particle fluxes at 2 km depth, from 2014 to 2294.

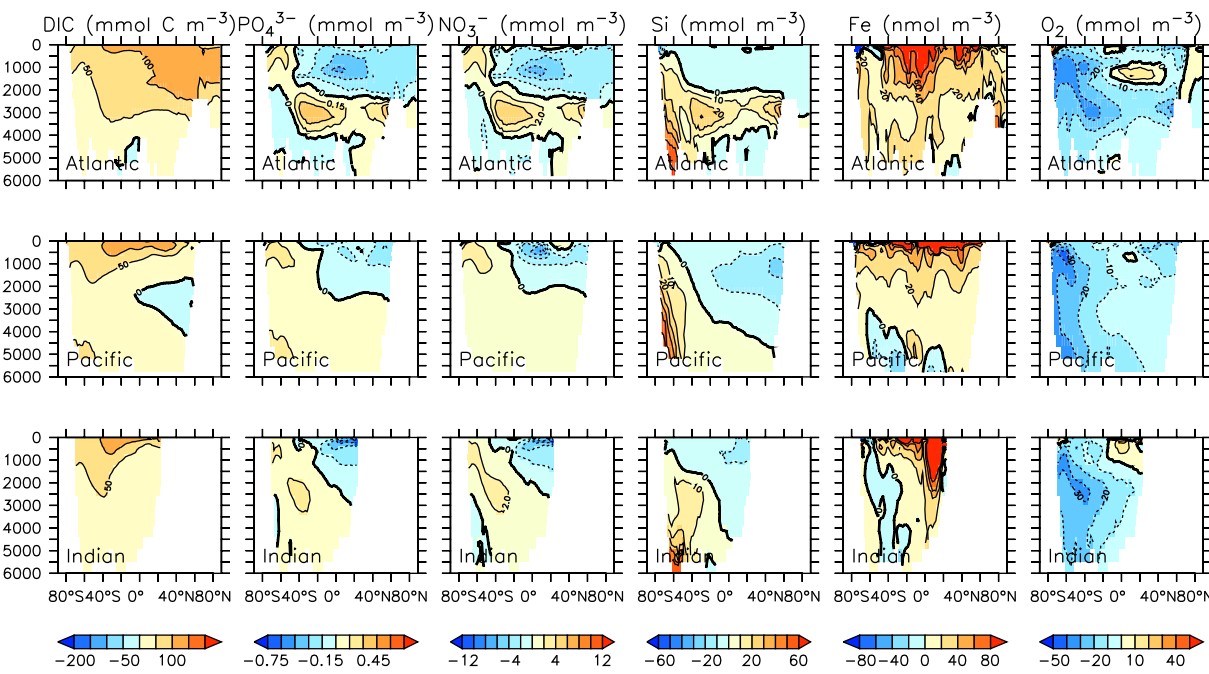

**Figure 28.** Modelled change in carbon and nutrient profiles by major ocean basin, from 2014 to 2294.

**Table 3.** KMBM3 phytoplankton production and mortality parameters.

| Parameter | Symbol | Units | Value |
|---|---|---|---|
| Growth rate | $a_{LP}$ | day$^{-1}$ | 0.4 |
| | $a_{CP}$ | | 0.6 |
| | $a_{DT}$ | | 0.7 |
| | $a_{DZ}$ | | $0.81 \times a_{LP}$ |
| Half-saturation constant N | $k_{\mathrm{N}LP}$ | mmol m$^{-3}$ | 0.1 |
| | $k_{\mathrm{N}CP}$ | | 0.2 |
| | $k_{\mathrm{N}DT}$ | | 0.5 |
| Half-saturation constant P | $k_{\mathrm{P}DZ}$ | mmol m$^{-3}$ | $k_{\mathrm{N}LP}/16$ |
| Half-saturation constant Fe | $k_{\mathrm{Fe}LP}$ | nmol m$^{-3}$ | 0.08 |
| | $k_{\mathrm{Fe}CP}$ | | 0.34 |
| | $k_{\mathrm{Fe}DT}$ | | 0.78 |
| | $k_{\mathrm{Fe}DZ}$ | | 0.08 |
| Half-saturation constant Si | $k_{\mathrm{Si}}$ | mmol m$^{-3}$ | variable |
| Half-saturation Si scaling | $k_{\mathrm{Si}}^{*}$ | mol Si m$^{-3}$ | 0.03 |
| Initial slope of P-I curve | $\alpha_{\min}^{\mathrm{chl}}$ | (W m$^{-2}$)$^{-1}$ d$^{-1}$ | 1.59 |
| | $\alpha_{\max}^{\mathrm{chl}}$ | | 6.36 |
| Chl:C ratio | $\theta_{\min}$ | unitless | 0.01 |
| | $\theta_{\max}$ | unitless | 0.04 |
| Phytoplankton mortality rate | $m_{LP}$ | day$^{-1}$ | 0.03 |
| | $m_{CP}$ | | 0.03 |
| | $m_{DT}$ | | 0.03 |
| Microbial fast recycling | $\mu_{0LP}^{*}$ | day$^{-1}$ | 0.015 |
| | $\mu_{0CP}^{*}$ | | 0.015 |
| | $\mu_{0DT}^{*}$ | | 0.015 |
| | $\mu_{0DZ}^{*}$ | | 0.015 |

**Table 4.** KMBM3 zooplankton parameters. Temperature-dependent parameter values are given for $0\,^\circ\text{C}$.

| Parameter | Symbol | Units | Value |
|---|---|---|---|
| Maximum grazing rate | $\mu_Z^{\theta}$ | $\text{day}^{-1}$ | 0.571 |
| Maximum grazing rate parameters | $b$ | unitless | 1.066 |
| | $c$ | $^\circ\text{C}^{-1}$ | 1.0 |
| Food preferences | $\psi_{LP}$ | unitless | 0.2 |
| | $\psi_{CP}$ | | 0.25 |
| | $\psi_{Z}$ | | 0.2 |
| | $\psi_{DZ}$ | | 0.05 |
| | $\psi_{DT}$ | | 0.25 |
| | $\psi_{Det}$ | | 0.05 |
| Half saturation constant | $k_z$ | $\text{mmol m}^{-3}$ | 0.15 |
| Growth efficiency constant | $\varpi$ | unitless | 0.4 |
| Food assimilation efficiency | $\gamma$ | unitless | 0.7 |
| Mortality rate | $m_z$ | $\text{day}^{-1}$ | 0.06 |

**Table 5.** KMBM3 particle export-production parameters.

| Parameter | Symbol | Units | Value |
|---|---|---|---|
| Detrital remineralisation rate | $\mu_{D,0}$ | $\text{day}^{-1}$ | 0.07 |
| Detrital sinking speed at surface | $w_{D,0}$ | $\text{m day}^{-1}$ | 12.28 |
| Detrital increase in sinking speed | $wdd$ | $\text{m day}^{-1}$ | 5.8 |
| $CaCO_3$ ballast:total detrital production ratio | $R_{\text{bal:tot}}$ | unitless | 0.05 |
| $CaCO_3$:POC production ratio | $R_{CaCO_3:POC}$ | unitless | 0.07 |
| $CaCO_3$ sinking speed at surface | $w_{C,0}$ | $\text{m day}^{-1}$ | 22.43 |
| $CaCO_3$ increase in sinking speed | $wdc$ | $\text{m day}^{-1}$ | 1.8 |
| Base opal:POC production ratio | $R_{\text{Opal:POC},0}$ | unitless | 0.5 |
| Opal dissolution rate constant | $\lambda_{\text{Opal}}$ | $\text{day}^{-1}$ | 0.03 |
| Opal sinking rate | $w_{\text{Opal}}$ | $\text{m day}^{-1}$ | 75.0 |
| Organic particle iron scavenging rate | $kFe_{\text{org}}$ | $(\text{m}^3(\text{gC d})^{-1})^{0.58}$ | 0.45 |
| Calcite iron scavenging rate | $kFe_{\text{ca}}$ | $(\text{m}^3(\text{gCaCO}_3\,\text{d})^{-1})^{0.58}$ | 0.45 |

**Table 6.** Globally integrated diagnosed biogeochemical properties at year 2004. Corresponding values are also given using KMBM1 (Keller et al., 2012).

| Property | KMBM3 | KMBM1 | Independent Estimate |
|---|---|---|---|
| Primary Production (Pg C y$^{-1}$) | 53.60 | 54.33 | 44–78[a] |
| Calcite Production (Pg C y$^{-1}$) | 0.87 | 0.86 | 1.08–1.60[b] |
| Opal Production (Tmol Si y$^{-1}$) | 134.0 | - | 255±52[c] |
| Nitrogen Fixation (Pg N y$^{-1}$) | 0.20 | 0.16 | 0.71–1.54[d] |
| POC flux at 130 m (Pg C y$^{-1}$) | 8.57 | 7.04 | 5.73[e] |
| POC flux at 2 km (Pg C y$^{-1}$) | 0.28 | 0.36 | 0.43 ± 0.05 |
| CaCO$_3$ flux at 130 m (Pg C y$^{-1}$) | 0.72 | 0.84 | 1.1 ± 0.3 |
| CaCO$_3$ flux at 2 km (Pg C y$^{-1}$) | 0.37 | 0.49 | 0.41 ± 0.05 |
| CaCO$_3$ dissolution (Pg C y$^{-1}$) | 0.40 | - | 0.5 ± 0.2[f] |
| Opal flux at 130 m (Tmol Si y$^{-1}$) | 122.6 | - | 112[c] |
| Opal flux at 2 km (Tmol Si y$^{-1}$) | 41.08 | - | 84[c] |
| Opal water column total dissolution (Tmol Si y$^{-1}$) | 127.5 | - | 171[c] |
| Opal water column dissolution 0 - 130 m (Tmol Si y$^{-1}$) | 10.30 | | |
| Opal water column dissolution 130 m - 2 km (Tmol Si y$^{-1}$) | 68.81 | | |
| Opal water column dissolution below 2 km (Tmol Si y$^{-1}$) | 48.39 | | |
| Opal total seafloor flux (Tmol Si y$^{-1}$) | 6.53 | - | 9.2 ± 1.6[c] |
| Opal seafloor flux 0 - 130 m (Tmol Si y$^{-1}$) | 1.11 | | |
| Opal seafloor flux 130 m - 2 km (Tmol Si y$^{-1}$) | 4.08 | | |
| Opal seafloor flux below 2 km (Tmol Si y$^{-1}$) | 1.34 | | |
| Silica river input (Tmol Si y$^{-1}$) | 2.38 | - | 8.1±2.0[c] |
| Total Phytoplankton (Pg C) | 0.55 | 0.53 | 0.5–2.4[g] |
| Phytoplankton LP (Pg C) | 0.03 | 0.51 | 0.28–0.64[h] |
| Phytoplankton CP (Pg C) | 0.05 | - | 0.001–0.03 |
| Phytoplankton DT (Pg C) | 0.02 | - | 0.1–0.94 |
| Phytoplankton DZ (Pg C) | 0.45 | 0.02 | 0.008–0.12 |
| Zooplankton (Pg C) | 0.40 | 0.52 | 0.03–0.67[i] |

[a]Carr et al. (2006), Jin et al. (2006)

[b]Smith and Gattuso (2011), Smith and Mackenzie (2016)

[c]Tréguer et al. (2021)

[d]Luo et al. (2014)

[e]all particle fluxes from Honjo et al. (2008) unless noted

[f]Luo et al. (2014)

[g]all biomass estimates from Buitenhuis et al. (2013)

[h]picophytoplankton

[i]pteropods