# Peer review of "Explicit silicate cycling in the Kiel Marine Biogeochemistry Model, version 3 (KMBM3) embedded in the UVic ESCM version 2.9"

_Geoscientific Model Development, 2020_

## Referee Comment (RC1) · M. Baird (Referee) · 6 Oct 2020

This paper describes a set of new processes in the KMBM3 model embedded in the UVIC Earth System model. I must apologise in advance that I have concentrated my review on the model presentation due to my time constraints and where my comments are likely to be most helpful.

The biogeochemical model, and the silica components in particular, are interesting, and simulations suggest they are an improvement on earlier versions. The interesting components of the silica model would be better presented if the model description was more cleanly separated from the model configuration, so that people wishing to apply

your silica equation could more easily work off your paper. For example, lines 254 - 258 and 270 - 278 should be separated out into a new subsection, titled something like 'Silica inputs in the UVic model configuration '. Look for other instances.

Comments for improved clarity / rigor:

1. The manuscript would benefit from a table of state variables, which could also define the many subscripts used in the manuscript.

2. The use of term 'mass conservation' in the model might be unclear to readers. Please distinguish between the model equations, and the model domain, conserving mass. An input or export of silica to the model domain should not be confused with a failure to account for a term in the equations. Do all case of non-conservative behaviour relate to inputs and exports?

3. On point 2, line 79, why do you balance export to the sediments with inputs to the surface ocean? Isn't the point that the pool of oceanic silica is changing over time. The artificial nature of this assumption is more limiting than any benefit in domain wide mass conservation and potentially obscures problems with the formulation.

4. Line 98. Description is loose. The symbols m, JX and  $\mu^{\star}$  are a rate coefficients,  $\mu^{\star}$  X is the term.

5. The use of T in Eq. 3 and elsewhere is awkward because it relies on use the Celsius scale. If you swapped to Kelvin,  $20/15 \neq (20 + 273)/(15 + 273)$ . Looking at Eq. 3, 'a' is a growth rate parameter (not a maximum growth rate as described in Table 2). Infact, it is not even an exponential growth rate parameter, since the exponential component is in the term e(T/Tb).

6. Eq. 4. It would be preferable that you use Fe for the Chemical Symbol, and [Fe] for the concentration of iron. Also for other elements. Reasons are highlighted in later comments.

7. Eq. 8 looks odd but behaves okay. In any case, there are three constants in the

GMDD
equation which should be parameters.

8. Line 144. Mortality from 'old-age' is a misleading description since you do not track age distribution of the population. How about simply non-grazing mortality.

9. Interesting that you have self-grazing in the zooplankton!

10. Line 157. Should state that the sum of the food preference parameters must be 1. This would make it clear that it is a 'relative food preference'. While this grazing form meets a local mass conservation criteria, it is, nonetheless, awkward.

For example, if you had one phytoplankton species, its preference would be 1, and G1 would be mu Z X / (X + k). If you split this into two identical classes with 0.5 preference, then G12 = mu Z [0.5 X/2 / (0.5 X/2 + 0.5 X/2 + k) + 0.5 X/2 / (0.5 X/2 + 0.5 X/2 + k) ] = mu Z X/2 / [X/2 + k]  $\neq$  mu Z X / (X + k). To test this, try running your model with two identical LP cases, starting with half the concentration. Thus it is not only relative preference, but also specific to the predators and prey you have in this configuration.

11. Eq. 20, 39 Brackets around max function are different than Eq. 15 and others.

12. L163 – what does 'sox' mean? I suspect it should be ro2sox. Think about how you have used subscripts to identify cases (i.e. phytoplankton type, element type) and superscripts to define its application (i.e. max). Eqs. 10 and 11 also appear to have the super- and sub-scripts the wrong way around.

13. Eq 30. Rethink subscript. If 0 is meant to identify an example of a ratio of Opal to POC, then it should be written as ROpal:POC,0. Lots of places subscripts would benefit from commas.

14. For example CaCO3liv would be better as [CaCO3]liv as proposed above. Eq. 25, Feorgads also mixes chemical symbolism with mathematical notation.

15. I don't understand Eq. 32. If it is a local rate of change why does it have units of m-1. [Reading later it looks like it is a benthic term – needs clarification].
16. There is inconsistent us of the (x) symbol. Equations in latex format work better without them. Only use where it improves clarity.

17. Instances of SCaCO3 should be S(CaCO3) (Eq. 29, 41, 42 etc.)

18. Eq. 43 is simplistic as the units don't work. I presume Pr(Opal) is a 3D flux, I am not sure what Di(Opal) is (see comment 15), and Si(Dust) is probably a 2D flux, and Siriv is probably multiple point sources. But Si in equation 8 and elsewhere is a concentration. Another reason to represent concentration of an element as [Si]. Eq. 40 has similar problems.

19. Line 230. Why does sinking increase with depth? I would have thought the increasing density with depth would result in a reduced negative buoyancy, and dissolution would reduce particle size slowing the Stokes' sinking rate? Coagulation might go the other way. Discuss more.

20. L266. M-M uptake rate? I prefer the line 102 description of iron availability. Perhaps a table of 'derived variables' with definitions of mathematical symbols that are in the equations, but not state variables or parameters would help in tightening up these descriptions.

21. Eq. 36. bsi would be better as bSi.

22. Eq 36 Exp is potentially confusing with the exponential function.

23. Eq. 21. Would it be better to use oxygen concentration of % saturation. I am not sure which it is that affects animal metabolism.

Other minor comments.

Line 13 'migrate' - poor word choice. Perhaps 'bloom further south'

Line 94 - what does a virtual flux mean in this context.

L276 replace 'before the biology' with 'faster than the biological processes'

GMDD

---

## Referee Comment (RC2) · Anonymous Referee #2 · 15 Oct 2020

In a first part, the paper describes the implementation of Silicon sub-model in the UVIC Earth System model. The model skills are first evaluated against a series of global scale observations. Then changes in ocean biogeochemistry are presented for two temporal windows: over the historical period and in the future until year 2300. I should admit that I have mixed feelings about that paper. On the one hand, I think it is useful because it describes a new version of the biogeochemical module that is embedded in the UVic ESM. The model is relatively well evaluated against a diverse collection of observations. And some interesting climate change experiments are presented. On the other hand, I don't like very much this paper for several reasons that I will try to present now.

[Figure]

The first problem I have is that I don't really see anything new in the new submodule that is presented in this study. Most of the parameterisations used to describe diatoms and the silicon cycle have been published elsewhere. Some parameterisations are also questionable. For instance, phytoplankton maximum growth rate is scaled by a Fe limitation term and is then multiplied by the minimum of the other limiting factors (N, P, Si, and light). Why is that parameterization chosen? It is a mixture of a multiplicative formulation and the Liebig's law (law of the minimum). Furthermore, the temperature dependency of biogenic silica dissolution has the same sensitivity as the remineralization of organic matter. The reason that is given in the paper is an organic coating that needs to be degraded before dissolution starts. But, their formulation implies that once the organic coating has been degraded, dissolution of opal is instantaneous and temperature insensitive. Otherwise, the temperature dependency would be more complex. Obviously, this is not the case in the real world. Thus, this formulation would deserve some more explanation and justification.

The second problem is that the model performance is not as good as what the authors state in the text. First, the simulated primary production suggests a huge production in the equatorial Pacific and in the northern Indian Ocean, much larger than what is estimated from satellite observations. This seems to be due to the DZ compartment and maybe the LP compartment (they don't show a spatial map of the LP distribution). It would have been interesting (necessary) to have a map of the simulated chlorophyll distribution and a comparison to ocean color products such as GLOBCOLOUR or OCCI. Biases on satellite-derived Chl are much smaller than the uncertainties on satellite-based NPP. I suspect that simulated Chl levels are certainly way too high in the equatorial Pacific and northern Indian Ocean. Furthermore, the DZ distribution looks strange and not what we would expect from observations. They are maximum in the (macro-)nutrient rich areas (right along the equator) whereas due to their competitive advantage at low N levels given by their ability to fix N2, there are traditionally believed to be successfull in the subtropical gyres (providing that enough P and Fe are available). In fact, that's what shows the MAREDAT compilation presented in Fig.

4. In that compilation, the zonally averaged DZ distribution exhibits a minimum at the equator in strong contrast with what is simulated by the model. An additional validation that would have been interesting is a comparison with a satellite-based distribution of chlorophyll by size or by main groups (Hirata et al., 2011 ; PHYSAT ; Brewin et al, 2010 ; ...). These products have strong uncertainties but prove to be useful to qualitatively evaluate the model skills. Finally Si concentrations at the surface are too high in the low latitudes and too low in the high latitudes, especially in the subarctic Pacific Ocean and in the Southern Ocean. This suggests significant biases either in the DT distribution and/or in the opal export/dissolution in the upper ocean.

My third concern is about the third part of the paper in which the authors discuss the response of their model to climate change. This part is rather interesting but is frustrating because the authors don't really analyse the processes that explain their results. They find some interesting features, such as a decline of DT during the historial period in the high latitudes (for instance in the Southern Ocean) followed than by a strong increase in the future. Why? What are the processes that explain this behavior? Same for NPP, which at the global scale decreases strongly until the mid of the 22nd century to then increase until the end of their simulation. They propose some explanations: a change in the community composition (more calcifiers, less diatoms) and a shallower recycling of POC due to a stronger temperature. It would have been interesting to have a more detailed analysis that evaluates the respective weight of these processes over time and that would explain the change of the trend during the 22nd century.

My recommandation is to strongly modify the paper to make it more interesting and convincing. There are certainly the materials to make a very interesting paper, which is, in my humble opinion, not the case currently.

Some more specific comments: Eq. 3: the maximum growth rate is multiplied by an iron limitation term and then by the minimum of the other nutrients and light. Thus, this is a mix of multiplicative formulation and a law of the minimum. This is quite unusual. It should be explained and justified.

[Figure]

Eq. 11: If I understand correctly, the Chl/C ratio only varies with Fe limitation. Why only iron and not also light and the other nutrients?

Eq. 17: The fast remineralization term would deserve some more explanation.

Eq. 18: I don't understand why the authors use a quadratic term to model senescence/old age for zooplanktan and a linear term for phytoplankton.

Eq. 20: the maximum gazing rate is capped at temperature above 20°C. Why?

Eq. 32: This parameterization is not very well justified. It would be nice to have a more detailed explanation. The authors invoke the degradation of the organic coating to justify the temperature dependency. However, the dissolution of opal has also been shown to be temperature sensitive (e.g., Kamatani, 1982) with a sensitivity to temperature higher than what has been shown for POC degradation. Using only one temperature sensitivity, i.e. that of POC, is equivalent to assuming that BSi dissolution is instantaneous (and temperature insensitive).

Line 287: I would not use the word deposition for river discharge.

Lines 396-397: Thus a low Si bias is explained by a low DT bias and too low export of opal. I don't understand that explanation. A too weak Si consumption by DT and a too small export of BSi should lead to too elevated surface Si concentrations. Did I miss something?

Lines 401-402: From what I see on the figure, diatoms do not really precede calcifiers, at least in the SOuthern Ocean. They rather grow south of the CP compartment. Similarly, LP and DZ do not grow at the same place: DZ are growing in the equatorial domain whereas LP are growing more successfully in the subtropical domain.

Lines 426-427: A reference would be nice here to support that statement.

Figures: Many figures could be improved by changing the colorscale and/or using a different range for the values. I would suggest to redraw Figures 2, 3, 4, 5, 9 (especially

for the surface), 10, 12, 13.

---

## Author Comment (AC2) · 2 Jan 2021

In a first part, the paper describes the implementation of Silicon sub-model in the UVIC Earth System model. The model skills are first evaluated against a series of global scale observations. Then changes in ocean biogeochemistry are presented for two temporal windows: over the historical period and in the future until year 2300. I should admit that I have mixed feelings about that paper. On the one hand, I think it is useful because it describes a new version of the biogeochemical module that is embedded in the UVic ESM. The model is relatively well evaluated against a diverse collection of observations. And some interesting climate change experiments are presented. On the other hand, I don't like very much this paper for several reasons that I will try to present now.

The authors thank the Reviewer for their careful analysis of our manuscript, which we have used to improve its quality. Please find the Review reproduced below in blue font, with our responses in black font. Changes made to the manuscript are shown in red font.

The first problem I have is that I don't really see anything new in the new submodule that is presented in this study. Most of the parameterisations used to describe diatoms and the silicon cycle have been published elsewhere.

This is correct. All but one parameterisation has been published elsewhere. We make no claim to novelty on these aspects, and cite the primary sources. However, the combination of these parameterisations in the UVic ESCM model is novel and it is therefore worthwhile to document how the parameterisations were implemented and how they affect model performance. Also, any state-of-the-art model used in any kind of climate research should be described in sufficient detail in a peer reviewed journal; GMD and JAMES are journals particularly designed for this purpose.

Some parameterisations are also questionable. For instance, phytoplankton maximum growth rate is scaled by a Fe limitation term and is then multiplied by the minimum of the other limiting factors (N, P, Si, and light). Why is that parameterization chosen? It is a mixture of a multiplicative formulation and the Liebig's law (law of the minimum).

This parameterisation has been part of the core KMBM (biogeochemical model) since Keller et al. (2012). It was adapted from Galbraith et al. (2010):

Galbraith, E. D., Gnanadesikan, A., Dunne, J. P., and Hiscock, M. R.: Regional impacts of iron-light colimitation in a global biogeochemical model, Biogeosciences, 7, 1043–1064, doi:10.5194/bg-7-1043-2010, 2010.

This parameterisation is used because it requires sufficient iron to be available first, for the phytoplankton to use macronutrients. A sentence is added to the text with the reference (P4, L 103):

This parameterisation assumes sufficient iron is required for the utilisation of other nutrients (Galbraith et al., 2010; Keller et al., 2012; Nickelsen et al., 2015).

Furthermore, the temperature dependency of biogenic silica dissolution has the same sensitivity as the remineralization of organic matter. The reason that is given in the paper is an organic coating that needs to be degraded before dissolution starts. But, their

formulation implies that once the organic coating has been degraded, dissolution of opal is instantaneous and temperature insensitive. Otherwise, the temperature dependency would be more complex. Obviously, this is not the case in the real world. Thus, this formulation would deserve some more explanation and justification.

This equation (32) was incorrectly presented in the manuscript, and we apologise for this and any resulting confusion. The model does not explicitly consider small scale dissolution dynamics (kinetics), including organic coatings. Dissolution is represented as an instantaneous distribution down the water column, with a steeper gradient in warmer temperatures. The equation has now been fixed, with additional explanation (also requested from the first Reviewer; P9, L225):

This parameterisation results in greater dissolution at warm temperatures and is similar to the instant-sinking-and-dissolution function applied to model calcite (Schmittner et al., 2008) (although, the function for calcite was replaced when a prognostic tracer was added by Kvale et al., 2015b).

The second problem is that the model performance is not as good as what the authors state in the text. First, the simulated primary production suggests a huge production in the equatorial Pacific and in the northern Indian Ocean, much larger than what is estimated from satellite observations. This seems to be due to the DZ compartment and maybe the LP compartment (they don't show a spatial map of the LP distribution). It would have been interesting (necessary) to have a map of the simulated chlorophyll distribution and a comparison to ocean color products such as GLOBCOLOUR or OCCI. Biases on satellite-derived Chl are much smaller than the uncertainties on satellite-based NPP. I suspect that simulated Chl levels are certainly way too high in the equatorial Pacific and northern Indian Ocean. Furthermore, the DZ distribution looks strange and not what we would expect from observations. They are maximum in the (macro-)nutrient rich areas (right along the equator) whereas due to their competitive advantage at low N levels given by their ability to fix N2, there are traditionally believed to be successfull in the subtropical gyres (providing that enough P and Fe are available). In fact, that's what shows the MAREDAT compilation presented in Fig. 4. In that compilation, the zonally averaged DZ distribution exhibits a minimum at the equator in strong contrast with what is simulated by the model. An additional validation that would have been interesting is a comparison with a satellite-based distribution of chlorophyll by size or by main groups (Hirata et al., 2011 ; PHYSAT ; Brewin et al, 2010 ; ...). These products have strong uncertainties but prove to be useful to qualitatively evaluate the model skills.

Section 3 assesses model performance against a range of observations as well as previous model versions and we discuss model biases, as well as possible remedies, at length. There is a historical regional bias in UVic ESCM NPP (which can be seen in Fig 2, where we compare our new model to a previous version). We demonstrate some model improvement with respect to the problematic regions mentioned above. Further improvement is something that will require a model calibration framework (and a separate study, see comment below with respect to process study).

Our model does not simulate chlorophyll- we apologise if this was unclear from the text. A new Table 1 has been added that lists the model state variables. The assumptions that would be required to diagnose chlorophyll from the model phytoplankton biomass introduce enough uncertainty that reduced error in the satellite chlorophyll observations

don't really aid comparison, especially since the large biases in modelled NPP are plainly visible in Figure 2.

In Section 3 we both show and discuss diazotroph distributions and their impact on nitrogen fixation (which is under-estimated, despite over-estimated diazotroph biomass). We have added more clarification:
(P13, L 353)

Thus in our modelling context, this phytoplankton type can be considered "slow-growing phytoplankton capable of fixing nitrogen when necessary". Constraints on this phytoplankton type will be explored in the future.

 We have also added LP biomass to Figures 4 and 15.

Direct comparison of modelled phytoplankton types to real ocean phytoplankton is always problematic because of the oversimplification we must use in our models to represent the real world. Given the large biases already shown compared to the MAREDAT data, additional model validation against other phytoplankton datasets would not provide substantial new insight. However, we thank the Reviewer for pointing out these additional datasets, which will be very useful in future model parameter calibrations that will use observational datasets to constrain model biomass.

Finally Si concentrations at the surface are too high in the low latitudes and too low in the high latitudes, especially in the subarctic Pacific Ocean and in the Southern Ocean. This suggests significant biases either in the DT distribution and/or in the opal export/dissolution in the upper ocean.

We assess the silica concentration biases in Section 3 and also discuss the possible reasons for them, including biases in DT distribution and export fluxes. Biases arise due to both deficiencies in model physics as well as biogeochemical model structure. Whether the Reviewer finds them acceptable is a value judgment (how good is good enough and for what purpose), but tuning such a complex model is difficult and can require multiple human years and special tools.  Attempts to reduce bias in the model are ongoing, and constitute the major activity that helps us as biogeochemical modelers learn about the inner workings of the ocean/climate system. We agree there is room for improvement! As we state in the paper, this assessment is meant to serve as a baseline for future model improvements.

My third concern is about the third part of the paper in which the authors discuss the response of their model to climate change. This part is rather interesting but is frustrating because the authors don't really analyse the processes that explain their results. They find some interesting features, such as a decline of DT during the historial period in the high latitudes (for instance in the Southern Ocean) followed than by a strong increase in the future. Why? What are the processes that explain this behavior?

Warming and stratification are the two dominant mechanisms controlling community trends, and are discussed in the second paragraph of section 4.1. We now make the processes more explicit earlier in the text:
(P15, L440)

…due to increasing thermal stratification… due to their high nutrient requirements…

Actually, the mechanisms of long-term trends are discussed (from the bottom of page 18), including the reversal in NPP (the microbial loop, see P19, L555).

Same for NPP, which at the global scale decreases strongly until the mid of the 22nd century to then increase until the end of their simulation. They propose some explanations: a change in the community composition (more calcifiers, less diatoms) and a shallower recycling of POC due to a stronger temperature. It would have been interesting to have a more detailed analysis that evaluates the respective weight of these processes over time and that would explain the change of the trend during the 22nd century.

As a model description paper, this (already very long) manuscript is meant to be descriptive. We prefer to leave such a detailed process analysis to future model studies with a more "scientific" focus, i.e. parameter optimisation studies (e.g., Yao et al.,2019). The reason for this is because (as is shown in Yao et al., 2019, and in his forthcoming manuscript that is currently in review) parameter optimisation can strongly affect the relative weighting of these various processes and nutrient pathways, with significant impacts on model behaviour with transient forcing. Its important to be extremely careful with what one concludes with process analysis, and it must be given proper context- a single simulation is insufficient and potentially misleading. We heartily agree that detailed analyses of the relative importance of different processes are interesting, and promise that they are coming, but will be given full and careful treatment as a separate publication. Such analyses also require decomposition techniques (e.g. Koeve et al., 2020, GRL) not yet implemented to this UVic version.

My recommandation is to strongly modify the paper to make it more interesting and convincing. There are certainly the materials to make a very interesting paper, which is, in my humble opinion, not the case currently.

Some more specific comments: Eq. 3: the maximum growth rate is multiplied by an iron limitation term and then by the minimum of the other nutrients and light. Thus, this is a mix of multiplicative formulation and a law of the minimum. This is quite unusual. It should be explained and justified.

Please see the explanation given above.

Eq. 11: If I understand correctly, the Chl/C ratio only varies with Fe limitation. Why only iron and not also light and the other nutrients?

Chl is not simulated. This is a one-off gross calculation used for the iron model to link iron availability to light affinity. It is kept for historical reasons (it was introduced by Nickelsen et al., 2015). Flexible ratios are not introduced to other aspects of the model because the focus of this manuscript is silica cycling. Other researchers (Markus Pahlow, Chia-Te Chien) are currently working on this aspect of the KMBM (e.g., Chien et al. (2020), GMD, doi.org/10.5194/gmd-13-4691-2020).

Eq. 17: The fast remineralization term would deserve some more explanation.

This parameterisation is a long-standing feature of the KMBM/UVic ESCM. A citation is now provided as well as a new sentence (P6, L151):

With this formulation, increasing seawater temperature increases respiration and the return of nutrients to the upper ocean.

Eq. 18: I don't understand why the authors use a quadratic term to model senescence/old age for zooplanktan and a linear term for phytoplankton.

This parameterisation is a part of the model since Keller et al. (2012). In the earlier biogeochemical model of Schmittner et al. (2008), phytoplankton mortality was quadratic. This was changed when the microbial loop (non-linear loss term) was added by Keller et al. (2012). A citation is now provided.

Eq. 20: the maximum gazing rate is capped at temperature above 20◦C. Why?
This parameterisation is a part of the model since Keller et al. (2012). This is done to prevent unrealistically high grazing rates in the tropics, and is a common feature of earth system models (Anderson, T. R., Gentleman, W. C., and Sinha, B.: Influence of grazing formulations on the emergent properties of a complex ecosystem model in a global ocean general circulation model, Prog. In Ocean.)
A citation is now provided.

Eq. 32: This parameterization is not very well justified. It would be nice to have a more detailed explanation. The authors invoke the degradation of the organic coating to justify the temperature dependency. However, the dissolution of opal has also been shown to be temperature sensitive (e.g., Kamatani, 1982) with a sensitivity to temperature higher than what has been shown for POC degradation. Using only one temperature sensitivity, i.e. that of POC, is equivalent to assuming that BSi dissolution is instantaneous (and temperature insensitive).
Please see our earlier comment, as well as the changes we made to the text. We tried a number of other previously published parameterisations, discussed in the text, that produced dSi profiles with far worse agreement to observations. We can tune the opal dissolution and sinking rate constants to be more or less the detrital remineralisation and sinking rates. Our chosen parameterisation is also consistent with the increase in organic particle sinking speed with depth (stronger dissolution in the warm upper ocean, where particles sink more slowly, compared to the cold deep ocean, where particle sinking rate is much faster). While opal dissolution and organic detritus remineralisation are not explicitly linked via kinetics in this first model version of silicate, it is useful to maintain some process consistency.

Line 287: I would not use the word deposition for river discharge.
The word is replaced

Lines 396-397: Thus a low Si bias is explained by a low DT bias and too low export of opal. I don't understand that explanation. A too weak Si consumption by DT and a too small export of BSi should lead to too elevated surface Si concentrations. Did I miss something?
The Reviewer is right, low consumption of Si can lead to more dSi in the surface water column. That dSi is probably mostly transported away via physical advection and mixing processes. We find that in our model, diatoms are very good at "trapping" dSi in the regions they inhabit, similar to dFe (which is also observed in the real ocean, e.g. Boyd et al. 2017). However, it might be that there are deficiencies in the regional marginal sea circulation that are producing the bias (Nishioka et al., 2020, PNAS), so the sentence is revised:
(P14, L416)
A low bias is also simulated in the surface North Pacific, which possibly suggests deficiencies in the circulation within and between regional marginal seas (Nishioka et al., 2020).

Boyd, P., Ellwood, M., Tagliabue, A. *et al.* Biotic and abiotic retention, recycling and remineralization of metals in the ocean. *Nature Geosci* **10,** 167–173 (2017). https://doi.org/10.1038/ngeo2876

Lines 401-402: From what I see on the figure, diatoms do not really precede calcifiers, at least in the SOuthern Ocean. They rather grow south of the CP compartment. Similarly, LP and DZ do not grow at the same place: DZ are growing in the equatorial domain whereas LP are growing more successfully in the subtropical domain.
We have clarified the wording:
(P15, L423)
(albeit, in separate zonal ranges, Figure 13).

Lines 426-427: A reference would be nice here to support that statement.
This is a result of our model and this is clarified (P15, L448). Differences in SO phytoplankton trends in our model compared to others are discussed at the bottom of the next paragraph.

Figures: Many figures could be improved by changing the colorscale and/or using a different range for the values. I would suggest to redraw Figures 2, 3, 4, 5, 9 (especially for the surface), 10, 12, 13.
 The figures have been re-made with a different colorscale.

The authors would like to again thank the Reviewer for considering our manuscript.

---

## Author Comment (AC1)

M. Baird (Referee)
mark.baird@csiro.au

This paper describes a set of new processes in the KMBM3 model embedded in the UVIC Earth System model. I must apologise in advance that I have concentrated my review on the model presentation due to my time constraints and where my comments are likely to be most helpful.
The biogeochemical model, and the silica components in particular, are interesting, and simulations suggest they are an improvement on earlier versions. The interesting components of the silica model would be better presented if the model description was more cleanly separated from the model configuration, so that people wishing to apply your silica equation could more easily work off your paper. For example, lines 254 – 258 and 270 – 278 should be separated out into a new subsection, titled something like 'Silica inputs in the UVic model configuration '. Look for other instances.

We would like to thank the Reviewer for this suggestion. Almost all of these equations are published elsewhere and are not new. We have extended existing equations to diatoms/silica, where appropriate, and adopted the parameterisations of others. Anyone wishing to do something similar in a different base model should base this on the primary references, which we cite when introduced. We appreciate the point that the explanation of all terms in each equation at the point of introduction lengthens each section, and that some terms could be explained in separate sections to concentrate the maths. This is a stylistic preference that would necessitate more flipping forwards and back in the manuscript without a clear improvement in clarity. However, we have separated out the boundary terms in Section 2.2.9 (e.g. new equation 45) since we assume that this is what motivates the Reviewer's comment.

Comments for improved clarity / rigor:
1. The manuscript would benefit from a table of state variables, which could also define the many subscripts used in the manuscript.
A new table has been added (new Table 1).

2. The use of term 'mass conservation' in the model might be unclear to readers. Please distinguish between the model equations, and the model domain, conserving mass. An input or export of silica to the model domain should not be confused with a failure to account for a term in the equations. Do all case of non-conservative behaviour relate to inputs and exports?
Both the model equations and model domain conserve mass, unless explicitly stated. This is a climate model that needs to be stable for millennial simulations. We have reconsidered all uses to make sure the meaning of "conservation" is clear, and revised where necessary.

3. On point 2, line 79, why do you balance export to the sediments with inputs to the surface ocean? Isn't the point that the pool of oceanic silica is changing over time. The artificial nature of this assumption is more limiting than any benefit in domain wide mass conservation and potentially obscures problems with the formulation.

While we agree with the Reviewer that the pool of oceanic nutrients changes over time, any earth system model needs to balance overall fluxes in and out of the model domain during equilibration, to avoid an unrealistic drift during the spin up. It is therefore common practice in models of intermediate complexity (and even more so in climate models) to compensate for sediment fluxes. Once equilibrated, these fluxes can be decoupled in the UVic model. While the sediment fluxes continue to be prognostically computed, the surface fluxes can be held either constant or calculated with a simple weathering model (for alkalinity and DIC, such an option has not yet been added for silicate and the carbon sediment model is not used here). For this paper, we are running the model transiently on very short timescales, during which any imbalance in these fluxes would not have a significant impact. We therefore continue to compensate during our transient run. We use the diagnosed global Si river flux as an indicator of overall model performance by comparing it to published estimates.

The sentence is revised for better clarity (+ "rates").

5. The use of T in Eq. 3 and elsewhere is awkward because it relies on use the Celsius scale. If you swapped to Kelvin, 20/15≠ (20 + 273)/(15 + 273). Looking at Eq. 3, 'a' is a growth rate parameter (not a maximum growth rate as described in Table 2). Infact, it is not even an exponential growth rate parameter, since the exponential component is in the term e(T/Tb).

Equation 3 is previously published with several earlier versions of the model. For consistency and referencing between model versions for model users, we prefer to leave it as-is and add the scale/units to the text. The 'a' term has been edited (P4, L 103 +"modifying a growth parameter (a).") and in Table 2.

6. Eq. 4. It would be preferable that you use Fe for the Chemical Symbol, and [Fe] for the concentration of iron. Also for other elements. Reasons are highlighted in later comments.

Chemical concentrations are now given in brackets.

7. Eq. 8 looks odd but behaves okay. In any case, there are three constants in the equation which should be parameters.

The constants in Eqn 8 are taken directly from Aumont et al. (2003)'s equation, who derived it as a fit to experimental data (and this is now more explicit in the paper). We have now replaced one of the constants with the original parameter name, but the other two were not given a name in the Aumont paper. The Aumont paper is cited with Eqn 8 and the text now reads (P5 L124):

Silica uptake uses the empirical Aumont et al. (2003) scaling of kSi in mol Si m−3:
… with a kSi value adopted from Aumont et al. (2003) of 3e ^-2 mol Si m-3 .

8. Line 144. Mortality from 'old-age' is a misleading description since you do not track age distribution of the population. How about simply non-grazing mortality.

The terminology is modified to 'non-grazing mortality' on P6.

9. Interesting that you have self-grazing in the zooplankton!
This has been a model feature since Keller et al. (2012).

10. Line 157. Should state that the sum of the food preference parameters must be 1. This would make it clear that it is a 'relative food preference'. While this grazing form meets a local mass conservation criteria, ....
The language is edited to 'relative preference' and ('and the sum of all preferences must equal 1' is added to L161).

.... it is, nonetheless, awkward. For example, if you had one phytoplankton species, its preference would be 1, and G1 would be mu Z X / (X + k). If you split this into two identical classes with 0.5 preference, then G12 = mu Z [0.5 X/2 / (0.5 X/2 + 0.5 X/2 +k ) + 0.5 X/2 / (0.5 X/2 + 0.5 X/2 +k ) ]=muZX/2/[X/2+k]≠muZX/(X+k). To test this, try running your model with two identical LP cases, starting with half the concentration. Thus it is not only relative preference, but also specific to the predators and prey you have in this configuration.
We agree with the Reviewer. However, since we only have one predator in our model, the equation is sufficient as given.

11. Eq. 20, 39 Brackets around max function are different than Eq. 15 and others.
Brackets are changed.

12. L163 – what does 'sox' mean? I suspect it should be ro2sox. Think about how you have used subscripts to identify cases (i.e. phytoplankton type, element type) and superscripts to define its application (i.e. max). Eqs. 10 and 11 also appear to have the super- and sub-scripts the wrong way around.
'sox' stands for sub-oxic. We apologize if this is unclear from the text- the original text said 'hypoxic', but this has been updated. All of these equations have been previously published in the form in which they are presented. Changing the naming will be confusing to model users who want to compare versions.

13. Eq 30. Rethink subscript. If 0 is meant to identify an example of a ratio of Opal to POC, then it should be written as ROpal:POC,0. Lots of places subscripts would benefit from commas.
A comma has been added to the subscript for this (now Eqn. 31)  and Eqns 33, 34.

14. For example CaCO3liv would be better as [CaCO3]liv as proposed above. Eq. 25, Feorgads also mixes chemical symbolism with mathematical notation.
These modifications are made (Eqn 29, 42,43, 45, and 25,26,27, 40, 41).

15. I don't understand Eq. 32. If it is a local rate of change why does it have units of m-1. [Reading later it looks like it is a benthic term – needs clarification].
The equation was incorrect and we apologise for this. This is a function that instantly distributes opal dissolution down the water column. It is similar to how earlier versions of UVic (since Schmittner et al., 2008) distribute calcite down the water column. The text is clarified as:

(P8 L222)
We approximate an exponential flux function and apply our e-folding temperature parameterisation to represent microbially enhanced dissolution in mol Si m−3 units:…
(P9 L225)
This parameterisation results in greater dissolution at warm temperatures and is similar to the instant-sinking-and-dissolution function applied to model calcite (Schmittner et al., 2008) (although, the function for calcite was replaced when a prognostic tracer was added by Kvale et al., 2015b).

16. There is inconsistent us of the (×) symbol. Equations in latex format work better without them. Only use where it improves clarity.
We have modified its use in a couple of cases. Because of the dense use of subscripts and superscripts, its visually helpful to use the latex symbol "\times" to separate terms.

17. Instances of SCaCO3 should be S(CaCO3) (Eq. 29, 41, 42 etc.)
We did not use parentheses here to differentiate between a quantity that is calculated but not traced. This is now clarified in the text (P8 L 209).

18. Eq. 43 is simplistic as the units don't work. I presume Pr(Opal) is a 3D flux, I am not sure what Di(Opal) is (see comment 15), and Si(Dust) is probably a 2D flux, and Siriv is probably multiple point sources. But Si in equation 8 and elsewhere is a concentration. Another reason to represent concentration of an element as [Si]. Eq. 40 has similar problems.
We apologise for the confusion. Equation 40 (now 40 and 41) was presented in the form of an earlier publication and Equation 32 was incorrect and this affected 43 (now 44 and 45). Chemical concentrations are now clarified and the equations are now corrected. More information is added to the start of Section 2.2.9 that clarifies units of S and B terms:
(P10 L270, L283)
…(in concentration/time units)…

19. Line 230. Why does sinking increase with depth? I would have thought the increasing density with depth would result in a reduced negative buoyancy, and dissolution would reduce particle size slowing the Stokes' sinking rate? Coagulation might go the other way. Discuss more.
This parameterisation has been used in UVic since Schmittner et al. (2005) because it produces a good fit to the Martin curve (Martin et al., 1987):
Martin, J. H., Knauer, G. A., Karl, D. M., and Broenkow, W. W. (1987). VERTEX: carbon cycling in the northeast Pacific. Deep Sea Res. A 34, 267–285. doi: 10.1016/0198-0149(87)90086-0

Observations also suggest increasing sinking speed with depth, e.g. Berelson (2001). DSR-II, Particle settling rates increase with depth in the ocean. 10.1016/S0967-0645(01)00102-3.

We have now cited the earlier UVic models and Berelson (2001) (P9, L239) and added a new sentence (L242).
Alternative parameterisations exist and their effects on fluxes and model performance make for interesting comparisons (e.g.,Cael and Bisson, 2018), but we do not explore them here.

20. L266. M-M uptake rate? I prefer the line 102 description of iron availability. Perhaps a table of 'derived variables' with definitions of mathematical symbols that are in the equations, but not state variables or parameters would help in tightening up these descriptions.

We apologise for the confusion. This is how this term has been presented in previous UVic model description publications. The terminology is now made consistent with the other nutrient availability terms and with more explanation:

P10, L 276

…nitrate availability (diazotrophs use nitrate when available).

21. Eq. 36. bsi would be better as bSi.

Done

22. Eq 36 Exp is potentially confusing with the exponential function.

Changed to F(Opal)

23. Eq. 21. Would it be better to use oxygen concentration of % saturation. I am not sure which it is that affects animal metabolism.

This formulation has been used in this model since Keller et al. (2012), and we prefer to leave it as-is for now.

Other minor comments.

Line 13 'migrate' – poor word choice. Perhaps 'bloom further south'

The sentence is changed to 'distribution moves southward'.

Line 94 – what does a virtual flux mean in this context.

A virtual flux is a correction to account for the evaporation/precipitation effect in a rigid lid ocean model. This is now made more explicit (+ 'evaporation-precipitation correction')

L276 replace 'before the biology' with 'faster than the biological processes'

Done

We again thank the Reviewer for their help in improving our manuscript.

---

## Author Response (AR2)

**Topical Editor Decision: Reconsider after major revisions** (26 Jan 2021) by Andrew Yool
Comments to the Author:
Dear Authors,

Thank you for your responses to your referees and for your revised manuscript draft.

I have now reviewed these materials, and while I judge that your answers address the majority of the points raised, I believe the discussion around the performance of the model's silicon cycle could be more thorough. This aspect of the model is the focus of this study after all and, as the referees remark, the model has some clear deficiencies.

One way to address this could be to contextualise the model's performance against comparable global-scale models, such as those produced as part of the CMIP process. Many of these models now include a silicon cycle, and the CMIP repository include a number of relevant diagnostics, including silicic acid and diatom biomass / production distributions. Models are, inevitably, always deficient, but a straightforward way to illustrate the utility of one is to show how it improves our ability to represent a real world system relative to the current state-of-the-art (or, at least, compares with this).

I would appreciate it if you could expand on this portion of your response to your referees. As both of the referees judged your manuscript to require major revisions, I will be returning it to them subsequently, but I believe more completely addressing this issue of performance (i.e. relative to other silicon cycle models) would strengthen the manuscript draft ahead of this.

Please get in contact with me should you have any questions or require any assistance about any of the above.

Thank you again for your response and revisions.

With best regards,

Andrew Yool.

Non-public comments to the Author:
Dear Authors,

Further to my comments, I would just add that I have some direct experience with CMIP models, so can provide some guidance if necessary.

Andrew Yool.

The authors thank Professor Yool for considering our manuscript. We have now included comparison to CMIP6 data. In addition, we have included comparison to a newly-published silicon cycle review that suggests our model is actually performing better than previously thought.

Best regards,
Karin

---

## Author Response (AR3)

The authors would like to thank the Reviewer for their careful assessment of our manuscript. Please find Reviewer comments reproduced below in blue font, with our responses provided in black font. Changes to the manuscript are provided in red font both below and in the accompanying manuscript.

**Main comments and recommendation**
* * *
The manuscript by Kvale at al. presents some new extensions to the ocean biogeochemistry in the widely used UVic Earth System Model, of which the most important one is a description of silicon cycling in the ocean. Together with the presentation of the new model, an evaluation of the models present-day biogeochemistry, and a future model scenario run that extends to the year 2300 are shown.

The future run is interesting, as it predicts a strong shift in phytoplankton species dominance after the year 2100, especially in the Southern Ocean, from diatoms to CaCO\$\_3\$-producing phytoplankton, which would have strong implications both for marine biology and biogeochemistry. I am, however, sceptical about the robustness of this result, given that the model description of the competitive advantages of the different phytoplankton functional groups in this model (as in most models) is overly simplistic and has been strongly tuned to agree with our (also still limited) knowledge about their present-day distribution. I think it is fair to say that at the present state of modelling, one can put some confidence in the modelled biogeochemistry, but not in the ecosystem functioning. The abstract of the manuscript almost entirely focusses on this, in my view still at least questionable, part of the results.

The authors agree with the assessment of the Reviewer that the future run is interesting and has implications for future biogeochemistry and biology, but that readers should be sceptical of the result, which is produced by a single parameter set of a single model. This is one reason why the last sentence of the abstract states that "These results are meant to serve as a baseline for sensitivity assessments to be undertaken with this model in the future"- it is a disclaimer against taking any results from this early version of the model too seriously.

We have re-arranged the abstract to place more emphasis on the historical simulations and model description itself, and to reiterate the RCP 8.5 projection is meant as a demonstration of the model function.

L 4:

This new model combines previously published parameterisations of a diatom functional type, opal production and export with a novel, temperature-dependent dissolution scheme. Model biogeochemical rates, carbon and nutrient distributions are similar to those found in previous model versions. We assess the fully-coupled model against modern ocean observations and the historical record since 1960...

L 10:

The model simulates a global decline in net primary production (NPP) of 1.8% having occurred since the 1960s, with the strongest declines in the tropics, northern mid-latitudes, and Southern Ocean. Based on a single parameter set tuned to observations, we also

**perform a first projection of potential biogeochemical and ecological changes under a business-as-usual atmospheric CO2 forcing to the year 2300.**

I think this neither fits to the overall scope of the manuscript, nor to the journal this is submitted to. If this part of the manuscript has indeed such importance, its uncertainties should have been discussed much more in the main text of the manuscript, and the manuscript should probably been submitted to a different type of journal.

We appreciate the ecological trends in the transient simulations may be considered to fall in a grey zone with respect to the journal's scope and might be interpreted by readers as "science" (in which case, a thorough discussion of model uncertainties is absolutely necessary and a different journal would be more appropriate). However, biogeochemical models are commonly used to make projections of future ecosystem functioning, so reporting first results from transient experiments is useful for possible applications of the model, or other models, in the future. The Copernicus website lists at point 5 of the scope of Model description papers, "Examples of model output should be provided, with evaluation against standard benchmarks, observations, and/or other model output included as appropriate". We consider these transient simulations, and their ecological responses, to be baseline examples of model output using a standardised forcing scenario (extended RCP 8.5) that are comparable to other model output (UVic ESCM ecosystem simulations are typically run to year 2300). Section 4.2 includes discussion of how the model's transient responses compare with other, previously-published "science" runs (Schmittner et al., 2008, Kvale et al., 2015a, Nickelsen et al, 2015, Moore et al., 2018, Fu et al., 2016). Encouragingly, the model shows several similar long-term responses to other, even unrelated, models.

We plan to conduct "science" experiments using this model, in which ecosystem responses are explored, but using a more rigorous approach that includes parameter optimisation techniques and uncertainty quantification- not a single parameter set, but suites of parameter values. This kind of experimental setup is outside the scope of GMD.

**I will limit my review therefore to the description of the model and the evaluation of its present-day state, with a focus on the silicon cycle.**

First to the description of the model. The two main new aspects of the silicon submodel that is presented here are a) the description of the dissolution of opal in the water column and b) a simple benthic transfer function for opal that thinks into the sediment. Other aspects, like the description of diatom growth and Si uptake are more standard (which is not a criticism). The second of the new aspects, the benthic transfer function, is fairly simple (30\% of the opal sinking into the sediment is permanently buried if the flux is above a certain flux threshold, and 5% below), but the description is clear and the parameterization is a reasonable approximation to our current understanding of global sedimentary Si fluxes.

**Thank you.**

The first new aspect, however, is first of all neither mathematically nor verbally described well enough. Opal sinking is not modelled explicitly, and only the divergence of its vertical flux (equivalent to the release of dissolved silicic acid) is represented. The textual

explanation of this in the single sentence "We approximate (What?) (by) an exponential flux function and apply our e-folding temperature parameterisation to represent microbially enhanced dissolution" states this in a rather unclear way.

Firstly, we apologize for Equation 32, which was presented incorrectly. This parameterisation of opal vertical dissolution takes a similar approach as the default UVic ESCM parameterisation of CaCO3 (Schmittner et al., 2008). We have rephrased the text (P 9, L 227):

Dissolution of opal in the water column is calculated by assuming instantaneous sinking of the vertically integrated production, where the flux of opal is distributed down the water column using the e-folding temperature parameterisation (unitless), scaled by a dissolution rate constant ( $\lambda$ Opal, in day-1) which is multiplied by the depth (z, in meters), divided by a sinking rate (wOpal, in meters day-1)

The following equation 32 is not better: first, the boundaries in the vertical integral are not stated. Well, we can assume that it is over the whole water column. But then follows a flux divergence term (which is not stated as such) d/dz(something), but it is completely unclear which of the terms that this derivative is applied to actually depends on z. In the form as it stands here, the only potentially variable term is ocean temperature, as both \$\lambda\$ and \$w\_D\$ are constants. That cannot be correct: if temperature was constant there would be no dissolution? I am also worried that by scaling the dissolution with the water column integral of production (and not the integral obove the depth where the dissolution is calculated), one could potentially dissolve more than is produced in the upper model layer, and given the very fast dissolution in warm waters used here I actually suspect that this happens in the model.

Again, we apologise for the incorrect formulation of Eqn 32, which did not reflect what is actually coded into the model. The equation is now corrected in the text (P 9, L 231).

$$\mathrm{Di}(\mathrm{Opal}) = \int_{0} \mathrm{Pr}(\mathrm{Opal}) dz \cdot \frac{d}{dz} \; (\frac{\lambda_{\mathrm{Opal}} \times z}{w_{\mathrm{Opal}}} \; \exp^{\frac{T}{T_{b}}}).$$

We have also added boundaries to the integral. Lambda is in units of per day, while w\_opal is in units of m/day. If temperature is constant then exp^(T/T\_b) is non-zero. A higher temperature produces a sharper gradient in dissolution profile between the surface and deep ocean. The model code constrains the formulations to ensure that there cannot be more dissolution than production, and this is now explicitly stated in the text (P9 L232): Dissolution is presented in units of mol Si m\$^{-3}\$. The code constrains total water column dissolution to be no greater than total water column production.

Besides being described in a clearer way, the new parameterization should also be justified better. The existing justification (line 225-234) is rather superficial. Making the dissolution of opal as strongly temperature dependent as the breakdown of organic matter is justified here by the requirement to strip away the organic coating of diatom frustules before dissolution can set in. But that argument (which is incorrectly ascribed to Sarmiento and Gruber, 2006 here; it is in fact from Bidle et al. Science, 2002) has originally been proposed to explain just the opposite, namely that very little dissolution happens within the euphotic zone, as the bacteria first need to break down the coating before dissolution can set in, while the frustules have by then already sunk down into the ocean's interiour. So this justification simply does not hold, and the only remaining justification is that it improves the fit to the WOA dissolved silicon distribution. That is ok, given the contradictig information from the many experimental studies and the widely varying temperature dependencies in models. But it should then be acknowledged that this parameterization may compensate for physical deficiencies in the model, e.g. an overly small vertical mixing due to the low vertical resolution.

We thank the reviewer for pointing out this mistake in our reasoning. We have removed discussion of microbial activity and instead re-phrase the paragraph as follows (L 234): We find this parameterisation offers improved model fit to World Ocean Atlas silica distributions relative to other parameterisations that we tested, e.g. the temperature-dependent parameterisation of Gnanadesikan (1999) or the temperature and oxygen-dependent parameterisation of Enright et al. (2014). The Gnanadesikan (1999) parameterisation yields lower dissolution rates at low temperatures than the Enright et al. (2014) parameterisation, which is similarly formulated but which includes an additional oxygen scaling. The Enright et al. (2014) oxygen scaling is not justified in their model description, but it has the effect of increasing Si dissolution rates in the deep ocean (exacerbating the overestimation of Si dissolution in this region by the Gnanadesikan (1999) scaling described in Ridgwell et al. (2002) and decreasing Si dissolution rates (to a lesser extent) in the near-surface. Our temperature scaling has the effect of raising dissolution rates at the surface may be necessary to compensate for the low vertical resolution of the model.

One aspect of the model evaluation, the modelled present-day distribution of dissolved silicate, has been already discussed by reviewer 2, and I have nothing to add here. But another aspect of the evaluation has left me completely confused, namely the comparison of the global Si fluxes with the published data-based estimates from Treguer et al. (2021) and Treguer and De La Rocha (2012). I will try to explain the inconsistencies in this comparison, as I perceived them, acknowledging that I got confused in several places. The comparison is done in lines 327-340 and in Table 6, and I have reverted the numbers given there to the more commonly used unit of Tmol Si/year.

- The authors state that "Diagnosed surface opal production is within the range of a recent estimate (Tréguer et al., 2021)". This seems to be the case. From my unit conversion I get a production of 270

**Tmol/yr, compared with 255 in Treguer.**

- But the Si export out of the euphotic zone is about half of the estimate given in Tréguer et al., 2021: 57, compared to 112 Tmol/yr. Assuming that most of the production is taking place in the upper 130 m of the water column, I thus calculate that the dissolution of opal within the euphotic zone alone is 218 Tmol/yr.

- But that contradicts with the statement of opal dissolution in Table 6, which is 138 Tmol/yr. They compare this number to a number of Treguer et al, which they state as 170 Tmol/yr. In Treguer et al., I find a dissolution within the euphotic zone of 143 Tmol/yr, and in the water column below the euphotic zone of 28 Tmol/yr, so I assume that they mean their 138 Tmol/yr as the total dissolution both within the euphotic zone and in the water column below. But that cannot be correct, see above.

- The dissolution of 138 Tmol/yr also cannot be the dissolution only below the euphotic zone, as we have learned above that the total export of opal is only 57 Tmol/yr. So, what is the dissolution number actually?

- in Treguer, the majority of the exported Si lands in the sediment (84 of the 112) and is mostly dissolved there, only 9 Tmol/yr gets buried. From Table 8 I see that this 9 Tmol is compared to a model value of 47 (they call this net seafloor Si flux). And from the model description I get that this is returned to the ocean with rivers. Is that so? On the other hand they say that they underestimate the riverine influx. This does not fit, maybe I misunderstand something serious here.

We again thank the reviewer for pointing out a serious mistake in our Table 6. This table had not been correctly updated in our last revision. The 2 km opal flux was from the Honjo et al. (2008) dataset, but we have now changed this to the Treguer et al. (2021) estimate. Also, we have converted our fluxes to Tmol Si/y to be consistent with convention. And, the seafloor flux in our model includes opal flux above 2 km depth, so there is more seafloor flux than flux below 2 km. Since our model does not estimate sedimentary processes, our seafloor flux is not a good comparison to the Treguer et al. (2021) burial estimate (and has been removed to avoid confusion). We have also updated the text to reflect these changes (throughout Section 3, see L 337-347 especially).

It is true that silica parameterised to be lost to the sediments in our model is replaced via the river fluxes. In our model, currently the estimated river input is 1.52 Tmol Si/y, which is below 8.1 Tmol Si/y estimated by Treguer et al. (2021). This is discussed at L 343: The calculated river flux is 1.52 Tmol Si y\$^{-1}\$; lower than the \citet{Treguer etal 2021} estimate of 8.1 Tmol Si y\$^{-1}\$...

We have now made it more clear that we compare the total water column dissolution (euphotic zone plus below) to the Treguer et al. (2021) estimate, and have added a discussion of production to dissolution ratios (L 343):

...but the ratio of total water column dissolution to biogenic production is more encouraging; 0.5 in our model compared to 0.67 calculated from Tréguer et al., 2021).

Overall, what we now show are silica flux and dissolution rates that are only about ¼ of the Treguer et al. (2021) estimates. We have also added some discussion of these results in the context of inverse modelling of silica cycling, which estimates rates based on a different approach than extrapolation of direct measurements, such as Treguer et al. (2021). L 347:

However, it is worth noting that previous inverse estimates of global silica fluxes (Holzer et al., 2014; Pasquier and Holzer, 2017) estimate global export production (166 and 171 Tmol Si y–1, respectively) substantially higher than the 112 Tmol Si y–1 estimated by Tréguer et al. (2021), which suggests poor agreement still exists across estimates of the global silica budget

My overall recommendation is therefore that both the description and justification of the new temperature-dependent opal dissolution, and the description of the present-day global Si flux balance still need major revisions for the manuscript to become a useful addition to the literature.

We hope our responses to the above points adequately address the Reviewer's concerns.

Minor comments

Line 97: Maybe it would be good here to mention in what unit the phytoplankton biomasses are calculated. This information is added (L 99)

Line 111-112: It has become common to call the Michalis-Menten uptake limitation term "iron availability" but this term creates the possibility to confuse it with "bioavailability of iron" used in the biological lierature, which is related to iron speciation (and actually has the unit of a concentration, not dimensionless, as the MM-term).

The original language was modified to "availability" at the request of an earlier Reviewer. We have changed it back to "uptake". (L 115)

Line 124, Eq. 8, line 125: First of all I find it a typographical crime to state a number \$8 \cdot 10^{-4}\$ as 8e-4, like in matlab code. Second, the numbers in the equation are stated without unit, which is false.

**The formatting is changed and units are added. (L 128/129)**

Equations 9, 10, 11: Making the ChI:C quota just dependent on iron is a gross simplification, as it takes away the much larger dependency of ChI:C on irradiance in acclimated cells. For the purpose here it still may be o.k., but that should be stated. Secondly, the two factors \$\alpha\$ and \$\theta\$ that are both made linearly dependent on Fe, appear only in the combination of \$\alpha \theta\$ in Eq. 9, so in effect the relationship is quadratic. Is there any justification for that?

None of these equations are new to this version of the model. Please note that Chl is not an explicit tracer. This is a one-off gross calculation used for the iron model to link iron availability to light affinity. It is kept for historical reasons (it was introduced by Nickelsen et al., 2015). Flexible ratios are not introduced to other aspects of the model because the focus of this manuscript is silica cycling. Other researchers (Markus Pahlow, Chia-Te Chien) are currently working on this aspect of the KMBM (e.g., Chien et al. (2020), GMD, doi.org/10.5194/gmd-13-4691-2020).

We have added a new sentence (L 138):

The approximation of \$\theta\$ is simplistic and neglects other factors, e.g. irradiance, which can affect the ratio.

And another at L 132:

Nickelsen et al., (2015) introduced this parameterisation to the model and discuss it at length; interested readers are recommended to read their Section 2.3.2. In the above equation...

Equations 26, 27, and several others later: The authors have the habit of indicating sources of tracers (like in equation 26 the scavenging loss of iron to particles) by an expression \$[Fe]\_{orgads}\$, i.e. with square brackets around the element symbol. The convention with square brackets in the chemical literature, however, is that these denote 'concentration', so \$[Fe]\$ is the concentration of Fe. \$[Fe]\_{orgads}\$ would then be one part of the dissolved Fe pool, not a rate. The same confusing convention is used thoughout the manuscript, e.g. equations 35, 36, 40 and 44. Moreover, it is completely confusing to write the scavenging rate \$k\mathrm{Fe}\_{org}\$ as one could confuse this with a product of a rate \$k\$ with some concentration \$\mathrm{Fe}\_{org}\$. I fell for this and searched for the definition of it.

We agree that this concentration-style formatting might lead to confusion, but it was explicitly requested by a reviewer in the earlier stages of the revision. We have changed the formatting for scavenging.

Equation 39: I think the scavenging by CaCO3 is missing here.

**It was, thank you. The equation has been corrected.**

Lines 301 - 304, equation 44: It is nowhere mentioned where the river input of Si, which is used to balance the Si budget, is applied. Does the model use a prescribed runoff distribution also for the Si input, or is the river input distributed homogeneously over the ocean surface, as e.g. in HAMOCC?

River inputs are scaled against river flow using a mask provided as a standard input with the model. This is now added (L 314):

River inputs of silicate, alkalinity, and DIC are scaled against seasonally-variable river flow using the standard UVic ESCM version 2.9 O\_rivflux.F forcing file.

Line 312: Everywhere else, Si fluxes are given in Pg Si /yr, why here now in Tmol / yr?

We have changed all Si units to Tmol/y in the text.

Lines 379 ff: It is unclear to me what one learns from a Taylor plot of CMIP6 model diatoms relative to the distribution in the model shown here. I would suggest to do the comparison otherwise, not all referenced to one specific model.

We had produced the Taylor plot in this way because the CMIP models are just so different from both each other and the diatom observations that a Taylor plot of biomass referenced to the very sparse dataset was meaningless. By using our model as a reference we tried to show how similar the CMIP models diatom biomass are to each other and our model. However, we have now changed this plot to a comparison between CMIP6 model mean diatom biomass (black dot) and the KMBM3 (red dot). We have changed the text accordingly (from L 390):

We compare KMBM3 model output relative to annual mean CMIP6 model output due to the very low normalised correlation (0.04-0.15) and normalised standard deviation (0.10-0.22) of all models against the sparse Leblanc et al. (2012) diatom biomass dataset. At the time of writing, available CMIP6 simulated annual average diatom biomass shows diverse quantities (maximum concentrations from 0.0035 to 0.03 mol C m-3), and spatial distributions ranging from global maximum concentrations at the Equator (CanESM5-CanOE, Swart et al. 2019), to shallow seas and coastlines (IPSL-CM6A-LR, Boucher et al. 2018), to the high latitudes (CMCC-ESM2, EC-Earth3-CC, GFDL-ESM4, CESM2; Lovato et al. 2021; EC-Earth Consortium 2021; Krasting et al. 2018; Danabasoglu 2019, respectively). Fig. 5 summarises the KMBM3 diatom biomass estimate relative to the mean CMIP6 diatom biomass, simulated at year 2014. The KMBM3 is most closely correlated with GFDL-ESM4 (not shown). As stated above, phytoplankton biomass is difficult to tune for (particularly when multiple functional types are represented) due to the under-constrained parameter space, the theoretical nature of phytoplankton functional categories, and the sparsity of gridded, annually-averaged biomass datasets. Therefore, a wide range in model biomass estimates is expected across the CMIP6 ensemble.

Line 399: What are the units of the given root mean square errors?

**Units are added to the text. (now L 405).**

Line 664: The scavenging of iron on calcite is not new, as far as I know it is used in Moore's BEC model as well.

This is very interesting to know, thank you. We have removed "novel".

Figures 2, 3, 4, 6 and several more: It might make sense to re-do the plots with a proper map projection

While we appreciate that map projections are look sleeker, very few UVic model description papers, or UVic science papers for that matter, have used them. Therefore, we prefer to keep the figures as they are, as it makes it easier to compare them with previously published figures.

Figure 5: What sense does it make to show a Taylor diagram with the model presented hare as the reference data set? Also, I would not call this 'normalized' to KMBM3, but 'referenced'.

Please see our answer given above. We have changed the wording.

Fig. 14: The model is clearly missing the HNLC region wth elevated Si in the North Pacific

The low bias in Si in our model appears to affect representation of all HNLC regions, despite our model performing well relative to the CMIP6 suite. Hopefully this will be resolvable with our future parameter sensitivity studies.

---

## Author Response (AR4)

**Topical Editor Decision: Publish subject to minor revisions (review by editor)** (03 Aug 2021) by Andrew Yool Comments to the Author: Dear Authors,

Thank you once again for your manuscript revisions and responses to referees.

Given the significant criticism levelled by one referee, and the specific and detailed points raised in their review, I returned the manuscript to them for an evaluation of your revisions.

In reply, the referee is broadly pleased with your revised manuscript, and finds that it addresses much of their criticism. Nonetheless, they still identify a significant lack of clarity in the description, and rationale, of the dissolution parameterisation used. As this aspect of your model is a key determinant of its operation and skill, having clarity in details of its formulation is highly important.

To this end, I am returning the manuscript to you to address these remaining points from your referee. As ever, should you have any questions about the referee's remarks, please feel free to get in contact with me and I will endeavour to have them clarified.

Finally, please accept my apologies for this extended review process of your manuscript.

With best regards,

Andrew Yool.

Dear Dr. Yool,

Thank you for coordinating this review of, and for the preliminary decision on, our manuscript. Please see Reviewer comments given in blue font, and our responses provided in black font, with changes to the manuscript reproduced in red font.

One of the new points arises from confusion, where it seems the reviewer did not see in the last rebuttal where I explained some of the flux numbers were outdated and replaced- over the past year I have continued to tune the parameters of the model and there was a stage where new plots were added but the table was not updated, which was the source of the original confusion and for which I would like to apologise. Since the last revision the table is consistent with the plots, but the reviewer apparently did not notice this statement in the rebuttal.

But I would like to thank the Reviewer for pressing the point about inconsistent opal fluxes. Upon examination I found that the model diagnostics output was not correctly reporting opal export, dissolution and sediment losses applied at model runtime. This has been fixed and the simulations have been re-run, with new data files and model code uploaded to the GEOMAR data repository. New figures have also been created for the manuscript and the text adjusted accordingly (in red font).

And we have tried again to represent the opal dissolution coded into the model in a form that is understandable to readers. I am sorry this equation (32) has produced such confusion!

You might be interested to know that a first paper has been published using this model: Saini et al. (2021) Southern Ocean Ecosystem Response to Last Glacial Maximum Boundary Conditions, Paleoceanography and Paleoclimatology https://doi.org/10.1029/2020PA004075

Best wishes, Karin Review of the revised version of "Explicit silicate cycling in the Kiel Marine Biogeochemistry Model, version 3 (KMBM3) embedded in the UVic ESCM version 2.9" by Kvale et al., submitted to Geoscientific Model Development in 2021

Main comments and recommendation

This is the second version of this manuscript that I review, and the authors have taken many of my comments to the previous version into account. My two main points of concern about the last version were unclear or wrong explanations in the model description and inconsistencies in the described global Si fluxes and the comparison with observation based estimates. I still think these two points need a bit of work, and so I would recommend that the manuscript should be further revised before it can be published.

We apologise for both the wrong presentation of model equations and for the confusion arising over inconsistent fluxes. As explained in the last rebuttal, the fluxes had not been updated after a switch to a better parameter set. Also an error was found in the model diagnostics. These problems are corrected here.

While most of the remaining unclear points in the model description are rather minor, one concerns a central part of the manuscript, the implicitly treated dissolution of the sinking opal. As it is one of the main points of the manuscript that this formulation of opal dissolution improves the modeled global silicic acid distribution, this is an important point.

Several biogeochemical ocean models use model-specific temperature-based formulations of opal dissolution. We tried 2 of them (Gnandesikan 1999, Enright et al. 2014) and neither produced acceptable silicate distributions, probably because the parameters applied were adjusted to their different model physics. Which is to say, we are not the first to use an approximation that works for our model (reducing the importance of our particular parameterisation, which might or might not be portable to other models). Explicit dissolution will be implemented over the next months, so this first approximation is meant as a 'placeholder' and also to provide a benchmark on performance as the code is improved in the coming years.

The dissolution of opal is described in equation 32. The equation has changed with respect to the last version by the multiplication with depth (z) within the flux divergence term on the right hand side, so that now the production Pr(Opal) and dissolution Di(Opal) have the same unit, which makes sense. In the manuscript is is stated that the unit of Di(Opal) is mol Si m{-3}, but I think, as it is a rate, it should be mol Si m{-3} yr{-1}.

Yes, that is correct. It should be per day and this is now corrected in the text (Line 232).

But more importantly I still do not understand the rationale behind equation 32, for two reasons:

- Is the chosen form of the flux compatible with a reasonable assumption on the implicit vertical distribution of sinking opal, or of the sinking flux? I do not think so. As a simple example, assume an ocean with uniform temperature (which is a good approximation to the ocean below 2 km depth). Then the immediate consequence of eq. 32 is that opal dissolution Di(Opal) is constant throughout the water column. If the concentration of opal decreases with depth (as it must, because of dissolution), then that would imply that the dissolution rate needs to increase with depth. Or, if the dissolution rate is constant, then opal concentration must also be constant vertically. Of course, the chosen temperature dependence will lead to a decrease of the scaled flux (the quantity within the d/dz) with depth, and thus to a more realistic vertical behaviour of the flux. But it still remains unclear to

me how the proportionality of the flux to depth z can follow from an assumption on the steady-state vertical distribution of opal biomass and dissolution rate.

We can not plot the vertical distribution of sinking opal because we do not model this quantity explicitly. However we find that our parameterization produces an average export rate of opal that decreases with depth. In so doing, observed dissolved silica distributions are reasonably reproduced.

We had some internal discussion about how the equation is represented and apologise for how confusing this equation has been. We have tried again, and hope that it now makes sense to readers. Its actually very similar to Gnandesikan (1999) Eqn 2. I think my earlier attempts to represent it were overly complicated.

- Also, if the parameterization in eq. 32 would be compatible with a vertical distribution of sinking opal, then it would automatically follow that the integral over the total water column dissolution would be smaller or equal to the integrated production, independent of the chosen dissolution rate and its temperature dependence. In the current formulation, however, this is not the case: Here it has to be ensured (probably by making lambda small enough) that the flux (scaled by the vertically integrated total production) at the bottom of the ocean lambda\*depth/w \* exp(T/Td) is not larger than one.

We apologize if the Reviewer thought we needed to artificially constrain the dissolution to be less than production. This line of code is in place just to be sure in the case of numerical instabilities, but it is not required. But, we have removed the sentence from the manuscript which we had added as a response to this point in the last rebuttal in order to avoid confusing readers. (As a note to the Editor: it seems this point is referring to a point made in the previous review, to which we responded by adding a sentence that seems to have only confused our answer. This new sentence is now removed.)

I think the authors should discuss the relationship of their chosen flux parameterization to assumptions on sinking speed and the vertical distribution of opal. Probably this cannot be done in a strict analytical way (with a temperature-dependent dissolution rate, an analytical solution for a steady state flux profile is hard to derive). But a qualitative idea why this flux profile was chosen should be possible.

We have added some text to the manuscript addressing opal flux (Line 235): Opal dissolution in the water column is primarily controlled by temperature and silicate saturation, with higher temperatures and lower silicate saturations leading to faster dissolution (Sarmiento and Gruber, 2006; Ridgwell et al., 2002). Secondary drivers include the aluminium content and surface area of diatom shells (Van Cappellen et al., 2002), and the presence or absence of organic coatings, where the loss of the coating increases opal dissolution rates (Bidle and Azam, 1999, Van Cappellen et al., 2002). We do not consider silicate or aluminum concentration dependencies explicitly in this model because we do not explicitly calculate thermodynamic dissolution of opal (nor do we simulate aluminium), but the temperature dependency exerts an influence on dissolution of the same sign (faster dissolution in warmer, i.e. shallower and undersaturated, water).

My other main criticicm of the last manuscript version were inconsistencies or confusing statements on the global Si fluxes. The authors have changed their notation now to Tmol Si/yr, which makes the comparison with the numbers in Treguer et al, 2021 easier. But in doing so, also some of the stements on numbers have also changed, and to me it was really unclear now how the different terms are defined:

The statements had to change because the values had to change because the dataset underlying the statements changed- we updated the model with a new parameter set 2-3

revisions ago, which somewhat improved model performance compared to the originally submitted dataset. Unfortunately the figures had been updated but the table had not. We apologise for the inconsistency, which has caused confusion. Now the table values are consistent with the figures.

- In the new manuscript (line 338-340 and table 6) it is said that the total opal production is 127.9 Tmol Si/yr, which would be low against the estimate of 255+-52 Tmol/yr in Treguer et al., 2021. But in the previous manuscript version the total production was stated as 7.68 Pg Si/yr, which approximately translates to 270 Tmol Si/yr, close to the Treguer et al. estimate. So, was the old number simply a miscalculation? There are more differences in numbers between the old and the new manuscript version, e.g. in the stated total dissolution. Is this due to a different way that the term has been defined?

No, it is an updated model output which was converted to Tmol Si/y. Table 6 numbers are now consistent and replaced with a new set of values from the latest simulation.

- I also do not understand the relation between production, water column dissolution and sediment export of opal. If we take the statement from the new table 6 of total opal production of 127.9 Tmol Si/yr, and substract the total opal water column dissolution of 64.4 Tmol Si/yr, the difference (63.6 Tmol/yr) does not dissolve in the water column and hence must be deposited on the sediment. How is that compatible with the statement at the vertical flux of opal at 2000 m depth is only 19.46 Tmol Si/yr? Does that mean that more than two thirds of the opal deposition into the sediment happen above 2000 m depth, i.e. on the shelves?

Note Table 6 values have changed from previous drafts of the manuscript following a rerunning of the model after finding a problem with opal flux diagnostics. We thank the reviewer for pointing out some inconsistencies in the opal budget, which are now corrected. According to the estimated global silica budget by Tréguer et al. (2021) about 40% of opal burial occurs in the continental margin. Our model currently produces about 69% removal at the seafloor above 2000 m depth, which might be too high (though there is a lot of uncertainty about processes like downslope transport and how much opal is actually transported into the deep ocean along the bottom, and we do not model sedimentary exchange, burial, preservation, distinguish between coastal and open ocean, etc). This is a useful benchmark as we improve our model.

I think the authors should try to make it possible for the reader to close the opal budget by stating a) how the total production of Si is distributed between the upper 130m and deeper, b) how much of that production dissolves with the layers 0m-130m, 130m-2000m, and below 2000m, c) how much opal is deposited into the sediment within these three layers, and d) how much of the deposited opal is permanently buried within these layers.

We have now included the requested information in Table 6. Comparison with previously published budgets is more straightforward on a global basis, because, e.g., neither of the Tréguer et al. 2013 and 2021 reviews distinguish depth levels in this manner and our model is missing relevant processes (i.e. coastal processes, reverse weathering, sponges, does not distinguish between coastal zone/open water, etc., and has a low spatial resolution).

**Minor comments**

\_\_\_\_\_

- line 82-83, "any opal that reaches the seafloor is replaced by external sources": This contradicts the formulation of the simple sediment scheme (lines 266 ff.), where it is explained that only a fraction of what arrives at the seafloor is permanently buried and needs to be replaced by external sources to keep the inventory stable. Please clarify.

**The sentence is modified to read (Line 82): "any opal that is lost to the seafloor is replaced by external sources"**

- in my last review I suggested to replace the word 'bioavailability' for the Michaelis-Mentenkinetics-like term in nutrient uptake. This has been done. The replacement 'uptake' (lines 107, 115, 117, ..) is, however, equally misleading: the quantity that is described is a dimensionless limitation term, uptake would imply some material change, probaby in mol/volume/time.

These instances have been changed to 'limitation', or reverted to 'availability' as per the instructions of an earlier Reviewer (Lines 106,115,117,126).

- On line 117, it is said that the nutrient limitation terms for nitrate, phosphate and silicic acid are multiplied with the maximum potential growth rate. This is a bit misleading because it suggests a multiplicative limitation by the different nutrients; later on (eq. 13) it is clarified, that only the minimum of the limitation terms, not their product, is used.

The nutrient limitation terms are in fact multiplied with the maximum potential growth rate. Those resulting products are then compared, with the minimum selected. The language is changed to "the minimum nutrient limitation term" (L118)

- in equation 8, the two numbers given are not pure numbers, but should have a unit, namely that of a silicic acid concentration. I would suggest these numbers are replaced by a symbol, and that these numbers with their proper units are then added to the parameter table. Besides, I had already mentioned in my last review, that writing a decimal number as 8E-4, when you mean 8 times 10 to the power of minus 4 is something that fortran understands, but that is violating typographical conventions.

The units are provided for all numbers in Eqn 8, see L126, "with constants given in the same units". We prefer to leave the values in the equation, as this is not our equation (it is from Aumont et al., 2003) and it is easily recognizable in its present form. We have now changed to exponential notation.

- the description of the dependency of the initial slope of the PI curve and on the ChI-a:C ratio on the degree of iron limitation is correct, but could use a bit more justification. The neglection of the dependency of the ChI:C ratio on the light level is justified here, I believe, because in the model, ChI is really only needed in the surface layer (to calculate photosynthesis rates and to compare with satellite estimates), and not at depth, where the dependency on light becomes crucial.

This equation is not new to this model. We prefer to focus discussion on the parts which are new and unique, and offer a reference to Nickelsen et al. (2015) which describes the equation in greater detail.

- In line 159, the temperature-dependent fast remineralization is called 'respiration', in contrast to the linear mortality. It is maybe a bit picky, but while from an ecostem perspective bacterial consumption is a respiration, from the phytoplankton perspective I would rather describe it as excretion (because it is originally dissolved organic carbon that is excreted and then respired by the bacteria).

The parameterisation is identical across phytoplankton types and has been described as "respiration" in previous UVic model publications, but we have removed "respiration" and shortened the sentence (L 161)

- in line 162-163, the quadratic mortality is called 'non-grazing mortality' and contrasted with a self-grazing term GZ. Again it is maybe a bit picky, but to my knowledge the quadratic zooplankton mortality (the so-called closure term, Steele and Henderson, 1992) is exactly meant to represent grazing by higher trophic levels. See Edwards and Yool (2000) for a discussion of the different forms of closure.

We have not modified our zooplankton equation, other than to add diatoms as a new food source. This equation has been published with other versions of UVic (e.g., Nickelsen et al., 2015). But it does appear to be inconsistent with the quadratic mortality function found in Edwards and Yool (2000), and we thank the Reviewer for pointing this out. The language used in the manuscript text was also incorrect, and has now been changed to (Line 162): In addition to mortality from higher trophic level predation calculated with a quadratic mortality function ( $m_z Z^2$ ), intra-guild predation is represented with a separate term ( $G_z$ ). This parameterization of zooplankton mortality will need to be investigated in the future, as there are plans in the UVic user community to modify the grazing formulation as well.

- equation 21 contains several numbers, but some of the numbers are meant to have units (the 8 should be 8 mmol m4-3), while others are pure numbers.

**Units have been added to the 8.**

- line 198: 'organic carbon' should probably be 'organic nitrogen'

**This has been corrected.**

- in equations 27 and 27, should the iron scavenging rate not also depend on the concentration of (organic and CaCO3) particulates? Also, the term [Fe]prime is not defined. It probably means the concentration of 'free' inorganic Fe (often denoted [Fe']), probably calculated from total dissolved iron and ligand concentration, but this is not stated. I also do not understand why the prime is shown here with letters, instead of Fe', as is the conventional usage.

We used the same notation as Nickelsen et al. (2015) so that readers would know it is the same model. But we have now changed to Fe' and have also defined the term. Also we apologize the detritus and carbonate terms had been left out of the equations, which is now corrected.

- in equation 30, the production of opal is made proportional to the mortality of diatoms, which makes sense, but one term is missing, namely the 'bacterial loop' mortality term (mu\* x DT). What happens with the opal produced by diatoms that die by that pathway?

As with CaCO3, opal produced by diatom losses to the bacterial loop are not considered. This is because, the bacterial loop recycles only in the upper ocean and very rapidly, so a fast exchange of dissolved-to-particle forms should not significantly influence the distribution of opal down the water column. It might be worth looking into this assumption once explicit dissolution is implemented.

- line 223-224: it is unclear whether in 'average surface opal:free detritus export value of 1' the detritus is meant in units of N or C. My guess is that the 'value' (rather a 'ratio', I would say) is meant to be in Si:C units, which implies a Si:N of larger than 6, in the range of observations.

The units are now specified.

- But that made me wonder: what is the non-limiting Si:N ratio implied by equation 31? If Si>kSi and Fe>kFeDT, the Si:POC ratio is 0.5 and hence a Si:PON ratio is around 3. This seems high to me, as Si:N in most diatoms under non-limiting conditions is about one.

We adopted this parameterization from Aumont et al. (2003) and cite their paper as the original reference. Note that the kSi value is also dependent on the ambient Si concentration. A PhD student is currently testing this aspect of the model (the Si:N ratio) and will prepare a manuscript on it in the coming months.

- the authors have chosen to keep their habit to put some terms in square brackes, which have a meaning of a rate of change of a concentration (e.g. on the right-hand side of equation 39). Of course the authors are free to use any notation that they want, but I reiterate that this notation is a) inconsistent with the conventional usage of square brackets in chemical texts, where the square brackets mean 'concentration of', i.e. [Fe] = concentration of Fe, and b) that it is used inconsistently: In equation 31, for example, [Fe] means concentration of Fe in micromol/m^3, while in eq. 35 [Fe]\_sed means a flux in micromol/m^2/yr, and in eq. 39 [Fe]\_col means a rate of change of concentration in micromol Fe/m^3/yr.

We appreciate the concerns of the reviewer on this, but these changes were explicitly requested by an earlier reviewer, who has approved the manuscript.

- in eq. 38, again, the two numbers are not pure numbers, but have a unit.

**Units are now provided in the equation, rather than only in the text**

- in eq. 42, the alkalinity change is assumed to be proportional to the phosphate change with a proportionality factor of -R\_C:P. I do not understand the rationale for that factor. Yes, biogenic phosphate and nitrate uptake change alkalinity. In Wolf-Gladrow et al., 2007, it is stated that "assimilation of 1 mole of nitrogen (atoms) leads to (i) an increase of alkalinity by 1 mole when nitrate or nitrite is the N source, (ii) to a decrease of alkalinity by 1 mole when ammonia is used, and (iii) to no change of alkalinity when molecular nitrogen is the N source" and "Uptake of 1 mole of phosphate (H3PO4, H2PO4, HPO42–, or PO43–) by algae in accordance with the + nutrient-H -compensation principle increases alkalinity by 1 mole per mole P (...). Please note that the change of TA is independent of the phosphate species taken up by the cell". So I think the proportionality factor should be rather minus (Redfield ratio between N and P plus one).

**We thank the Reviewer for finding this error in our equation. It should read R\_N:P. The equation has been corrected.**

**References:**

Edwards, A. M., & Yool, A. (2000). The role of higher predation in plankton population models. Journal of Plankton Research, 22(6),1085–1112. https://doi.org/10.1093/plankt/22.6.1085

Steele, J.H. and Henderson, E.W. (1992) The role of predation in plankton models. Journal of Plankton Research, 14, 157–172.

Wolf-Gladrow, D. A., Zeebe, R. E., Klaas, C., Körtzinger, A., & Dickson, A. G. (2007). Total alkalinity: The explicit conservative expression and its application to biogeochemical processes. Marine Chemistry, 106(1-2 SPEC. ISS.), 287–300. https://doi.org/10.1016/j.marchem.2007.01.006

We thank the Reviewer for providing these references.

---

## Author Response (AR5)

Dear Dr. Yool,

Thank you for your patience with this manuscript revision. Please see our responses below.

Dear Authors,

Thank you for your revised manuscript and response to your referee.

I have examined both and find that most of the issues raised by the referee have been sufficiently addressed. However, I have listed below a few points where I believe that the manuscript could be made clearer:

- Regarding your statement "We can not plot the vertical distribution of sinking opal because we do not model this quantity explicitly", it should be possible, even with implicit variables, to plot the vertical distribution of sinking opal (and/or its dissolution). It may be that you have not stored this diagnostic output, but that is different. As the description of this process is clearly complicated, if you could plot a profile of the opal flux or its dissolution, that would help readers.
We have now included a new Fig. 2 which provides fluxes of implicit opal, opal dissolution, and loss to the seafloor.

- Regarding Table 6, the seafloor Si fluxes at the 3 depth-bands don't sum to the total. Since there's no overlap in the bands, this is a little surprising. What's the explanation?
This was a mistake on my part, for which I apologize. While I had a script for calculating the depth bands, it appears I did not use it when I entered the values and used a mix of k vs z gridding in Ferret, which produced inconsistencies. The numbers are corrected and sum to the total.

- Also regarding Table 6, the KMBM3 model appears to have completely flipped the dominance of phytoplankton from LP (96.2% to 5.5%) to diazotrophs (3.8% to 81.8%), but there is only slightly more N2-fixation (+25%). Is this the correct interpretation? While your text already acknowledges that your diazotrophs can uptake nitrate rather than fix N2, this isn't as clear as it could be when the phytoplankton types are introduced (ln. 75-78).
Yes this is correct. We have clarified this now at lines 75-78:
"slow growing phytoplankton which can fix nitrogen when necessary, including diazotrophs"

- Ln. 82: your amendment here is still ambiguous. Could you rephrase as "lost due to burial at the seafloor"? The referee rightly questions whether all material reaching the seafloor is "lost", or whether it is just a fraction of this.
I apologize that the amendment was inadequate. Burial is not modelled explicitly, but the model prescribes a loss based on opal flux rate. The language is changed to:
"is lost due to implicit burial at the seafloor"

- Ln. 118: Are you just describing Leibig's law here? That is, realised growth is a function of potential growth multiplied by the strongest limitation factor. If so, it may be worth simply noting this.
Yes, thank you. This is now explicitly stated:

"These equations are applied to obtain maximum possible growth rates as a function of temperature

and nutrients following Liebig's law of the minimum"

- Regarding eq. 8 and ln. 129, the amendments are OK, but would these read more straightforwardly if the numbers were expressed in mmol Si / m3 rather than mol Si / m3? Eq. 8 would then become something like ... k_si = 0.8 + 7.2 * ( [Si] / ( 30 + [Si] ) )

The units are changed back to mmol. Early on the units were in mmol, but I recall an early reviewer requested consistency.

- Regarding your statement "As with CaCO3, opal produced by diatom losses to the bacterial loop are not considered", perhaps it may help to add a qualification like the following: "In linking opal production to diatom mortality, our model essentially focuses on the importance of its export pathway in establishing vertical gradients. Opal that is not exported is effectively assumed to be recycled to dissolved silicate within the surface layer."
This is added as a new sentence (L 228)

As an aside, I also noticed that your revised manuscript also contains mark-up that should really only appear in the tracked-changes version.
I apologise for this. Two different versions are uploaded in this iteration. In the marked up version the colour for the most recent modifications are shown in blue.

If any of the above points are unclear, please get in contact with me. Hopefully these should only be minor amendments.
Thanks again for your patience and assistance with this manuscript.

Best wishes,
Karin

With best regards,

Andrew Yool

---

## Author Response (AR6)

Dear Dr. Yool,

Thank you for your efforts in editing our manuscript, the quality of which has been vastly improved by the diligence of both reviewers and yourself.

I have added a few sentences to the description of Fig. 2 in Section 2.2.7, page 9 line 252. I have also added another sentence to the caption of Fig. 2 explaining the units.

Best wishes,
Karin